# How Does Unlabeled Data Provably Help Out-of-Distribution Detection?

**Xuefeng Du**[1,*]**, Zhen Fang**[2,*]**, Ilias Diakonikolas**[1]**, Yixuan Li**[1]
[1]Department of Computer Sciences, University of Wisconsin-Madison
[2]Australian Artificial Intelligence Institute, University of Technology Sydney
{xfdu,ilias,sharonli}@cs.wisc.edu, zhen.fang@uts.edu.au

## Abstract

Using unlabeled data to regularize the machine learning models has demonstrated promise for improving safety and reliability in detecting out-of-distribution (OOD) data. Harnessing the power of unlabeled in-the-wild data is non-trivial due to the heterogeneity of both in-distribution (ID) and OOD data. This lack of a clean set of OOD samples poses significant challenges in learning an optimal OOD classifier. Currently, there is a lack of research on formally understanding how unlabeled data helps OOD detection. This paper bridges the gap by introducing a new learning framework SAL (**S**eparate **A**nd **L**earn) that offers both strong theoretical guarantees and empirical effectiveness. The framework separates candidate outliers from the unlabeled data and then trains an OOD classifier using the candidate outliers and the labeled ID data. Theoretically, we provide rigorous error bounds from the lens of separability and learnability, formally justifying the two components in our algorithm. Our theory shows that SAL can separate the candidate outliers with small error rates, which leads to a generalization guarantee for the learned OOD classifier. Empirically, SAL achieves state-of-the-art performance on common benchmarks, reinforcing our theoretical insights. Code is publicly available at https://github.com/deeplearning-wisc/sal.

## 1 Introduction

When deploying machine learning models in real-world environments, their safety and reliability are often challenged by the occurrence of out-of-distribution (OOD) data, which arise from unknown categories and should not be predicted by the model. Concerningly, neural networks are brittle and lack the necessary awareness of OOD data in the wild (Nguyen et al., 2015). Identifying OOD inputs is a vital but fundamentally challenging problem—the models are not explicitly exposed to the unknown distribution during training, and therefore cannot capture a reliable boundary between in-distribution (ID) vs. OOD data. To circumvent the challenge, researchers have started to explore training with additional data, which can facilitate a conservative and safe decision boundary against OOD data. In particular, a recent work by Katz-Samuels et al. (2022) proposed to leverage unlabeled data in the wild to regularize model training, while learning to classify labeled ID data. Such unlabeled wild data offer the benefits of being freely collectible upon deploying any machine learning model in its operating environment, and allow capturing the true test-time OOD distribution.

Despite the promise, harnessing the power of unlabeled wild data is non-trivial due to the heterogeneous mixture of ID and OOD data. This lack of a clean set of OOD training data poses significant challenges in designing effective OOD learning algorithms. Formally, the unlabeled data can be characterized by a Huber contamination model $\mathbb{P}_{\text{wild}} := (1 - \pi)\mathbb{P}_{\text{in}} + \pi\mathbb{P}_{\text{out}}$, where $\mathbb{P}_{\text{in}}$ and $\mathbb{P}_{\text{out}}$ are the marginal distributions of the ID and OOD data. It is important to note that the learner only observes samples drawn from such mixture distributions, without knowing the clear membership of whether being ID or OOD. Currently, a formalized understanding of the problem is lacking for the field. This prompts the question underlying the present work:

---

*Equal contributions

> *How does unlabeled wild data provably help OOD detection?*

**Algorithmic contribution.** In this paper, we propose a new learning framework SAL (**S**eparate **A**nd **L**earn), that effectively exploits the unlabeled wild data for OOD detection. At a high level, our framework SAL builds on two consecutive components: **(1)** filtering—separate *candidate outliers* from the unlabeled data, and **(2)** classification—learn an OOD classifier with the candidate outliers, in conjunction with the labeled ID data. To separate the candidate outliers, our key idea is to perform singular value decomposition on a gradient matrix, defined over all the unlabeled data whose gradients are computed based on a classification model trained on the clean labeled ID data. In the SAL framework, unlabeled wild data are considered candidate outliers when their projection onto the top singular vector exceeds a given threshold. The filtering strategy for identifying candidate outliers is theoretically supported by Theorem 1. We show in Section 3 (Remark 1) that under proper conditions, with a high probability, there exist some specific directions (e.g., the top singular vector direction) where the mean magnitude of the gradients for the wild outlier data is larger than that of ID data. After obtaining the outliers from the wild data, we train an OOD classifier that optimizes the classification between the ID vs. candidate outlier data for OOD detection.

**Theoretical significance.** Importantly, we provide new theories from the lens of *separability* and *learnability*, formally justifying the two components in our algorithm. Our main Theorem 1 analyzes the separability of outliers from unlabeled wild data using our filtering procedure, and gives a rigorous bound on the error rate. Our theory has practical implications. For example, when the size of the labeled ID data and unlabeled data is sufficiently large, Theorems 1 and 2 imply that the error rates of filtering outliers can be bounded by a small bias proportional to the optimal ID risk, which is a small value close to zero in reality (Frei et al., 2022). Based on the error rate estimation, we give a generalization error of the OOD classifier in Theorem 3, to quantify its learnability on the ID data and a noisy set of candidate outliers. Under proper conditions, the generalization error of the learned OOD classifier is upper bounded by the risk associated with the optimal OOD classifier.

**Empirical validation.** Empirically, we show that the generalization bound w.r.t. SAL (Theorem 3) indeed translates into strong empirical performance. SAL can be broadly applicable to non-convex models such as modern neural networks. We extensively evaluate SAL on common OOD detection tasks and establish state-of-the-art performance. For completeness, we compare SAL with two families of methods: (1) trained with only $\mathbb{P}_{\text{in}}$, and (2) trained with both $\mathbb{P}_{\text{in}}$ and an unlabeled dataset. On CIFAR-100, compared to a strong baseline KNN+ (Sun et al., 2022) using only $\mathbb{P}_{\text{in}}$, SAL outperforms by 44.52% (FPR95) on average. While methods such as Outlier Exposure (Hendrycks et al., 2019) require a clean set of auxiliary unlabeled data, our results are achieved without imposing any such assumption on the unlabeled data and hence offer stronger flexibility. Compared to the most related baseline WOODS (Katz-Samuels et al., 2022), our framework can reduce the FPR95 from 7.80% to 1.88% on CIFAR-100, establishing near-perfect results on this challenging benchmark.

## 2 PROBLEM SETUP

Formally, we describe the data setup, models and losses and learning goal.

**Labeled ID data and ID distribution.** Let $\mathcal{X}$ be the input space, and $\mathcal{Y} = \{1, ..., K\}$ be the label space for ID data. Given an unknown ID joint distribution $\mathbb{P}_{\mathcal{XY}}$ defined over $\mathcal{X} \times \mathcal{Y}$, the labeled ID data $\mathcal{S}^{\text{in}} = \{(\mathbf{x}_1, y_1), ..., (\mathbf{x}_n, y_n)\}$ are drawn independently and identically from $\mathbb{P}_{\mathcal{XY}}$. We also denote $\mathbb{P}_{\text{in}}$ as the marginal distribution of $\mathbb{P}_{\mathcal{XY}}$ on $\mathcal{X}$, which is referred to as the ID distribution.

**Out-of-distribution detection.** Our framework concerns a common real-world scenario in which the algorithm is trained on the labeled ID data, but will then be deployed in environments containing OOD data from unknown class, i.e., $y \notin \mathcal{Y}$, and therefore should not be predicted by the model. At test time, the goal is to decide whether a test-time input is from ID or not (OOD).

**Unlabeled wild data.** A key challenge in OOD detection is the lack of labeled OOD data. In particular, the sample space for potential OOD data can be prohibitively large, making it expensive to collect labeled OOD data. In this paper, to model the realistic environment, we incorporate unlabeled wild data $\mathcal{S}_{\text{wild}} = \{\tilde{\mathbf{x}}_1, ..., \tilde{\mathbf{x}}_m\}$ into our learning framework. Wild data consists of both ID and OOD data, and can be collected freely upon deploying an existing model trained on $\mathcal{S}^{\text{in}}$. Following Katz-Samuels et al. (2022), we use the Huber contamination model to characterize the

marginal distribution of the wild data

$$\mathbb{P}_{\text{wild}} := (1 - \pi)\mathbb{P}_{\text{in}} + \pi\mathbb{P}_{\text{out}}, \tag{1}$$

where $\pi \in (0, 1]$ and $\mathbb{P}_{\text{out}}$ is the OOD distribution defined over $\mathcal{X}$. Note that the case $\pi = 0$ is straightforward since no novelties occur.

**Models and losses.** We denote by $\mathbf{h_w} : \mathcal{X} \mapsto \mathbb{R}^K$ a predictor for ID classification with parameter $\mathbf{w} \in \mathcal{W}$, where $\mathcal{W}$ is the parameter space. $\mathbf{h_w}$ returns the soft classification output. We consider the loss function $\ell : \mathbb{R}^K \times \mathcal{Y} \mapsto \mathbb{R}$ on the labeled ID data. In addition, we denote the OOD classifier $\mathbf{g_\theta} : \mathcal{X} \mapsto \mathbb{R}$ with parameter $\boldsymbol{\theta} \in \Theta$, where $\Theta$ is the parameter space. We use $\ell_{\text{b}}(\mathbf{g_\theta}(\mathbf{x}), y_{\text{b}})$ to denote the binary loss function *w.r.t.* $\mathbf{g_\theta}$ and binary label $y_{\text{b}} \in \mathcal{Y}_{\text{b}} := \{y_+, y_-\}$, where $y_+ \in \mathbb{R}_{>0}$ and $y_- \in \mathbb{R}_{<0}$ correspond to the ID class and the OOD class, respectively.

**Learning goal.** Our learning framework aims to build the OOD classifier $\mathbf{g_\theta}$ by leveraging data from both $\mathcal{S}^{\text{in}}$ and $\mathcal{S}_{\text{wild}}$. In evaluating our model, we are interested in the following measurements:

$$\begin{aligned} &(1) \downarrow \text{FPR}(\mathbf{g_\theta}; \lambda) := \mathbb{E}_{\mathbf{x} \sim \mathbb{P}_{\text{out}}}(\mathbb{1}\{\mathbf{g_\theta}(\mathbf{x}) > \lambda\}), \\ &(2) \uparrow \text{TPR}(\mathbf{g_\theta}; \lambda) := \mathbb{E}_{\mathbf{x} \sim \mathbb{P}_{\text{in}}}(\mathbb{1}\{\mathbf{g_\theta}(\mathbf{x}) > \lambda\}), \end{aligned} \tag{2}$$

where $\lambda$ is a threshold, typically chosen so that a high fraction of ID data is correctly classified.

## 3 PROPOSED METHODOLOGY

In this section, we introduce a new learning framework SAL that performs OOD detection by leveraging the unlabeled wild data. The framework offers substantial advantages over the counterpart approaches that rely only on the ID data, and naturally suits many applications where machine learning models are deployed in the open world. SAL has two integral components: **(1)** filtering—separate the candidate outlier data from the unlabeled wild data (Section 3.1), and **(2)** classification—train a binary OOD classifier with the ID data and candidate outliers (Section 3.2). In Section 4, we provide theoretical guarantees for SAL, provably justifying the two components in our method.

### 3.1 SEPARATING CANDIDATE OUTLIERS FROM THE WILD DATA

To separate candidate outliers from the wild mixture $\mathcal{S}_{\text{wild}}$, our framework employs a level-set estimation based on the gradient information. The gradients are estimated from a classification predictor $\mathbf{h_w}$ trained on the ID data $\mathcal{S}^{\text{in}}$. We describe the procedure formally below.

**Estimating the reference gradient from ID data.** To begin with, SAL estimates the reference gradients by training a classifier $\mathbf{h_w}$ on the ID data $\mathcal{S}^{\text{in}}$ by empirical risk minimization (ERM):

$$\mathbf{w}_{\mathcal{S}^{\text{in}}} \in \arg\min_{\mathbf{w} \in \mathcal{W}} R_{\mathcal{S}^{\text{in}}}(\mathbf{h_w}), \quad \text{where} \quad R_{\mathcal{S}^{\text{in}}}(\mathbf{h_w}) = \frac{1}{n} \sum_{(\mathbf{x}_i, y_i) \in \mathcal{S}^{\text{in}}} \ell(\mathbf{h_w}(\mathbf{x}_i), y_i), \tag{3}$$

$\mathbf{w}_{\mathcal{S}^{\text{in}}}$ is the learned parameter and $n$ is the size of ID training set $\mathcal{S}^{\text{in}}$. The average gradient $\bar{\nabla}$ is

$$\bar{\nabla} = \frac{1}{n} \sum_{(\mathbf{x}_i, y_i) \in \mathcal{S}^{\text{in}}} \nabla\ell(\mathbf{h}_{\mathbf{w}_{\mathcal{S}^{\text{in}}}}(\mathbf{x}_i), y_i), \tag{4}$$

where $\bar{\nabla}$ acts as a reference gradient that allows measuring the deviation of any other points from it.

**Separate candidate outliers from the unlabeled wild data.** After training the classification predictor on the labeled ID data, we deploy the trained predictor $\mathbf{h}_{\mathbf{w}_{\mathcal{S}^{\text{in}}}}$ in the wild, and naturally receives data $\mathcal{S}_{\text{wild}}$—a mixture of unlabeled ID and OOD data. Key to our framework, we perform a filtering procedure on the wild data $\mathcal{S}_{\text{wild}}$, identifying candidate outliers based on a filtering score. To define the filtering score, we represent each point in $\mathcal{S}_{\text{wild}}$ as a gradient vector, relative to the reference gradient $\bar{\nabla}$. Specifically, we calculate the gradient matrix (after subtracting the reference gradient $\bar{\nabla}$) for the wild data as follows:

$$\mathbf{G} = \begin{bmatrix} \nabla\ell(\mathbf{h}_{\mathbf{w}_{\mathcal{S}^{\text{in}}}}(\tilde{\mathbf{x}}_1), \hat{y}_{\tilde{\mathbf{x}}_1}) - \bar{\nabla} \\ \dots \\ \nabla\ell(\mathbf{h}_{\mathbf{w}_{\mathcal{S}^{\text{in}}}}(\tilde{\mathbf{x}}_m), \hat{y}_{\tilde{\mathbf{x}}_m}) - \bar{\nabla} \end{bmatrix}^\top, \tag{5}$$

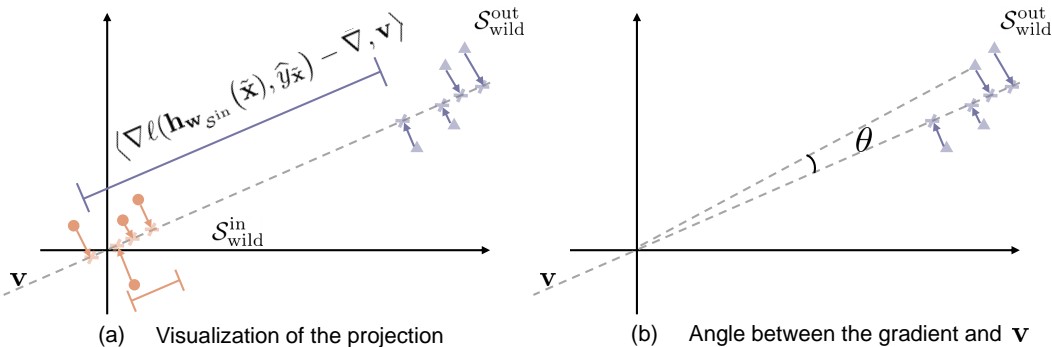

(a)   Visualization of the projection                    (b)   Angle between the gradient and $\mathbf{v}$

Figure 1: (a) Visualization of the gradient vectors, and their projection onto the top singular vector $\mathbf{v}$ (in gray dashed line). The gradients of inliers from $\mathcal{S}_{\text{wild}}^{\text{in}}$ (colored in orange) are close to the origin (reference gradient $\bar{\nabla}$). In contrast, the gradients of outliers from $\mathcal{S}_{\text{wild}}^{\text{out}}$ (colored in purple) are farther away.    (b) The angle $\theta$ between the gradient of set $\mathcal{S}_{\text{wild}}^{\text{out}}$ and the singular vector $\mathbf{v}$. Since $\mathbf{v}$ is searched to maximize the distance from the projected points (cross marks) to the origin (sum over all the gradients in $\mathcal{S}_{\text{wild}}$), $\mathbf{v}$ points to the direction of OOD data in the wild with a small $\theta$. This further translates into a high filtering score $\tau$, which is essentially the norm after projecting a gradient vector onto $\mathbf{v}$. As a result, filtering outliers by $\mathcal{S}_T = \{\tilde{\mathbf{x}}_i \in \mathcal{S}_{\text{wild}} : \tau_i > T\}$ will approximately return the purple OOD samples in the wild data.

where $m$ denotes the size of the wild data, and $\widehat{y}_{\tilde{\mathbf{x}}}$ is the predicted label for a wild sample $\tilde{\mathbf{x}}$. For each data point $\tilde{\mathbf{x}}_i$ in $\mathcal{S}_{\text{wild}}$, we then define our filtering score as follows:

$$\tau_i = \left\langle \nabla \ell(\mathbf{h}_{\mathbf{w}_{\mathcal{S}^{\text{in}}}}(\tilde{\mathbf{x}}_i), \widehat{y}_{\tilde{\mathbf{x}}_i}) - \bar{\nabla}, \mathbf{v} \right\rangle^2, \tag{6}$$

where $\langle \cdot, \cdot \rangle$ is the dot product operator and $\mathbf{v}$ is the top singular vector of $\mathbf{G}$. The top singular vector $\mathbf{v}$ can be regarded as the principal component of the matrix $\mathbf{G}$ in Eq. 5, which maximizes the total distance from the projected gradients (onto the direction of $\mathbf{v}$) to the origin (sum over all points in $\mathcal{S}_{\text{wild}}$) (Hotelling, 1933). Specifically, $\mathbf{v}$ is a unit-norm vector and can be computed as follows:

$$\mathbf{v} \in \underset{\|\mathbf{u}\|_2=1}{\arg\max} \sum_{\tilde{\mathbf{x}}_i \in \mathcal{S}_{\text{wild}}} \left\langle \mathbf{u}, \nabla \ell(\mathbf{h}_{\mathbf{w}_{\mathcal{S}^{\text{in}}}}(\tilde{\mathbf{x}}_i), \widehat{y}_{\tilde{\mathbf{x}}_i}) - \bar{\nabla} \right\rangle^2. \tag{7}$$

Essentially, the filtering score $\tau_i$ in Eq. 6 measures the $\ell_2$ norm of the projected vector. To help readers better understand our design rationale, we provide an illustrative example of the gradient vectors and their projections in Figure 1 (see caption for details). Theoretically, Remark 1 below shows that the projection of the OOD gradient vector to the top singular vector of the gradient matrix $\mathbf{G}$ is on average provably larger than that of the ID gradient vector, which rigorously justifies our idea of using the score $\tau$ for separating the ID and OOD data.

**Remark 1.** *Theorem 4 in Appendix D.1 has shown that under proper assumptions, if we have sufficient data and large-size model, then with the high probability:*

- *the mean projected magnitude of OOD gradients in the direction of the top singular vector of $\mathbf{G}$ can be lower bounded by a positive constant $C/\pi$;*

- *the mean projected magnitude of ID gradients in the direction of the top singular vector is upper bounded by a small value close to zero.*

Finally, we regard $\mathcal{S}_T = \{\tilde{\mathbf{x}}_i \in \mathcal{S}_{\text{wild}} : \tau_i > T\}$ as the (potentially noisy) candidate outlier set, where $T$ is the filtering threshold. The threshold can be chosen on the ID data $\mathcal{S}^{\text{in}}$ so that a high fraction (e.g., 95%) of ID samples is below it. In Section 4, we will provide formal guarantees, rigorously justifying that the set $\mathcal{S}_T$ returns outliers with a large probability. We discuss and compare with alternative gradient-based scores (e.g., GradNorm (Huang et al., 2021)) for filtering in Section 5.2. In Appendix K, we discuss the variants of using multiple singular vectors, which yield similar results.

**An illustrative example of algorithm effect.** To see the effectiveness of our filtering score, we test on two simulations in Figure 2 (a). These simulations are constructed with simplicity in mind, to facilitate understanding. Evaluations on complex high-dimensional data will be provided in Section 5. In particular, the wild data is a mixture of ID (multivariate Gaussian with three classes) and OOD. We consider two scenarios of OOD distribution, with ground truth colored in purple. Figure 2

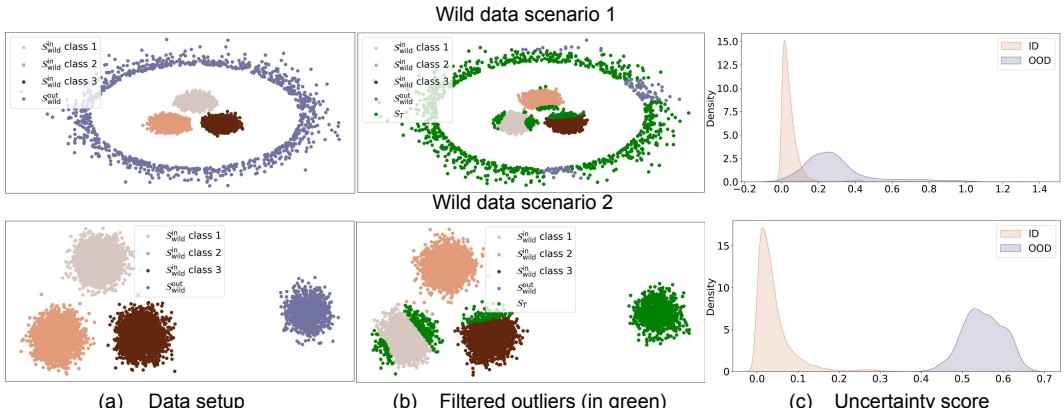

Figure 2: Example of SAL on two different scenarios of the unlabeled wild data. (a) Setup of the ID/inlier $\mathcal{S}_{\text{wild}}^{\text{in}}$ and OOD/outlier data $\mathcal{S}_{\text{wild}}^{\text{out}}$ in the wild. The inliers are sampled from three multivariate Gaussians. We construct two different distributions of outliers (see details in Appendix Q). (b) The filtered outliers (in green) by SAL, where the error rate of filtered outliers $\mathcal{S}_T$ containing inlier data is $8.4\%$ and $6.4\%$, respectively. (c) The density distribution of the filtering score $\tau$, which is separable for inlier and outlier data in the wild and thus benefits the training of the OOD classifier leveraging the filtered outlier data for binary classification.

(b) exemplifies the outliers (in green) identified using our proposed method, which largely aligns with the ground truth. The error rate of $\mathcal{S}_T$ containing ID data is only $8.4\%$ and $6.4\%$ for the two scenarios considered. Moreover, the filtering score distribution displays a clear separation between the ID vs. OOD parts, as evidenced in Figure 2 (c).

**Remark 2.** *Our filtering process can be easily extended into $K$-class classification. In this case, one can maintain a class-conditional reference gradient $\bar{\nabla}_k$, one for each class $k \in [1, K]$, estimated on ID data belonging to class $k$, which captures the characteristics for each ID class. Similarly, the top singular vector computation can also be performed in a class-conditional manner, where we replace the gradient matrix with the class-conditional $\mathbf{G}_k$, containing gradient vectors of wild samples being predicted as class $k$.*

## 3.2 Training the OOD Classifier with the Candidate Outliers

After obtaining the candidate outlier set $\mathcal{S}_T$ from the wild data, we train an OOD classifier $\mathbf{g}_{\boldsymbol{\theta}}$ that optimizes for the separability between the ID vs. candidate outlier data. In particular, our training objective can be viewed as explicitly optimizing the level-set based on the model output (threshold at 0), where the labeled ID data $\mathbf{x}$ from $\mathcal{S}^{\text{in}}$ has positive values and vice versa.

$$R_{\mathcal{S}^{\text{in}}, \mathcal{S}_T}(\mathbf{g}_{\boldsymbol{\theta}}) = R_{\mathcal{S}^{\text{in}}}^{+}(\mathbf{g}_{\boldsymbol{\theta}}) + R_{\mathcal{S}_T}^{-}(\mathbf{g}_{\boldsymbol{\theta}})$$
$$= \mathbb{E}_{\mathbf{x} \in \mathcal{S}^{\text{in}}} \ \mathbb{1}\{\mathbf{g}_{\boldsymbol{\theta}}(\mathbf{x}) \leq 0\} + \mathbb{E}_{\tilde{\mathbf{x}} \in \mathcal{S}_T} \ \mathbb{1}\{\mathbf{g}_{\boldsymbol{\theta}}(\tilde{\mathbf{x}}) > 0\}. \tag{8}$$

To make the $0/1$ loss tractable, we replace it with the binary sigmoid loss, a smooth approximation of the $0/1$ loss. We train $\mathbf{g}_{\boldsymbol{\theta}}$ along with the ID risk in Eq. 3 to ensure ID accuracy. Notably, the training enables strong generalization performance for test OOD samples drawn from $\mathbb{P}_{\text{out}}$. We provide formal guarantees on the generalization bound in Theorem 3, as well as empirical support in Section 5. A pseudo algorithm of SAL is in Appendix (see Algorithm 1).

## 4 Theoretical Analysis

We now provide theory to support our proposed algorithm. Our main theorems justify the two components in our algorithm. As an overview, Theorem 1 provides a provable bound on the error rates using our filtering procedure. Based on the estimations on error rates, Theorem 3 gives the generalization bound *w.r.t.* the empirical OOD classifier $\mathbf{g}_{\boldsymbol{\theta}}$, learned on ID data and noisy set of outliers. We specify several mild assumptions and necessary notations for our theorems in Appendix B. Due to space limitation, we omit unimportant constants and simplify the statements of our theorems. We defer the **full formal** statements in Appendix C. All proofs can be found in Appendices D and E.

### 4.1 ANALYSIS ON SEPARABILITY

Our main theorem quantifies the separability of the outliers in the wild by using the filtering procedure (c.f. Section 3.1). Let $\text{ERR}_{\text{out}}$ and $\text{ERR}_{\text{in}}$ be the error rate of OOD data being regarded as ID and the error rate of ID data being regarded as OOD, i.e., $\text{ERR}_{\text{out}} = |\{\tilde{\mathbf{x}}_i \in \mathcal{S}_{\text{wild}}^{\text{out}} : \tau_i \leq T\}|/|\mathcal{S}_{\text{wild}}^{\text{out}}|$ and $\text{ERR}_{\text{in}} = |\{\tilde{\mathbf{x}}_i \in \mathcal{S}_{\text{wild}}^{\text{in}} : \tau_i > T\}|/|\mathcal{S}_{\text{wild}}^{\text{in}}|$, where $\mathcal{S}_{\text{wild}}^{\text{in}}$ and $\mathcal{S}_{\text{wild}}^{\text{out}}$ denote the sets of inliers and outliers from the wild data $\mathcal{S}_{\text{wild}}$. Then $\text{ERR}_{\text{out}}$ and $\text{ERR}_{\text{in}}$ have the following generalization bounds.

---

**Theorem 1.** *(Informal). Under mild conditions, if $\ell(\mathbf{h}_{\mathbf{w}}(\mathbf{x}), y)$ is $\beta_1$-smooth w.r.t. $\mathbf{w}$, $\mathbb{P}_{wild}$ has $(\gamma, \zeta)$-discrepancy w.r.t. $\mathbb{P}_{\mathcal{X}\mathcal{Y}}$ (c.f. Appendices B.2, B.3), and there is $\eta \in (0, 1)$ s.t. $\Delta = (1 - \eta)^2\zeta^2 - 8\beta_1 R_{in}^* > 0$, then when $n = \Omega(d/\min\{\eta^2\Delta, (\gamma - R_{in}^*)^2\})$, $m = \Omega(d/\eta^2\zeta^2)$, with the probability at least $0.9$, for $0 < T < 0.9M'$ ($M'$ is the upper bound of score $\tau_i$),*

$$\text{ERR}_{in} \leq \frac{8\beta_1}{T} R_{in}^* + O\left(\frac{1}{T}\sqrt{\frac{d}{n}}\right) + O\left(\frac{1}{T}\sqrt{\frac{d}{(1 - \pi)m}}\right), \tag{9}$$

$$\text{ERR}_{out} \leq \delta(T) + O\left(\sqrt{\frac{d}{\pi^2 n}}\right) + O\left(\sqrt{\frac{\max\{d, \Delta_\zeta^{\eta^2}/\pi^2\}}{\pi^2(1 - \pi)m}}\right), \tag{10}$$

*where $R_{in}^*$ is the optimal ID risk, i.e., $R_{in}^* = \min_{\mathbf{w} \in \mathcal{W}} \mathbb{E}_{(\mathbf{x}, y) \sim \mathbb{P}_{\mathcal{X}\mathcal{Y}}}\ell(\mathbf{h}_{\mathbf{w}}(\mathbf{x}), y)$,*

$$\delta(T) = \max\{0, 1 - \Delta_\zeta^\eta/\pi\}/(1 - T/M'), \quad \Delta_\zeta^\eta = 0.98\eta^2\zeta^2 - 8\beta_1 R_{in}^*, \tag{11}$$

*$d$ is the dimension of the space $\mathcal{W}$, and $\pi$ is the OOD class-prior probability in the wild.*

---

**Practical implications of Theorem 1.** The above theorem states that under mild assumptions, the errors $\text{ERR}_{\text{out}}$ and $\text{ERR}_{\text{in}}$ are upper bounded. For $\text{ERR}_{\text{in}}$, if the following two regulatory conditions hold: 1) the sizes of the labeled ID $n$ and wild data $m$ are sufficiently large; 2) the optimal ID risk $R_{\text{in}}^*$ is small, then the upper bound is tight. For $\text{ERR}_{\text{out}}$, $\delta(T)$ defined in Eq. 11 becomes the main error, if we have sufficient data. To further study the main error $\delta(T)$ in Eq. 10, Theorem 2 shows that the error $\delta(T)$ could be close to zero under practical conditions.

---

**Theorem 2.** *(Informal). 1) If $\Delta_\zeta^\eta \geq (1 - \epsilon)\pi$ for a small error $\epsilon \geq 0$, then the main error $\delta(T)$ defined in Eq. 11 satisfies that*

$$\delta(T) \leq \frac{\epsilon}{1 - T/M'}. \tag{12}$$

*2) If $\zeta \geq 2.011\sqrt{8\beta_1 R_{in}^*} + 1.011\sqrt{\pi}$, then there exists $\eta \in (0, 1)$ ensuring that $\Delta > 0$ and $\Delta_\zeta^\eta > \pi$ hold, which implies that the main error $\delta(T) = 0$.*

---

**Practical implications of Theorem 2.** Theorem 2 states that if the discrepancy $\zeta$ between two data distributions $\mathbb{P}_{\text{wild}}$ and $\mathbb{P}_{\text{in}}$ is larger than some small values, the main error $\delta(T)$ could be close to zero. Therefore, by combining with the two regulatory conditions mentioned in Theorem 1, the error $\text{ERR}_{\text{out}}$ could be close to zero. Empirically, we verify the conditions of Theorem 2 in Appendix F, which can hold true easily in practice. In addition, given fixed optimal ID risk $R_{\text{in}}^*$ and fixed sizes of the labeled ID $n$ and wild data $m$, we observe that the bound of $\text{ERR}_{\text{in}}$ will increase when $\pi$ goes from 0 to 1. In contrast, the bound of $\text{ERR}_{\text{out}}$ is non-monotonic when $\pi$ increases, which will firstly decrease and then increase. The observations align well with empirical results in Appendix F.

**Impact of using predicted labels for the wild data.** Recall in Section 3.1 that the filtering step uses the predicted labels to estimate the gradient for wild data, which is unlabeled. To analyze the impact theoretically, we show in Appendix Assumption 2 that the loss incurred by using the predicted label is smaller than the loss by using any label in the label space. This property is included in Appendix Lemmas 5 and 6 to constrain the filtering score in Appendix Theorem 5 and then filtering error in Theorem 1. In harder classification cases, the predicted label deviates more from the true label for the wild ID data, which leads to a looser bound for the filtering accuracy in Theorem 1.

Empirically, we calculate and compare the filtering accuracy and its OOD detection result on CIFAR-10 and CIFAR-100 ( TEXTURES (Cimpoi et al., 2014) as the wild OOD). SAL achieves a result of $\text{ERR}_{\text{in}} = 0.018$ and $\text{ERR}_{\text{out}} = 0.17$ on CIFAR-10 (easier classification case), which outperforms the result of $\text{ERR}_{\text{in}} = 0.037$ and $\text{ERR}_{\text{out}} = 0.30$ on CIFAR-100 (harder classification case), aligning with our reasoning above. The experimental details are provided in Appendix P. Analysis of using random labels for the wild data is provided in Appendix O.

## 4.2 ANALYSIS ON LEARNABILITY

Leveraging the filtered outliers $\mathcal{S}_T$, SAL then trains an OOD classifier $\mathbf{g}_{\boldsymbol{\theta}}$ with the data from in-distribution $\mathcal{S}^{\text{in}}$ and data from $\mathcal{S}_T$ as OOD. In this section, we provide the generalization error bound for the learned OOD classifier to quantify its learnability. Specifically, we show that a small error guarantee in Theorem 1 implies that we can get a tight generalization error bound.

---

**Theorem 3.** *(Informal). Let $L$ be the upper bound of $\ell_b(\mathbf{g}_{\boldsymbol{\theta}}(\mathbf{x}), y_b)$, i.e., $\ell_b(\mathbf{g}_{\boldsymbol{\theta}}(\mathbf{x}), y_b) \leq L$. Under conditions in Theorem 1, if we further require $n = \Omega(d/\min\{\pi, \Delta_\zeta^\eta\}^2)$, $m = \Omega((d + \Delta_\zeta^\eta)/(\pi^2(1-\pi)\min\{\pi, \Delta_\zeta^\eta\}^2))$,*
*then with the probability at least $0.89$, for any $0 < T < 0.9M' \min\{1, \Delta_\zeta^\eta/\pi\}$, the OOD classifier $\mathbf{g}_{\widehat{\boldsymbol{\theta}}_T}$ learned by SAL satisfies*

$$R_{\mathbb{P}_{in}, \mathbb{P}_{out}}(\mathbf{g}_{\widehat{\boldsymbol{\theta}}_T}) \leq \min_{\boldsymbol{\theta} \in \Theta} R_{\mathbb{P}_{in}, \mathbb{P}_{out}}(\mathbf{g}_{\boldsymbol{\theta}}) + \frac{3.5L}{1 - \delta(T)}\delta(T) + \frac{9(1-\pi)L\beta_1}{\pi(1 - \delta(T))T}R_{in}^*$$
$$+ O\left(\frac{\max\{\sqrt{d}, \sqrt{d'}\}}{\min\{\pi, \Delta_\zeta^\eta\}T'}\sqrt{\frac{1}{n}}\right) + O\left(\frac{\max\{\sqrt{d}, \sqrt{d'}, \Delta_\zeta^\eta\}}{\min\{\pi, \Delta_\zeta^\eta\}T'}\sqrt{\frac{1}{\pi^2(1-\pi)m}}\right), \tag{13}$$

*where $\Delta_\zeta^\eta$, $d$ and $\pi$ are shown in Theorem 1, $d'$ is the dimension of space $\Theta$, $T' = T/(1+T)$, and the risk $R_{\mathbb{P}_{in}, \mathbb{P}_{out}}(\mathbf{g}_{\boldsymbol{\theta}})$ corresponds to the empirical risk in Eq. 8 with loss $\ell_b$, i.e.,*

$$R_{\mathbb{P}_{in}, \mathbb{P}_{out}}(\mathbf{g}_{\widehat{\boldsymbol{\theta}}_T}) = \mathbb{E}_{\mathbf{x} \sim \mathbb{P}_{in}}\ell_b(\mathbf{g}_{\boldsymbol{\theta}}(\mathbf{x}), y_+) + \mathbb{E}_{\mathbf{x} \sim \mathbb{P}_{out}}\ell_b(\mathbf{g}_{\boldsymbol{\theta}}(\mathbf{x}), y_-). \tag{14}$$

---

**Insights.** The above theorem presents the generalization error bound of the OOD classifier $\mathbf{g}_{\widehat{\boldsymbol{\theta}}_T}$ learned by using the filtered OOD data $\mathcal{S}_T$. When we have sufficient labeled ID data and wild data, then the risk of the OOD classifier $\mathbf{g}_{\widehat{\boldsymbol{\theta}}_T}$ is close to the optimal risk, i.e., $\min_{\boldsymbol{\theta} \in \Theta} R_{\mathbb{P}_{in}, \mathbb{P}_{out}}(\mathbf{g}_{\boldsymbol{\theta}})$, if the optimal ID risk $R_{\text{in}}^*$ is small, and either one of the conditions in Theorem 2 is satisfied.

## 5 EXPERIMENTS

In this section, we verify the effectiveness of our algorithm on modern neural networks. We aim to show that the generalization bound of the OOD classifier (Theorem 3) indeed translates into strong empirical performance, establishing state-of-the-art results (Section 5.2).

### 5.1 EXPERIMENTAL SETUP

**Datasets.** We follow exactly the same experimental setup as WOODS (Katz-Samuels et al., 2022), which introduced the problem of learning OOD detectors with wild data. This allows us to draw fair comparisons. WOODS considered CIFAR-10 and CIFAR-100 (Krizhevsky et al., 2009) as ID datasets ($\mathbb{P}_{\text{in}}$). For OOD test datasets ($\mathbb{P}_{\text{out}}$), we use a suite of natural image datasets including TEXTURES (Cimpoi et al., 2014), SVHN (Netzer et al., 2011), PLACES365 (Zhou et al., 2017), LSUN-RESIZE & LSUN-C (Yu et al., 2015). To simulate the wild data ($\mathbb{P}_{\text{wild}}$), we mix a subset of ID data (as $\mathbb{P}_{\text{in}}$) with the outlier dataset (as $\mathbb{P}_{\text{out}}$) under the default $\pi = 0.1$, which reflects the practical scenario that most data would remain ID. Take SVHN as an example, we use CIFAR+SVHN as the unlabeled wild data and test on SVHN as OOD. We simulate this for all OOD datasets and provide analysis of differing $\pi \in \{0.05, 0.1, ..., 1.0\}$ in Appendix F. Note that we split CIFAR datasets into two halves: $25,000$ images as ID training data, and the remainder $25,000$ for creating the wild mixture data. We use the weights from the penultimate layer for gradient calculation, which was shown to be the most informative for OOD detection (Huang et al., 2021). Experimental details are provided in Appendix G.

Table 1: OOD detection performance on CIFAR-100 as ID. All methods are trained on Wide ResNet-40-2 for 100 epochs. For each dataset, we create corresponding wild mixture distribution $\mathbb{P}_{wild} = (1 - \pi)\mathbb{P}_{in} + \pi\mathbb{P}_{out}$ for training and test on the corresponding OOD dataset. Values are percentages averaged over 10 runs. Bold numbers highlight the best results. Table format credit to Katz-Samuels et al. (2022).

| | OOD Datasets | | | | | | | | | | | | ID ACC |
| Methods | SVHN | | PLACES365 | | LSUN-C | | LSUN-RESIZE | | TEXTURES | | Average | | |
| | FPR95 | AUROC | FPR95 | AUROC | FPR95 | AUROC | FPR95 | AUROC | FPR95 | AUROC | FPR95 | AUROC | |
| | | | | | | With $\mathbb{P}_{in}$ only | | | | | | | |
| MSP | 84.59 | 71.44 | 82.84 | 73.78 | 66.54 | 83.79 | 82.42 | 75.38 | 83.29 | 73.34 | 79.94 | 75.55 | 75.96 |
| ODIN | 84.66 | 67.26 | 87.88 | 71.63 | 55.55 | 87.73 | 71.96 | 81.82 | 79.27 | 73.45 | 75.86 | 76.38 | 75.96 |
| Mahalanobis | 57.52 | 86.01 | 88.83 | 67.87 | 91.18 | 69.69 | 21.23 | 96.00 | 39.39 | 90.57 | 59.63 | 82.03 | 75.96 |
| Energy | 85.82 | 73.99 | 80.56 | 75.44 | 35.32 | 93.53 | 79.47 | 79.23 | 79.41 | 76.28 | 72.12 | 79.69 | 75.96 |
| KNN | 66.38 | 83.76 | 79.17 | 71.91 | 70.96 | 83.71 | 77.83 | 78.85 | 88.00 | 67.19 | 76.47 | 77.08 | 75.96 |
| ReAct | 74.33 | 88.04 | 81.33 | 74.32 | 39.30 | 91.19 | 79.86 | 73.69 | 67.38 | 82.80 | 68.44 | 82.01 | 75.96 |
| DICE | 88.35 | 72.58 | 81.61 | 75.07 | 26.77 | 94.74 | 80.21 | 78.50 | 76.29 | 76.07 | 70.65 | 79.39 | 75.96 |
| ASH | 21.36 | 94.28 | 68.37 | 71.22 | 15.27 | 95.65 | 68.18 | 85.42 | 40.87 | 92.29 | 42.81 | 87.77 | 75.96 |
| CSI | 64.70 | 84.97 | 82.25 | 73.63 | 38.10 | 92.52 | 91.55 | 63.42 | 74.70 | 92.66 | 70.26 | 81.44 | 69.90 |
| KNN+ | 32.21 | 93.74 | 68.30 | 75.31 | 40.37 | 86.13 | 44.86 | 88.88 | 46.26 | 87.40 | 46.40 | 86.29 | 73.78 |
| | | | | | | With $\mathbb{P}_{in}$ and $\mathbb{P}_{wild}$ | | | | | | | |
| OE | 1.57 | 99.63 | 60.24 | 83.43 | 3.83 | 99.26 | 0.93 | 99.79 | 27.89 | 93.35 | 18.89 | 95.09 | 71.65 |
| Energy (w/ OE) | 1.47 | 99.68 | 54.67 | 86.09 | 2.52 | 99.44 | 2.68 | 99.50 | 37.26 | 91.26 | 19.72 | 95.19 | 73.46 |
| WOODS | 0.12 | **99.96** | 29.58 | 90.60 | 0.11 | **99.96** | 0.07 | **99.96** | 9.12 | 96.65 | 7.80 | 97.43 | 75.22 |
| SAL | **0.07** | 99.95 | **3.53** | **99.06** | **0.06** | 99.94 | **0.02** | 99.95 | **5.73** | **98.65** | **1.88** | **99.51** | 73.71 |
| (Ours) | ±0.02 | ±0.00 | ±0.17 | ±0.06 | ±0.01 | ±0.21 | ±0.00 | ±0.03 | ±0.34 | ±0.02 | ±0.11 | ±0.02 | ±0.78 |

**Evaluation metrics.** We report the following metrics: (1) the false positive rate (FPR95↓) of OOD samples when the true positive rate of ID samples is 95%, (2) the area under the receiver operating characteristic curve (AUROC↑), and (3) ID classification Accuracy (ID ACC↑).

## 5.2 EMPIRICAL RESULTS

**SAL achieves superior empirical performance.** We present results in Table 1 on CIFAR-100, where SAL outperforms the state-of-the-art method. Our comparison covers an extensive collection of competitive OOD detection methods, which can be divided into two categories: trained with and without the wild data. For methods using ID data $\mathbb{P}_{in}$ only, we compare with methods such as MSP (Hendrycks & Gimpel, 2017), ODIN (Liang et al., 2018), Mahalanobis distance (Lee et al., 2018b), Energy score (Liu et al., 2020b), ReAct (Sun et al., 2021), DICE (Sun & Li, 2022), KNN distance (Sun et al., 2022), and ASH (Djurisic et al., 2023)—all of which use a model trained with cross-entropy loss. We also include the method based on contrastive loss, including CSI (Tack et al., 2020) and KNN+ (Sun et al., 2022). For methods using both ID and wild data, we compare with Outlier Exposure (OE) (Hendrycks et al., 2019) and energy-regularization learning (Liu et al., 2020b), which regularize the model by producing lower confidence or higher energy on the auxiliary outlier data. Closest to ours is WOODS (Katz-Samuels et al., 2022), which leverages wild data for OOD learning with a constrained optimization approach. For a fair comparison, all the methods in this group are trained using the same ID and in-the-wild data, under the same mixture ratio $\pi = 0.1$.

The results demonstrate that: **(1)** Methods trained with both ID and wild data perform much better than those trained with only ID data. For example, on PLACES365, SAL reduces the FPR95 by 64.77% compared with KNN+, which highlights the advantage of using in-the-wild data for model regularization. **(2)** SAL performs even better compared to the competitive methods using $\mathbb{P}_{wild}$. On CIFAR-100, SAL achieves an average FPR95 of 1.88%, which is a 5.92% improvement from WOODS. At the same time, SAL maintains a comparable ID accuracy. The slight discrepancy is due to that our method only observes 25,000 labeled ID samples, whereas baseline methods (without using wild data) utilize the entire CIFAR training data with 50,000 samples. **(3)** The strong empirical performance achieved by SAL directly justifies and echoes our theoretical result in Section 4, where we showed the algorithm has a provably small generalization error. *Overall, our algorithm enjoys both theoretical guarantees and empirical effectiveness.*

**Comparison with GradNorm as filtering score.** Huang et al. (2021) proposed directly employing the vector norm of gradients, backpropagated from the KL divergence between the softmax output and a uniform probability distribution for OOD detection. Differently, our SAL derives the filtering score by performing singular value decomposition and using the norm of the projected gradient onto the top singular vector (*c.f.* Section 3.1). We compare SAL with a variant in Table 2, where we replace the filtering score in SAL with the GradNorm score and then train the OOD classifier. The result underperforms SAL, showcasing the effectiveness of our filtering score.

**Additional ablations.** Due to space limitations, we defer additional experiments in the Appendix, including **(1)** analyzing the effect of ratio $\pi$ (Appendix F), **(2)** results on CIFAR-10 (Appendix H),

Table 2: Comparison with using GradNorm as the filtering score. We use CIFAR-100 as ID. All methods are trained on Wide ResNet-40-2 for 100 epochs with $\pi = 0.1$. Bold numbers are superior results.

| Filter score | OOD Datasets | | | | | | | | | | | | ID ACC |
| | SVHN | | PLACES365 | | LSUN-C | | LSUN-RESIZE | | TEXTURES | | Average | | |
| | FPR95 | AUROC | FPR95 | AUROC | FPR95 | AUROC | FPR95 | AUROC | FPR95 | AUROC | FPR95 | AUROC | |
| GradNorm | 1.08 | 99.62 | 62.07 | 84.08 | 0.51 | 99.77 | 5.16 | 98.73 | 50.39 | 83.39 | 23.84 | 93.12 | 73.89 |
| Ours | **0.07** | **99.95** | **3.53** | **99.06** | **0.06** | **99.94** | **0.02** | **99.95** | **5.73** | **98.65** | **1.88** | **99.51** | 73.71 |

**(3)** evaluation on **unseen** OOD datasets (Appendix I), **(4)** near OOD evaluations (Appendix J), and **(5)** the effect of using multiple singular vectors for calculating the filtering score (Appendix K) .

## 6    RELATED WORK

**OOD detection** has attracted a surge of interest in recent years (Fort et al., 2021; Yang et al., 2021b; Fang et al., 2022; Zhu et al., 2022; Ming et al., 2022a;c; Yang et al., 2022; Wang et al., 2022b; Galil et al., 2023; Djurisic et al., 2023; Tao et al., 2023; Zheng et al., 2023; Wang et al., 2022a; 2023b; Narasimhan et al., 2023; Yang et al., 2023; Uppaal et al., 2023; Zhu et al., 2023b;a; Bai et al., 2023; Ming & Li, 2023; Zhang et al., 2023; Gu et al., 2023; Ghosal et al., 2024). One line of work performs OOD detection by devising scoring functions, including confidence-based methods (Bendale & Boult, 2016; Hendrycks & Gimpel, 2017; Liang et al., 2018), energy-based score (Liu et al., 2020b; Wang et al., 2021; Wu et al., 2023), distance-based approaches (Lee et al., 2018b; Tack et al., 2020; Ren et al., 2021; Sehwag et al., 2021; Sun et al., 2022; Du et al., 2022a; Ming et al., 2023; Ren et al., 2023), gradient-based score (Huang et al., 2021), and Bayesian approaches (Gal & Ghahramani, 2016; Lakshminarayanan et al., 2017; Maddox et al., 2019; Malinin & Gales, 2019; Wen et al., 2020; Kristiadi et al., 2020). Another line of work addressed OOD detection by training-time regularization (Bevandić et al., 2018; Malinin & Gales, 2018; Geifman & El-Yaniv, 2019; Hein et al., 2019; Meinke & Hein, 2020; Jeong & Kim, 2020; Liu et al., 2020a; van Amersfoort et al., 2020; Yang et al., 2021a; Wei et al., 2022; Du et al., 2022b; 2023; Wang et al., 2023a). For example, the model is regularized to produce lower confidence (Lee et al., 2018a; Hendrycks et al., 2019) or higher energy (Liu et al., 2020b; Du et al., 2022c; Ming et al., 2022b) on the outlier data. Most regularization methods assume the availability of a *clean* set of auxiliary OOD data. Several works (Zhou et al., 2021; Katz-Samuels et al., 2022; He et al., 2023) relaxed this assumption by leveraging the unlabeled wild data, but did not have an explicit mechanism for filtering the outliers. Compared to positive-unlabeled learning, which learns classifiers from positive and unlabeled data (Letouzey et al., 2000; Hsieh et al., 2015; Du Plessis et al., 2015; Niu et al., 2016; Gong et al., 2018; Chapel et al., 2020; Garg et al., 2021; Xu & Denil, 2021; Garg et al., 2022; Zhao et al., 2022; Acharya et al., 2022), the key difference is that it only considers the task of distinguishing $\mathbb{P}_{out}$ and $\mathbb{P}_{in}$, not the task of doing classification simultaneously. Moreover, we propose a new filtering score to separate outliers from the unlabeled data, which has a bounded error guarantee.

**Robust statistics** has systematically studied the estimation in the presence of outliers since the pioneering work of (Tukey, 1960). Popular methods include RANSAC (Fischler & Bolles, 1981), minimum covariance determinant (Rousseeuw & Driessen, 1999), Huberizing the loss (Owen, 2007), removal based on $k$-nearest neighbors (Breunig et al., 2000). More recently, there are several works that scale up the robust estimation into high-dimensions (Awasthi et al., 2014; Kothari & Steurer, 2017; Steinhardt, 2017; Diakonikolas & Kane, 2019; Diakonikolas et al., 2019a; 2022a;b). Diakonikolas et al. (2019b) designed a gradient-based score for outlier removal but they focused on the error bound for the ID classifier. Instead, we provide new theoretical guarantees on outlier filtering (Theorem 1 and Theorem 2) and the generalization bound of OOD detection (Theorem 3).

## 7    CONCLUSION

In this paper, we propose a novel learning framework SAL that exploits the unlabeled in-the-wild data for OOD detection. SAL first explicitly filters the candidate outliers from the wild data using a new filtering score and then trains a binary OOD classifier leveraging the filtered outliers. Theoretically, SAL answers the question of *how does unlabeled wild data help OOD detection* by analyzing the separability of the outliers in the wild and the learnability of the OOD classifier, which provide provable error guarantees for the two integral components. Empirically, SAL achieves strong performance compared to competitive baselines, echoing our theoretical insights. A broad impact statement is included in Appendix V. We hope our work will inspire future research on OOD detection with unlabeled wild data.

ACKNOWLEDGEMENT

We thank Yifei Ming and Yiyou Sun for their valuable suggestions on the draft. The authors would also like to thank ICLR anonymous reviewers for their helpful feedback. Du is supported by the Jane Street Graduate Research Fellowship. Li gratefully acknowledges the support from the AFOSR Young Investigator Program under award number FA9550-23-1-0184, National Science Foundation (NSF) Award No. IIS-2237037 & IIS-2331669, Office of Naval Research under grant number N00014-23-1-2643, Philanthropic Fund from SFF, and faculty research awards/gifts from Google and Meta.

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

# How Does Unlabeled Data Provably
# Help Out-of-Distribution Detection? (Appendix)

## A    ALGORITHM OF SAL

We summarize our algorithm in implementation as follows.

---
**Algorithm 1** SAL: Separate And Learn
---
**Input:** In-distribution data $\mathcal{S}^{\text{in}} = \{(\mathbf{x}_i, y_i)\}_{i=1}^n$. Unlabeled wild data $\mathcal{S}_{\text{wild}} = \{\tilde{\mathbf{x}}_i\}_{i=1}^m$. $K$-way classification model $\mathbf{h_w}$ and OOD classifier $\mathbf{g_\theta}$. Parameter spaces $\mathcal{W}$ and $\Theta$. Learning rate lr for $\mathbf{g_\theta}$.
**Output:** Learned OOD classifier $\mathbf{g}_{\widehat{\boldsymbol{\theta}}_T}$.
**# Filtering stage**
1) Perform ERM: $\mathbf{w}_{\mathcal{S}^{\text{in}}} \in \text{argmin}_{\mathbf{w} \in \mathcal{W}} \mathrm{R}_{\mathcal{S}^{\text{in}}}(\mathbf{h_w})$.
2) Calculate the reference gradient as $\bar{\nabla} = \frac{1}{n} \sum_{(\mathbf{x}_i, y_i) \in \mathcal{S}^{\text{in}}} \nabla \ell(\mathbf{h}_{\mathbf{w}_{\mathcal{S}^{\text{in}}}}(\mathbf{x}_i), y_i)$.
3) Calculate gradient on $\mathcal{S}_{\text{wild}}$ as $\nabla \ell(\mathbf{h}_{\mathbf{w}_{\mathcal{S}^{\text{in}}}}(\tilde{\mathbf{x}}_i), \widehat{y}_{\tilde{\mathbf{x}}_i})$ and calculate the gradient matrix $\mathbf{G}$.
4) Calculate the top singular vector $\mathbf{v}$ of $\mathbf{G}$ and the score $\tau_i = \left\langle \nabla \ell(\mathbf{h}_{\mathbf{w}_{\mathcal{S}^{\text{in}}}}(\tilde{\mathbf{x}}_i), \widehat{y}_{\tilde{\mathbf{x}}_i}) - \bar{\nabla}, \mathbf{v} \right\rangle^2$.
5) Get the candidate outliers $\mathcal{S}_T = \{\tilde{\mathbf{x}}_i \in \mathcal{S}_{\text{wild}}, \tau_i \geq T\}$.
**# Training Stage**
**for** *epoch in epochs* **do**
    6) Sample batches of data $\mathcal{B}^{\text{in}}, \mathcal{B}_T$ from ID and candidate outliers $\mathcal{S}^{\text{in}}, \mathcal{S}_T$.
    7) Calculate the binary classification loss $R_{\mathcal{B}^{\text{in}}, \mathcal{B}_T}(\mathbf{g_\theta})$.
    8) Update the parameter by $\widehat{\boldsymbol{\theta}}_T = \boldsymbol{\theta} - \text{lr} \cdot \nabla R_{\mathcal{B}^{\text{in}}, \mathcal{B}_T}(\mathbf{g_\theta})$.
**end**

---

## B    NOTATIONS, DEFINITIONS, ASSUMPTIONS AND IMPORTANT CONSTANTS

Here we summarize the important notations and constants in Tables 3 and 4, restate necessary definitions and assumptions in Sections B.2 and B.3.

### B.1    NOTATIONS

Please see Table 3 for detailed notations.

### B.2    DEFINITIONS

**Definition 1** ($\beta$-smooth). *We say a loss function $\ell(\mathbf{h_w}(\mathbf{x}), y)$ (defined over $\mathcal{X} \times \mathcal{Y}$) is $\beta$-smooth, if for any $\mathbf{x} \in \mathcal{X}$ and $y \in \mathcal{Y}$,*

$$\left\| \nabla \ell(\mathbf{h_w}(\mathbf{x}), y) - \nabla \ell(\mathbf{h}_{\mathbf{w}'}(\mathbf{x}), y) \right\|_2 \leq \beta \|\mathbf{w} - \mathbf{w}'\|_2$$

**Definition 2** (Gradient-based Distribution Discrepancy). *Given distributions $\mathbb{P}$ and $\mathbb{Q}$ defined over $\mathcal{X}$, the Gradient-based Distribution Discrepancy w.r.t. predictor $\mathbf{h_w}$ and loss $\ell$ is*

$$d_{\mathbf{w}}^{\ell}(\mathbb{P}, \mathbb{Q}) = \left\| \nabla R_{\mathbb{P}}(\mathbf{h_w}, \widehat{\mathbf{h}}) - \nabla R_{\mathbb{Q}}(\mathbf{h_w}, \widehat{\mathbf{h}}) \right\|_2, \tag{15}$$

*where $\widehat{\mathbf{h}}$ is a classifier which returns the closest one-hot vector of $\mathbf{h_w}$, $R_{\mathbb{P}}(\mathbf{h_w}, \widehat{\mathbf{h}}) = \mathbb{E}_{\mathbf{x} \sim \mathbb{P}} \ell(\mathbf{h_w}, \widehat{\mathbf{h}})$ and $R_{\mathbb{Q}}(\mathbf{h_w}, \widehat{\mathbf{h}}) = \mathbb{E}_{\mathbf{x} \sim \mathbb{Q}} \ell(\mathbf{h_w}, \widehat{\mathbf{h}})$.*

**Definition 3** ($(\gamma, \zeta)$-discrepancy). *We say a wild distribution $\mathbb{P}_{wild}$ has $(\gamma, \zeta)$-discrepancy w.r.t. an ID joint distribution $\mathbb{P}_{in}$, if $\gamma > \min_{\mathbf{w} \in \mathcal{W}} R_{\mathbb{P}_{\mathcal{X}\mathcal{Y}}}(\mathbf{h_w})$, and for any parameter $\mathbf{w} \in \mathcal{W}$ satisfying that $R_{\mathbb{P}_{\mathcal{X}\mathcal{Y}}}(\mathbf{h_w}) \leq \gamma$ should meet the following condition*

$$d_{\mathbf{w}}^{\ell}(\mathbb{P}_{in}, \mathbb{P}_{wild}) > \zeta,$$

*where $R_{\mathbb{P}_{\mathcal{X}\mathcal{Y}}}(\mathbf{h_w}) = \mathbb{E}_{(\mathbf{x}, y) \sim \mathbb{P}_{\mathcal{X}\mathcal{Y}}} \ell(\mathbf{h_w}(\mathbf{x}), y)$.*

Table 3: Main notations and their descriptions.

| Notation | Description |
|---|---|
| Spaces | |
| $\mathcal{X}, \mathcal{Y}$ | the input space and the label space. |
| $\mathcal{W}, \Theta$ | the hypothesis spaces |
| Distributions | |
| $\mathbb{P}_{\text{wild}}, \mathbb{P}_{\text{in}}, \mathbb{P}_{\text{out}}$ | data distribution for wild data, labeled ID data and OOD data |
| $\mathbb{P}_{\mathcal{X}\mathcal{Y}}$ | the joint data distribution for ID data. |
| Data and Models | |
| $\mathbf{w}, \mathbf{x}, \mathbf{v}$ | weight/input/the top-1 right singular vector of $G$ |
| $\widehat{\nabla}, \tau$ | the average gradients on labeled ID data, uncertainty score |
| $y$ and $y_b$ | label for ID classification and binary label for OOD detection |
| $\widehat{y}_{\mathbf{x}}$ | Predicted one-hot label for input $\mathbf{x}$ |
| $\mathbf{h}_{\mathbf{w}}$ and $\mathbf{g}_{\boldsymbol{\theta}}$ | predictor on labeled in-distribution and binary predictor for OOD detection |
| $\mathcal{S}_{\text{wild}}^{\text{in}}, \mathcal{S}_{\text{wild}}^{\text{out}}$ | inliers and outliers in the wild dataset. |
| $\mathcal{S}^{\text{in}}, \mathcal{S}_{\text{wild}}$ | labeled ID data and unlabeled wild data |
| $n, m$ | size of $\mathcal{S}^{\text{in}}$, size of $\mathcal{S}_{\text{wild}}$ |
| $T$ | the filtering threshold |
| $\mathcal{S}_T$ | wild data whose uncertainty score higher than threshold $T$ |
| Distances | |
| $r_1$ and $r_2$ | the radius of the hypothesis spaces $\mathcal{W}$ and $\Theta$, respectively |
| $\| \cdot \|_2$ | $\ell_2$ norm |
| Loss, Risk and Predictor | |
| $\ell(\cdot, \cdot), \ell_b(\cdot, \cdot)$ | ID loss function, binary loss function |
| $R_{\mathcal{S}}(\mathbf{h}_{\mathbf{w}})$ | the empirical risk w.r.t. predictor $\mathbf{h}_{\mathbf{w}}$ over data $\mathcal{S}$ |
| $R_{\mathbb{P}_{\mathcal{X}\mathcal{Y}}}(\mathbf{h}_{\mathbf{w}})$ | the risk w.r.t. predictor $\mathbf{h}_{\mathbf{w}}$ over joint distribution $\mathbb{P}_{\mathcal{X}\mathcal{Y}}$ |
| $R_{\mathbb{P}_{\text{in}}, \mathbb{P}_{\text{out}}}(\mathbf{g}_{\boldsymbol{\theta}})$ | the risk defined in Eq. 14 |
| $\text{ERR}_{\text{in}}, \text{ERR}_{\text{out}}$ | the error rates of regarding ID as OOD and OOD as ID |

In Section F, we empirically calculate the values of the distribution discrepancy between the ID joint distribution $\mathbb{P}_{\mathcal{X}\mathcal{Y}}$ and the wild distribution $\mathbb{P}_{\text{wild}}$.

### B.3 ASSUMPTIONS

**Assumption 1.**

- *The parameter space $\mathcal{W} \subset B(\mathbf{w}_0, r_1) \subset \mathbb{R}^d$ ($\ell_2$ ball of radius $r_1$ around $\mathbf{w}_0$);*

- *The parameter space $\Theta \subset B(\boldsymbol{\theta}_0, r_2) \subset \mathbb{R}^{d'}$ ($\ell_2$ ball of radius $r_2$ around $\boldsymbol{\theta}_0$);*

- *$\ell(\mathbf{h}_{\mathbf{w}}(\mathbf{x}), y) \geq 0$ and $\ell(\mathbf{h}_{\mathbf{w}}(\mathbf{x}), y)$ is $\beta_1$-smooth;*

- *$\ell_b(\mathbf{g}_{\boldsymbol{\theta}}(\mathbf{x}), y_b) \geq 0$ and $\ell_b(\mathbf{g}_{\boldsymbol{\theta}}(\mathbf{x}), y_b)$ is $\beta_2$-smooth;*

- *$\sup_{(\mathbf{x},y) \in \mathcal{X} \times \mathcal{Y}} \|\nabla \ell(\mathbf{h}_{\mathbf{w}_0}(\mathbf{x}), y)\|_2 = b_1$, $\sup_{(\mathbf{x}, y_b) \in \mathcal{X} \times \mathcal{Y}_b} \|\nabla \ell(\mathbf{g}_{\boldsymbol{\theta}_0}(\mathbf{x}), y_b)\|_2 = b_2$;*

- *$\sup_{(\mathbf{x},y) \in \mathcal{X} \times \mathcal{Y}} \ell(\mathbf{h}_{\mathbf{w}_0}(\mathbf{x}), y) = B_1$, $\sup_{(\mathbf{x}, y_b) \in \mathcal{X} \times \mathcal{Y}_b} \ell(\mathbf{g}_{\boldsymbol{\theta}_0}(\mathbf{x}), y_b) = B_2$.*

**Remark 2.** *For neural networks with smooth activation functions and softmax output function, we can check that the norm of the second derivative of the loss functions (cross-entropy loss and sigmoid loss) is bounded given the bounded parameter space, which implies that the $\beta$-smoothness of the loss functions can hold true. Therefore, our assumptions are reasonable in practice.*

**Assumption 2.** $\ell(\mathbf{h}(\mathbf{x}), \widehat{y}_{\mathbf{x}}) \leq \min_{y \in \mathcal{Y}} \ell(\mathbf{h}(\mathbf{x}), y)$, *where $\widehat{y}_{\mathbf{x}}$ returns the closest one-hot label of the predictor $\mathbf{h}$'s output on $\mathbf{x}$.*

**Remark 3.** *The assumption means the loss incurred by using the predicted labels given by the classifier itself is smaller or equal to the loss incurred by using any label in the label space. If $y = \widehat{y}_{\mathbf{x}}$, the assumption is satisfied obviously. If $y \neq \widehat{y}_{\mathbf{x}}$, then we provide two examples to illustrate the validity of the assumption. For example, (1) if the loss $\ell$ is the cross entropy loss, let $K = 2$, $\mathbf{h}(\mathbf{x}) = [h_1, h_2]$ (classification output after softmax) and $h_1 > h_2$. Therefore, we have $\widehat{y}_{\mathbf{x}} = 0$. Suppose $y = 1$, we can get $\ell(\mathbf{h}(\mathbf{x}), \widehat{y}_{\mathbf{x}}) = -\log(h_1) < \ell(\mathbf{h}(\mathbf{x}), y) = -\log(h_2)$. (2) If $\ell$ is the hinge loss for binary classification, thus we have $K = 1$, let $\mathbf{h}(\mathbf{x}) = h_1 < 0$ and thus $\widehat{y}_{\mathbf{x}} = -1$. Suppose $y = 1$, we can get $\ell(\mathbf{h}(\mathbf{x}), \widehat{y}_{\mathbf{x}}) = \max(0, 1 + h_1) < \max(0, 1 - h_1) = \ell(\mathbf{h}(\mathbf{x}), y)$.*

### B.4 Constants in Theory

Table 4: Constants in theory.

| Constants | Description |
|---|---|
| $M = \beta_1 r_1^2 + b_1 r_1 + B_1$ | the upper bound of loss $\ell(\mathbf{h}_{\mathbf{w}}(\mathbf{x}), y)$, see Proposition 1 |
| $M' = 2(\beta_1 r_1 + b_1)^2$ | the upper bound of filtering score $\tau$ |
| $\tilde{M} = \beta_1 M$ | a constant for simplified representation |
| $L = \beta_2 r_2^2 + b_2 r_2 + B_2$ | the upper bound of loss $\ell_{\mathsf{b}}(\mathbf{g}_{\boldsymbol{\theta}}(\mathbf{x}), y_{\mathsf{b}})$, see Proposition 1 |
| $d$, $d'$ | the dimensions of parameter spaces $\mathcal{W}$ and $\Theta$, respectively |
| $R_{in}^*$ | the optimal ID risk, i.e., $R_{in}^* = \min_{\mathbf{w} \in \mathcal{W}} \mathbb{E}_{(\mathbf{x},y) \sim \mathbb{P}_{\mathcal{X}\mathcal{Y}}} \ell(\mathbf{h}_{\mathbf{w}}(\mathbf{x}), y)$ |
| $\delta(T)$ | the main error in Eq. 10 |
| $\zeta$ | the discrepancy between $\mathbb{P}_{\text{in}}$ and $\mathbb{P}_{\text{wild}}$ |
| $\pi$ | the ratio of OOD distribution in $\mathbb{P}_{\text{wild}}$ |

## C Main Theorems

In this section, we provide a detailed and formal version of our main theorems with a complete description of the constant terms and other additional details that are omitted in the main paper.

**Theorem 1.** *If Assumptions 1 and 2 hold, $\mathbb{P}_{wild}$ has $(\gamma, \zeta)$-discrepancy w.r.t. $\mathbb{P}_{\mathcal{X}\mathcal{Y}}$, and there exists $\eta \in (0, 1)$ s.t. $\Delta = (1 - \eta)^2 \zeta^2 - 8\beta_1 R_{in}^* > 0$, then for*

$$n = \Omega\Big(\frac{\tilde{M} + M(r_1 + 1)d}{\eta^2 \Delta} + \frac{M^2 d}{(\gamma - R_{in}^*)^2}\Big), \quad m = \Omega\Big(\frac{\tilde{M} + M(r_1 + 1)d}{\eta^2 \zeta^2}\Big),$$

*with the probability at least $9/10$, for any $0 < T < M'$ (here $M' = 2(\beta_1 r_1 + b_1)^2$ is the upper bound of filtering score $\tau_i$, i.e., $\tau_i \leq M'$),*

$$\mathrm{ERR}_{in} \leq \frac{8\beta_1}{T} R_{in}^* + O\Big(\frac{\tilde{M}}{T}\sqrt{\frac{d}{n}}\Big) + O\Big(\frac{\tilde{M}}{T}\sqrt{\frac{d}{(1 - \pi)m}}\Big), \tag{16}$$

$$\mathrm{ERR}_{out} \leq \delta(T) + O\Big(\frac{\tilde{M}}{1 - T/M'}\sqrt{\frac{d}{\pi^2 n}}\Big) + O\Big(\frac{\max\{\tilde{M}\sqrt{d}, \Delta_\zeta^\eta/\pi\}}{1 - T/M'}\sqrt{\frac{1}{\pi^2(1 - \pi)m}}\Big), \tag{17}$$

*where $R_{in}^*$ is the optimal ID risk, i.e., $R_{in}^* = \min_{\mathbf{w} \in \mathcal{W}} \mathbb{E}_{(\mathbf{x},y) \sim \mathbb{P}_{\mathcal{X}\mathcal{Y}}} \ell(\mathbf{h}_{\mathbf{w}}(\mathbf{x}), y)$,*

$$\delta(T) = \frac{\max\{0, 1 - \Delta_\zeta^\eta/\pi\}}{(1 - T/M')}, \quad \Delta_\zeta^\eta = 0.98\eta^2\zeta^2 - 8\beta_1 R_{in}^*,$$
$$M = \beta_1 r_1^2 + b_1 r_1 + B_1, \quad \tilde{M} = M\beta_1, \tag{18}$$

*and $d$ is the dimension of the parameter space $\mathcal{W}$, here $\beta_1, r_1, B_1$ are given in Assumption 1.*

**Theorem 2.** *1) If $\Delta_\zeta^\eta \geq (1-\epsilon)\pi$ for a small error $\epsilon \geq 0$, then the main error $\delta(T)$ defined in Eq.* *11 satisfies that*

$$\delta(T) \leq \frac{\epsilon}{1 - T/M'}.$$

*2) If $\zeta \geq 2.011\sqrt{8\beta_1 R_{in}^*} + 1.011\sqrt{\pi}$, then there exists $\eta \in (0,1)$ ensuring that $\Delta > 0$ and $\Delta_\zeta^\eta > \pi$ hold, which implies that the main error $\delta(T) = 0$.*

**Theorem 3.** *Given the same conditions in Theorem 1, if we further require that*

$$n = \Omega\Big(\frac{\tilde{M}^2 d}{\min\{\pi, \Delta_\zeta^\eta\}^2}\Big), \quad m = \Omega\Big(\frac{\tilde{M}^2 d + \Delta_\zeta^\eta}{\pi^2(1-\pi)\min\{\pi, \Delta_\zeta^\eta\}^2}\Big),$$

*then with the probability at least $89/100$, for any $0 < T < 0.9M'\min\{1, \Delta_\zeta^\eta/\pi\}$, the OOD classifier $\mathbf{g}_{\widehat{\boldsymbol{\theta}}_T}$ learned by the proposed algorithm satisfies the following risk estimation*

$$R_{\mathbb{P}_{in}, \mathbb{P}_{out}}(\mathbf{g}_{\widehat{\boldsymbol{\theta}}_T}) \leq \inf_{\boldsymbol{\theta} \in \Theta} R_{\mathbb{P}_{in}, \mathbb{P}_{out}}(\mathbf{g}_{\boldsymbol{\theta}}) + \frac{3.5L}{1 - \delta(T)}\delta(T) + \frac{9(1-\pi)L\beta_1}{\pi(1 - \delta(T))T}R_{in}^*$$

$$+ O\Big(\frac{L\max\{\tilde{M}\sqrt{d}, \sqrt{d'}\}}{\min\{\pi, \Delta_\zeta^\eta\}T'}\sqrt{\frac{1}{n}}\Big) + O\Big(\frac{L\max\{\tilde{M}\sqrt{d}, \sqrt{d'}, \Delta_\zeta^\eta\}}{\min\{\pi, \Delta_\zeta^\eta\}T'}\sqrt{\frac{1}{\pi^2(1-\pi)m}}\Big), \tag{19}$$

*where $R_{in}^*$, $\Delta_\zeta^\eta$, $M$, $M'$, $\tilde{M}$ and $d$ are shown in Theorem 1, $d'$ is the dimension of space $\Theta$,*

$$L = \beta_2 r_2^2 + b_2 r_2 + B_2, \quad T' = T/(1+T),$$

*and the risk $R_{\mathbb{P}_{in}, \mathbb{P}_{out}}(\mathbf{g}_{\boldsymbol{\theta}})$ is defined as follows:*

$$R_{\mathbb{P}_{in}, \mathbb{P}_{out}}(\mathbf{g}_{\widehat{\boldsymbol{\theta}}_T}) = \mathbb{E}_{\mathbf{x} \sim \mathbb{P}_{in}}\ell_b(\mathbf{g}_{\boldsymbol{\theta}}(\mathbf{x}), y_+) + \mathbb{E}_{\mathbf{x} \sim \mathbb{P}_{out}}\ell_b(\mathbf{g}_{\boldsymbol{\theta}}(\mathbf{x}), y_-).$$

**Theorem 4.** *Given the same conditions in Theorem 1, with the probability at least $9/10$,*

$$\mathbb{E}_{\tilde{\mathbf{x}}_i \sim \mathcal{S}_{wild}^{in}}\tau_i \leq 8\beta_1 R_{in}^* + O(\beta_1 M\sqrt{\frac{d}{n}}) + O(\beta_1 M\sqrt{\frac{d}{(1-\pi)m}}),$$

$$\mathbb{E}_{\tilde{\mathbf{x}}_i \sim \mathcal{S}_{wild}^{out}}\tau_i \geq \frac{0.98\eta^2\zeta^2}{\pi} - \frac{8\beta_1 R_{in}^*}{\pi} - \epsilon'(n,m),$$

*furthermore, if the realizability assumption for ID distribution holds (Shalev-Shwartz & Ben-David, 2014; Fang et al., 2022), then*

$$\mathbb{E}_{\tilde{\mathbf{x}}_i \sim \mathcal{S}_{wild}^{in}}\tau_i \leq O(\beta_1 M\sqrt{\frac{d}{n}}) + O(\beta_1 M\sqrt{\frac{d}{(1-\pi)m}})$$

$$\mathbb{E}_{\tilde{\mathbf{x}}_i \sim \mathcal{S}_{wild}^{out}}\tau_i \geq \frac{0.98\eta^2\zeta^2}{\pi} - \epsilon'(n,m),$$

*where*

$$\epsilon'(n,m) \leq O(\frac{\beta_1 M}{\pi}\sqrt{\frac{d}{n}}) + O\Big((\beta_1 M\sqrt{d} + \sqrt{1-\pi}\Delta_\zeta^\eta/\pi)\sqrt{\frac{1}{\pi^2(1-\pi)m}}\Big),$$

*and $R_{in}^*$, $\Delta_\zeta^\eta$, $M$ and $d$ are shown in Theorem 1.*

# D  PROOFS OF MAIN THEOREMS

## D.1  PROOF OF THEOREM 1

**Step 1.** With the probability at least $1 - \frac{7}{3}\delta > 0$,

$$
\begin{aligned}
\mathbb{E}_{\tilde{\mathbf{x}}_i \sim \mathcal{S}_{\text{wild}}^{\text{in}}} \tau_i \leq & \, 8\beta_1 R_{\text{in}}^* \\
& + 4\beta_1 \Big[ C\sqrt{\frac{Mr_1(\beta_1 r_1 + b_1)d}{n}} + C\sqrt{\frac{Mr_1(\beta_1 r_1 + b_1)d}{(1-\pi)m - \sqrt{m\log(6/\delta)/2}}} \\
& + 3M\sqrt{\frac{2\log(6/\delta)}{n}} + M\sqrt{\frac{2\log(6/\delta)}{(1-\pi)m - \sqrt{m\log(6/\delta)/2}}} \Big],
\end{aligned}
$$

This can be proven by Lemma 7 and following inequality

$$
\mathbb{E}_{\tilde{\mathbf{x}}_i \sim \mathcal{S}_{\text{wild}}^{\text{in}}} \tau_i \leq \mathbb{E}_{\tilde{\mathbf{x}}_i \sim \mathcal{S}_{\text{wild}}^{\text{in}}} \big\| \nabla\ell(\mathbf{h}_{\mathbf{w}_{\mathcal{S}^{\text{in}}}}(\tilde{\mathbf{x}}_i), \widehat{\mathbf{h}}_{\mathbf{w}_{\mathcal{S}^{\text{in}}}}(\tilde{\mathbf{x}}_i)) - \mathbb{E}_{(\mathbf{x}_j, y_j) \sim \mathcal{S}^{\text{in}}} \nabla\ell(\mathbf{h}_{\mathbf{w}_{\mathcal{S}^{\text{in}}}}(\mathbf{x}_j), y_j) \big\|_2^2,
$$

**Step 2.** It is easy to check that

$$
\mathbb{E}_{\tilde{\mathbf{x}}_i \sim \mathcal{S}_{\text{wild}}} \tau_i = \frac{|\mathcal{S}_{\text{wild}}^{\text{in}}|}{|\mathcal{S}_{\text{wild}}|} \mathbb{E}_{\tilde{\mathbf{x}}_i \sim \mathcal{S}_{\text{wild}}^{\text{in}}} \tau_i + \frac{|\mathcal{S}_{\text{wild}}^{\text{out}}|}{|\mathcal{S}_{\text{wild}}|} \mathbb{E}_{\tilde{\mathbf{x}}_i \sim \mathcal{S}_{\text{wild}}^{\text{out}}} \tau_i.
$$

**Step 3.** Let

$$
\begin{aligned}
\epsilon(n, m) = & \, 4\beta_1 \Big[ C\sqrt{\frac{Mr_1(\beta_1 r_1 + b_1)d}{n}} + C\sqrt{\frac{Mr_1(\beta_1 r_1 + b_1)d}{(1-\pi)m - \sqrt{m\log(6/\delta)/2}}} \\
& + 3M\sqrt{\frac{2\log(6/\delta)}{n}} + M\sqrt{\frac{2\log(6/\delta)}{(1-\pi)m - \sqrt{m\log(6/\delta)/2}}} \Big].
\end{aligned}
$$

Under the condition in Theorem 5, with the probability at least $\frac{97}{100} - \frac{7}{3}\delta > 0$,

$$
\begin{aligned}
\mathbb{E}_{\tilde{\mathbf{x}}_i \sim \mathcal{S}_{\text{wild}}^{\text{out}}} \tau_i \geq & \, \frac{m}{|\mathcal{S}_{\text{wild}}^{\text{out}}|} \Big[ \frac{98\eta^2\zeta^2}{100} - \frac{|\mathcal{S}_{\text{wild}}^{\text{in}}|}{m} 8\beta_1 R_{\text{in}}^* - \frac{|\mathcal{S}_{\text{wild}}^{\text{in}}|}{m}\epsilon(n, m) \Big] \\
\geq & \, \frac{m}{|\mathcal{S}_{\text{wild}}^{\text{out}}|} \Big[ \frac{98\eta^2\zeta^2}{100} - 8\beta_1 R_{\text{in}}^* - \epsilon(n, m) \Big] \\
\geq & \, \Big[ \frac{1}{\pi} - \frac{\sqrt{\log 6/\delta}}{\pi^2\sqrt{2m} + \pi\sqrt{\log(6/\delta)}} \Big] \Big[ \frac{98\eta^2\zeta^2}{100} - 8\beta_1 R_{\text{in}}^* - \epsilon(n, m) \Big].
\end{aligned}
$$

In this proof, we set

$$
\Delta(n, m) = \Big[ \frac{1}{\pi} - \frac{\sqrt{\log 6/\delta}}{\pi^2\sqrt{2m} + \pi\sqrt{\log(6/\delta)}} \Big] \Big[ \frac{98\eta^2\zeta^2}{100} - 8\beta_1 R_{\text{in}}^* - \epsilon(n, m) \Big].
$$

Note that $\Delta_\zeta^\eta = 0.98\eta^2\zeta^2 - 8\beta_1 R_{\text{in}}^*$, then

$$
\Delta(n, m) = \frac{1}{\pi}\Delta_\zeta^\eta - \frac{1}{\pi}\epsilon(n, m) - \Delta_\zeta^\eta \epsilon(m) + \epsilon(n)\epsilon(n, m),
$$

where $\epsilon(m) = \sqrt{\log 6/\delta}/(\pi^2\sqrt{2m} + \pi\sqrt{\log(6/\delta)})$.

**Step 4.** Under the condition in Theorem 5, with the probability at least $\frac{97}{100} - \frac{7}{3}\delta > 0$,

$$
\frac{|\{\tilde{\mathbf{x}}_i \in \mathcal{S}_{\text{wild}}^{\text{out}} : \tau_i \leq T\}|}{|\mathcal{S}_{\text{wild}}^{\text{out}}|} \leq \frac{1 - \min\{1, \Delta(n, m)\}}{1 - T/M'}, \tag{20}
$$

and

$$\frac{|\{\tilde{\mathbf{x}}_i \in \mathcal{S}_{\text{wild}}^{\text{in}} : \tau_i > T\}|}{|\mathcal{S}_{\text{wild}}^{\text{in}}|} \leq \frac{8\beta_1 R_{\text{in}}^* + \epsilon(n, m)}{T}. \tag{21}$$

We prove this step: let $Z$ be the **uniform** random variable with $\mathcal{S}_{\text{wild}}^{\text{out}}$ as its support and $Z(i) = \tau_i/(2(\beta_1 r_1 + b_1)^2)$, then by the Markov inequality, we have

$$\frac{|\{\tilde{\mathbf{x}}_i \in \mathcal{S}_{\text{wild}}^{\text{out}} : \tau_i > T\}|}{|\mathcal{S}_{\text{wild}}^{\text{out}}|} = P(Z(i) > T/(2(\beta_1 r_1 + b_1)^2)) \geq \frac{\Delta(n, m) - T/(2(\beta_1 r_1 + b_1)^2)}{1 - T/(2(\beta_1 r_1 + b_1)^2)}. \tag{22}$$

Let $Z$ be the **uniform** random variable with $\mathcal{S}_{\text{wild}}^{\text{in}}$ as its support and $Z(i) = \tau_i$, then by the Markov inequality, we have

$$\frac{|\{\tilde{\mathbf{x}}_i \in \mathcal{S}_{\text{wild}}^{\text{in}} : \tau_i > T\}|}{|\mathcal{S}_{\text{wild}}^{\text{in}}|} = P(Z(i) > T) \leq \frac{\mathbb{E}[Z]}{T} = \frac{8\beta_1 R_{\text{in}}^* + \epsilon(n, m)}{T}. \tag{23}$$

**Step 5.** If $\pi \leq \Delta_\zeta^\eta/(1 - \epsilon/M')$, then with the probability at least $\frac{97}{100} - \frac{7}{3}\delta > 0$,

$$\frac{|\{\tilde{\mathbf{x}}_i \in \mathcal{S}_{\text{wild}}^{\text{out}} : \tau_i \leq T\}|}{|\mathcal{S}_{\text{wild}}^{\text{out}}|} \leq \frac{\epsilon + M'\epsilon'(n, m)}{M' - T}, \tag{24}$$

and

$$\frac{|\{\tilde{\mathbf{x}}_i \in \mathcal{S}_{\text{wild}}^{\text{in}} : \tau_i > T\}|}{|\mathcal{S}_{\text{wild}}^{\text{in}}|} \leq \frac{8\beta_1 R_{\text{in}}^* + \epsilon(n, m)}{T}, \tag{25}$$

where $\epsilon'(n, m) = \epsilon(n, m)/\pi + \Delta_\zeta^\eta \epsilon(m) - \epsilon(n)\epsilon(n, m)$.

**Step 6.** If we set $\delta = 3/100$, then it is easy to see that

$$\epsilon(m) \leq O(\frac{1}{\pi^2 \sqrt{m}}),$$

$$\epsilon(n, m) \leq O(\beta_1 M \sqrt{\frac{d}{n}}) + O(\beta_1 M \sqrt{\frac{d}{(1 - \pi)m}}),$$

$$\epsilon'(n, m) \leq O(\frac{\beta_1 M}{\pi}\sqrt{\frac{d}{n}}) + O\Big((\beta_1 M \sqrt{d} + \sqrt{1 - \pi}\Delta_\zeta^\eta/\pi)\sqrt{\frac{1}{\pi^2(1 - \pi)m}}\Big).$$

**Step 7.** By results in Steps 4, 5 and 6, We complete this proof.

## D.2 Proof of Theorem 2

The first result is trivial. Hence, we omit it. We mainly focus on the second result in this theorem. In this proof, then we set

$$\eta = \sqrt{8\beta_1 R_{\text{in}}^* + 0.99\pi}/(\sqrt{0.98}\sqrt{8\beta_1 R_{\text{in}}^*} + \sqrt{8\beta_1 R_{\text{in}}^* + \pi})$$

Note that it is easy to check that

$$\zeta \geq 2.011\sqrt{8\beta_1 R_{\text{in}}^*} + 1.011\sqrt{\pi} \geq \sqrt{8\beta_1 R_{\text{in}}^*} + 1.011\sqrt{8\beta_1 R_{\text{in}}^* + \pi}.$$

Therefore,

$$\eta\zeta \geq \frac{1}{\sqrt{0.98}}\sqrt{8\beta_1 R_{\text{in}}^* + 0.99\pi} > \sqrt{8\beta_1 R_{\text{in}}^* + \pi},$$

which implies that $\Delta_\zeta^\eta > \pi$. Note that

$$(1-\eta)\zeta \geq \frac{1}{\sqrt{0.98}}\left(\sqrt{0.98}\sqrt{8\beta_1 R_{\text{in}}^*} + \sqrt{8\beta_1 R_{\text{in}}^* + \pi} - \sqrt{8\beta_1 R_{\text{in}}^* + 0.99\pi}\right) > \sqrt{8\beta_1 R_{\text{in}}^*},$$

which implies that $\Delta > 0$. We have completed this proof.

## D.3 Proof of Theorem 3

Let

$$\boldsymbol{\theta}^* \in \arg\min_{\boldsymbol{\theta}\in\Theta} R_{\mathbb{P}_{\text{in}},\mathbb{P}_{\text{out}}}(\boldsymbol{\theta}).$$

Then by Lemma 1 and Lemma 14, we obtain that with the high probability

$$
\begin{aligned}
&R_{\mathbb{P}_{\text{in}},\mathbb{P}_{\text{out}}}(\mathbf{g}_{\widehat{\boldsymbol{\theta}}_T}) - R_{\mathbb{P}_{\text{in}},\mathbb{P}_{\text{out}}}(\mathbf{g}_{\boldsymbol{\theta}^*}) \\
=&R_{\mathbb{P}_{\text{in}},\mathbb{P}_{\text{out}}}(\mathbf{g}_{\widehat{\boldsymbol{\theta}}_T}) - R_{\mathcal{S}^{\text{in}},\mathcal{S}_T}(\mathbf{g}_{\widehat{\boldsymbol{\theta}}_T}) + R_{\mathcal{S}^{\text{in}},\mathcal{S}_T}(\mathbf{g}_{\widehat{\boldsymbol{\theta}}_T}) - R_{\mathcal{S}^{\text{in}},\mathcal{S}_T}(\mathbf{g}_{\boldsymbol{\theta}^*}) \\
&+R_{\mathcal{S}^{\text{in}},\mathcal{S}_T}(\mathbf{g}_{\boldsymbol{\theta}^*}) - R_{\mathbb{P}_{\text{in}},\mathbb{P}_{\text{out}}}(\mathbf{g}_{\boldsymbol{\theta}^*}) \\
\leq&R_{\mathbb{P}_{\text{in}},\mathbb{P}_{\text{out}}}(\mathbf{g}_{\widehat{\boldsymbol{\theta}}_T}) - R_{\mathcal{S}^{\text{in}},\mathcal{S}_T}(\mathbf{g}_{\widehat{\boldsymbol{\theta}}_T}) \\
&+R_{\mathcal{S}^{\text{in}},\mathcal{S}_T}(\mathbf{g}_{\boldsymbol{\theta}^*}) - R_{\mathbb{P}_{\text{in}},\mathbb{P}_{\text{out}}}(\mathbf{g}_{\boldsymbol{\theta}^*}) \\
\leq&2\sup_{\boldsymbol{\theta}\in\Theta}\left|R_{\mathcal{S}^{\text{in}}}^+(\mathbf{g}_{\boldsymbol{\theta}}) - R_{\mathbb{P}_{\text{in}}}^+(\mathbf{g}_{\boldsymbol{\theta}})\right| \\
&+\sup_{\boldsymbol{\theta}\in\Theta}\left(R_{\mathcal{S}^{\text{out}}}^-(\mathbf{g}_{\boldsymbol{\theta}}) - R_{\mathbb{P}_{\text{out}}}^-(\mathbf{g}_{\boldsymbol{\theta}})\right) + \sup_{\boldsymbol{\theta}\in\Theta}\left(R_{\mathbb{P}_{\text{out}}}^-(\mathbf{g}_{\boldsymbol{\theta}}) - R_{\mathcal{S}^{\text{out}}}^-(\mathbf{g}_{\boldsymbol{\theta}})\right) \\
\leq&\frac{3.5L}{1-\delta(T)}\delta(T) + \frac{9(1-\pi)L\beta_1}{\pi(1-\delta(T))T}R_{\text{in}}^* \\
&+O\left(\frac{L\max\{\beta_1 M\sqrt{d}, \sqrt{d'}\}(1+T)}{\min\{\pi, \Delta_\zeta^\eta\}T}\sqrt{\frac{1}{n}}\right) \\
&+O\left(\frac{L\max\{\beta_1 M\sqrt{d}, \sqrt{d'}, \Delta_\zeta^\eta\}(1+T)}{\min\{\pi, \Delta_\zeta^\eta\}T}\sqrt{\frac{1}{\pi^2(1-\pi)m}}\right),
\end{aligned}
$$

## D.4 Proof of Theorem 4

The result is induced by the Steps **1**, **3** and **6** in Proof of Theorem 1 (see section D.1).

# E   NECESSARY LEMMAS, PROPOSITIONS AND THEOREMS

## E.1   BOUNDEDNESS

**Proposition 1.** *If Assumption 1 holds,*

$$\sup_{\mathbf{w}\in\mathcal{W}}\sup_{(\mathbf{x},y)\in\mathcal{X}\times\mathcal{Y}}\|\nabla\ell(\mathbf{h}_{\mathbf{w}}(\mathbf{x}),y)\|_2 \leq \beta_1 r_1 + b_1 = \sqrt{M'/2},$$

$$\sup_{\boldsymbol{\theta}\in\Theta}\sup_{(\mathbf{x},y_b)\in\mathcal{X}\times\mathcal{Y}_b}\|\nabla\ell(\mathbf{g}_{\boldsymbol{\theta}}(\mathbf{x}),y_b)\|_2 \leq \beta_2 r_2 + b_2.$$

$$\sup_{\mathbf{w}\in\mathcal{W}}\sup_{(\mathbf{x},y)\in\mathcal{X}\times\mathcal{Y}}\ell(\mathbf{h}_{\mathbf{w}}(\mathbf{x}),y) \leq \beta_1 r_1^2 + b_1 r_1 + B_1 = M,$$

$$\sup_{\boldsymbol{\theta}\in\Theta}\sup_{(\mathbf{x},y_b)\in\mathcal{X}\times\mathcal{Y}_b}\ell_b(\mathbf{g}_{\boldsymbol{\theta}}(\mathbf{x}),y_b) \leq \beta_2 r_2^2 + b_2 r_2 + B_2 = L.$$

*Proof.* One can prove this by *Mean Value Theorem of Integrals* easily.   □

**Proposition 2.** *If Assumption 1 holds, for any* $\mathbf{w}\in\mathcal{W}$,

$$\left\|\nabla\ell(\mathbf{h}_{\mathbf{w}}(\mathbf{x}),y)\right\|_2^2 \leq 2\beta_1\ell(\mathbf{h}_{\mathbf{w}}(\mathbf{x}),y).$$

*Proof.* The details of the self-bounding property can be found in Appendix B of Lei & Ying (2021).
   □

**Proposition 3.** *If Assumption 1 holds, for any labeled data* $\mathcal{S}$ *and distribution* $\mathbb{P}$,

$$\left\|\nabla R_{\mathcal{S}}(\mathbf{h}_{\mathbf{w}})\right\|_2^2 \leq 2\beta_1 R_{\mathcal{S}}(\mathbf{h}_{\mathbf{w}}), \quad \forall \mathbf{w}\in\mathcal{W},$$

$$\left\|\nabla R_{\mathbb{P}}(\mathbf{h}_{\mathbf{w}})\right\|_2^2 \leq 2\beta_1 R_{\mathbb{P}}(\mathbf{h}_{\mathbf{w}}), \quad \forall \mathbf{w}\in\mathcal{W}.$$

*Proof.* Jensen's inequality implies that $R_{\mathcal{S}}(\mathbf{h}_{\mathbf{w}})$ and $R_{\mathbb{P}}(\mathbf{h}_{\mathbf{w}})$ are $\beta_1$-smooth. Then Proposition 2 implies the results.   □

### E.2 CONVERGENCE

**Lemma 1** (Uniform Convergence-I). *If Assumption 1 holds, then for any distribution $\mathbb{P}$, with the probability at least $1 - \delta > 0$, for any $\mathbf{w} \in \mathcal{W}$,*

$$|R_{\mathcal{S}}(\mathbf{h}_{\mathbf{w}}) - R_{\mathbb{P}}(\mathbf{h}_{\mathbf{w}})| \leq M\sqrt{\frac{2\log(2/\delta)}{n}} + C\sqrt{\frac{Mr_1(\beta_1 r_1 + b_1)d}{n}},$$

*where $n = |\mathcal{S}|$, $M = \beta_1 r_1^2 + b_1 r_1 + B_1$, $d$ is the dimension of $\mathcal{W}$, and $C$ is a uniform constant.*

*Proof of Lemma 1.* Let

$$X_{\mathbf{h}_{\mathbf{w}}} = \mathbb{E}_{(\mathbf{x},y)\sim\mathbb{P}}\ell(\mathbf{h}_{\mathbf{w}}(\mathbf{x}), y) - \mathbb{E}_{(\mathbf{x},y)\sim\mathcal{S}}\ell(\mathbf{h}_{\mathbf{w}}(\mathbf{x}), y).$$

Then it is clear that

$$\mathbb{E}_{S\sim\mathbb{P}^n} X_{\mathbf{h}_{\mathbf{w}}} = 0.$$

By Proposition 2.6.1 and Lemma 2.6.8 in Vershynin (2018),

$$\|X_{\mathbf{h}_{\mathbf{w}}} - X_{\mathbf{h}_{\mathbf{w}'}}\|_{\Phi_2} \leq \frac{c_0}{\sqrt{n}}\|\ell(\mathbf{h}_{\mathbf{w}}(\mathbf{x}), y) - \ell(\mathbf{h}_{\mathbf{w}'}(\mathbf{x}), y)\|_{L^\infty(\mathcal{X}\times\mathcal{Y})},$$

where $\|\cdot\|_{\Phi_2}$ is the sub-gaussian norm and $c_0$ is a uniform constant. Therefore, by Dudley's entropy integral (Vershynin, 2018), we have

$$\mathbb{E}_{S\sim\mathbb{P}^n} \sup_{\mathbf{w}\in\mathcal{W}} X_{\mathbf{h}_{\mathbf{w}}} \leq \frac{b_0}{\sqrt{n}} \int_0^{+\infty} \sqrt{\log\mathcal{N}(\mathcal{F}, \epsilon, L^\infty)}\mathrm{d}\epsilon,$$

where $b_0$ is a uniform constant, $\mathcal{F} = \{\ell(\mathbf{h}_{\mathbf{w}}; \mathbf{x}, y) : \mathbf{w} \in \mathcal{W}\}$, and $\mathcal{N}(\mathcal{F}, \epsilon, L^\infty)$ is the covering number under the $L^\infty$ norm. Note that

$$\begin{aligned}
\mathbb{E}_{S\sim\mathbb{P}^n} \sup_{\mathbf{w}\in\mathcal{W}} X_{\mathbf{h}_{\mathbf{w}}} &\leq \frac{b_0}{\sqrt{n}} \int_0^{+\infty} \sqrt{\log\mathcal{N}(\mathcal{F}, \epsilon, L^\infty)}\mathrm{d}\epsilon \\
&= \frac{b_0}{\sqrt{n}} \int_0^{M} \sqrt{\log\mathcal{N}(\mathcal{F}, \epsilon, L^\infty)}\mathrm{d}\epsilon \\
&= \frac{b_0}{\sqrt{n}} M \int_0^{1} \sqrt{\log\mathcal{N}(\mathcal{F}, M\epsilon, L^\infty)}\mathrm{d}\epsilon.
\end{aligned}$$

Then, we use the McDiarmid's Inequality, then with the probability at least $1 - e^{-t} > 0$, for any $\mathbf{w} \in \mathcal{W}$,

$$X_{\mathbf{h}_{\mathbf{w}}} \leq \frac{b_0}{\sqrt{n}} M \int_0^{1} \sqrt{\log\mathcal{N}(\mathcal{F}, M\epsilon, L^\infty)}\mathrm{d}\epsilon + M\sqrt{\frac{2t}{n}}.$$

Similarly, we can also prove that with the probability at least $1 - e^{-t} > 0$, for any $\mathbf{w} \in \mathcal{W}$,

$$-X_{\mathbf{h}_{\mathbf{w}}} \leq \frac{b_0}{\sqrt{n}} M \int_0^{1} \sqrt{\log\mathcal{N}(\mathcal{F}, M\epsilon, L^\infty)}\mathrm{d}\epsilon + M\sqrt{\frac{2t}{n}}.$$

Therefore, with the probability at least $1 - 2e^{-t} > 0$, for any $\mathbf{w} \in \mathcal{W}$,

$$|X_{\mathbf{h}_{\mathbf{w}}}| \leq \frac{b_0}{\sqrt{n}} M \int_0^{1} \sqrt{\log\mathcal{N}(\mathcal{F}, M\epsilon, L^\infty)}\mathrm{d}\epsilon + M\sqrt{\frac{2t}{n}}.$$

Note that $\ell(\mathbf{h}_{\mathbf{w}}(\mathbf{x}), y)$ is $(\beta_1 r_1 + b_1)$-Lipschitz w.r.t. variables $\mathbf{w}$ under the $\|\cdot\|_2$ norm. Then

$$\mathcal{N}(\mathcal{F}, M\epsilon, L^\infty) \leq \mathcal{N}(\mathcal{W}, M\epsilon/(\beta_1 r_1 + b_1), \|\cdot\|_2) \leq (1 + \frac{2r_1(\beta_1 r_1 + b_1)}{M\epsilon})^d,$$

which implies that

$$\begin{aligned}
\int_0^1 \sqrt{\log(\mathcal{N}(\mathcal{F}, M\epsilon, L^\infty)}\mathrm{d}\epsilon &\leq \sqrt{d} \int_0^1 \sqrt{\log(1 + \frac{2r_1(\beta_1 r_1 + b_1)}{M\epsilon})}\mathrm{d}\epsilon \\
&\leq \sqrt{d} \int_0^1 \sqrt{\frac{2r_1(\beta_1 r_1 + b_1)}{M\epsilon}}\mathrm{d}\epsilon = 2\sqrt{\frac{2r_1 d(\beta_1 r_1 + b_1)}{M}}.
\end{aligned}$$

We have completed this proof. $\qquad\square$

**Lemma 2** (Uniform Convergence-II). *If Assumption 1 holds, then for any distribution $\mathbb{P}$, with the probability at least $1 - \delta > 0$,*

$$\left\| \nabla R_{\mathcal{S}}(\mathbf{h_w}) - \nabla R_{\mathbb{P}}(\mathbf{h_w}) \right\|_2 \leq B \sqrt{\frac{2 \log(2/\delta)}{n}} + C \sqrt{\frac{M(r_1 + 1)d}{n}},$$

*where $n = |\mathcal{S}|$, $d$ is the dimension of $\mathcal{W}$, and $C$ is a uniform constant.*

*Proof of Lemma 2.* Denote $\ell(\mathbf{v}, \mathbf{h_w}(\mathbf{x}), y) = \langle \nabla \ell(\mathbf{h_w}(\mathbf{x}), y), \mathbf{v} \rangle$ by the loss function over parameter space $\mathcal{W} \times \{\mathbf{v} = 1 : \mathbf{v} \in \mathbb{R}^d\}$. Let $b$ is the upper bound of $\ell(\mathbf{v}, \mathbf{h_w}(\mathbf{x}), y)$. Using the same techniques used in Lemma 1, we can prove that with the probability at least $1 - \delta > 0$, for any $\mathbf{w} \in \mathcal{W}$ and any unit vector $\mathbf{v} \in \mathbb{R}^d$,

$$\langle \nabla R_{\mathcal{S}}(\mathbf{h_w}) - \nabla R_{\mathbb{P}}(\mathbf{h_w}), \mathbf{v} \rangle \leq b \sqrt{\frac{2 \log(2/\delta)}{n}} + C \sqrt{\frac{b(r_1 + 1)\beta_1 d}{n}},$$

which implies that

$$\left\| \nabla R_{\mathcal{S}}(\mathbf{h_w}) - \nabla R_{\mathbb{P}}(\mathbf{h_w}) \right\|_2 \leq b \sqrt{\frac{2 \log(2/\delta)}{n}} + C \sqrt{\frac{b(r_1 + 1)\beta_1 d}{n}}.$$

Note that Proposition 1 implies that

$$b\beta_1 \leq M.$$

Proposition 2 implies that

$$b \leq \sqrt{2\beta_1 M}.$$

We have completed this proof. $\qquad\square$

**Lemma 3.** *Let $\mathcal{S}_{wild}^{in} \subset \mathcal{S}_{wild}$ be the samples drawn from $\mathbb{P}_{in}$. With the probability at least $1 - \delta > 0$,*

$$\left| |\mathcal{S}_{wild}^{in}|/|\mathcal{S}_{wild}| - (1 - \pi) \right| \leq \sqrt{\frac{\log(2/\delta)}{2|\mathcal{S}_{wild}|}},$$

*which implies that*

$$\left| |\mathcal{S}_{wild}^{in}| - (1 - \pi)|\mathcal{S}_{wild}| \right| \leq \sqrt{\frac{\log(2/\delta)|\mathcal{S}_{wild}|}{2}}.$$

*Proof of Lemma 3.* Let $X_i$ be the random variable corresponding to the case whether $i$-th data in the wild data is drawn from $\mathbb{P}_{in}$, i.e., $X_i = 1$, if $i$-th data is drawn from $\mathbb{P}_{in}$; otherwise, $X_i = 0$. Applying Hoeffding's inequality, we can get that with the probability at least $1 - \delta > 0$,

$$\left| |\mathcal{S}_{wild}^{in}|/|\mathcal{S}_{wild}| - (1 - \pi) \right| \leq \sqrt{\frac{\log(2/\delta)}{2|\mathcal{S}_{wild}|}}. \tag{26}$$

$\qquad\square$

### E.3 Necessary Lemmas and Theorems for Theorem 1

**Lemma 4.** *With the probability at least $1 - \delta > 0$, the ERM optimizer $\mathbf{w}_{\mathcal{S}}$ is the $\min_{\mathbf{w} \in \mathcal{W}} \mathbf{R}_{\mathbb{P}}(\mathbf{h}_{\mathbf{w}}) + O(1/\sqrt{n})$-risk point, i.e.,*

$$R_{\mathcal{S}}(\mathbf{h}_{\mathbf{w}_{\mathcal{S}}}) \leq \min_{\mathbf{w} \in \mathcal{W}} R_{\mathbb{P}}(\mathbf{h}_{\mathbf{w}}) + M\sqrt{\frac{\log(1/\delta)}{2n}},$$

*where $n = |\mathcal{S}|$.*

*Proof of Lemma 4.* Let $\mathbf{w}^* \in \arg\min_{\mathbf{w} \in \mathcal{W}} \mathbf{R}_{\mathbb{P}}(\mathbf{h}_{\mathbf{w}})$. Applying Hoeffding's inequality, we obtain that with the probability at least $1 - \delta > 0$,

$$R_{\mathcal{S}}(\mathbf{h}_{\mathbf{w}_{\mathcal{S}}}) - \min_{\mathbf{w} \in \mathcal{W}} R_{\mathbb{P}}(\mathbf{h}_{\mathbf{w}}) \leq R_{\mathcal{S}}(\mathbf{h}_{\mathbf{w}^*}) - R_{\mathbb{P}}(\mathbf{h}_{\mathbf{w}^*}) \leq M\sqrt{\frac{\log(1/\delta)}{2n}}.$$

$\square$

**Lemma 5.** *If Assumptions 1 and 2 hold, then for any data $\mathcal{S} \sim \mathbb{P}^n$ and $\mathcal{S}' \sim \mathbb{P}^{n'}$, with the probability at least $1 - \delta > 0$,*

$$R_{\mathcal{S}'}(\mathbf{h}_{\mathbf{w}_{\mathcal{S}}}) \leq \min_{\mathbf{w} \in \mathcal{W}} R_{\mathbb{P}}(\mathbf{h}_{\mathbf{w}}) + C\sqrt{\frac{Mr_1(\beta_1 r_1 + b_1)d}{n}} + C\sqrt{\frac{Mr_1(\beta_1 r_1 + b_1)d}{n'}}$$
$$+ 2M\sqrt{\frac{2\log(6/\delta)}{n}} + M\sqrt{\frac{2\log(6/\delta)}{n'}},$$

$$R_{\mathbb{P}}(\mathbf{h}_{\mathbf{w}_{\mathcal{S}}}, \widehat{\mathbf{h}}) \leq R_{\mathbb{P}}(\mathbf{h}_{\mathbf{w}_{\mathcal{S}}}) \leq \min_{\mathbf{w} \in \mathcal{W}} R_{\mathbb{P}}(\mathbf{h}_{\mathbf{w}}) + C\sqrt{\frac{Mr_1(\beta_1 r_1 + b_1)d}{n}} + 2M\sqrt{\frac{2\log(6/\delta)}{n}},$$

*where $C$ is a uniform constant, and*

$$R_{\mathbb{P}}(\mathbf{h}_{\mathbf{w}_{\mathcal{S}}}, \widehat{\mathbf{h}}) = \mathbb{E}_{(\mathbf{x},y) \sim \mathbb{P}} \ell(\mathbf{h}_{\mathbf{w}_{\mathcal{S}}}(\mathbf{x}), \widehat{\mathbf{h}}(\mathbf{x})).$$

*Proof of Lemma 5.* Let $\mathbf{w}_{\mathcal{S}'} \in \arg\min_{\mathbf{w} \in \mathcal{W}} R_{\mathcal{S}'}(\mathbf{h}_{\mathbf{w}})$ and $\mathbf{w}^* \in \arg\min_{\mathbf{w} \in \mathcal{W}} R_{\mathbb{P}}(\mathbf{h}_{\mathbf{w}^*})$. By Lemma 1 and Hoeffding Inequality, we obtain that with the probability at least $1 - \delta > 0$,

$$R_{\mathcal{S}'}(\mathbf{h}_{\mathbf{w}_{\mathcal{S}}}) - R_{\mathbb{P}}(\mathbf{h}_{\mathbf{w}^*})$$
$$\leq R_{\mathcal{S}'}(\mathbf{h}_{\mathbf{w}_{\mathcal{S}}}) - R_{\mathbb{P}}(\mathbf{h}_{\mathbf{w}_{\mathcal{S}}}) + R_{\mathbb{P}}(\mathbf{h}_{\mathbf{w}_{\mathcal{S}}}) - R_{\mathcal{S}}(\mathbf{h}_{\mathbf{w}_{\mathcal{S}}}) + R_{\mathcal{S}}(\mathbf{w}^*) - R_{\mathbb{P}}(\mathbf{w}^*)$$
$$\leq C\sqrt{\frac{Mr_1(\beta_1 r_1 + b_1)d}{n}} + C\sqrt{\frac{Mr_1(\beta_1 r_1 + b_1)d}{n'}} + 2M\sqrt{\frac{2\log(6/\delta)}{n}} + M\sqrt{\frac{2\log(6/\delta)}{n'}},$$
$$R_{\mathbb{P}}(\mathbf{h}_{\mathbf{w}_{\mathcal{S}}}) - R_{\mathbb{P}}(\mathbf{h}_{\mathbf{w}^*}) \leq R_{\mathbb{P}}(\mathbf{h}_{\mathbf{w}_{\mathcal{S}}}) - R_{\mathcal{S}}(\mathbf{h}_{\mathbf{w}_{\mathcal{S}}}) + R_{\mathcal{S}}(\mathbf{w}^*) - R_{\mathbb{P}}(\mathbf{w}^*)$$
$$\leq C\sqrt{\frac{Mr_1(\beta_1 r_1 + b_1)d}{n}} + 2M\sqrt{\frac{2\log(6/\delta)}{n}}.$$

$\square$

**Lemma 6.** *If Assumptions 1 and 2 hold, then for any data $\mathcal{S} \sim \mathbb{P}^n$ and $\mathcal{S}' \sim \mathbb{P}^{n'}$, with the probability at least $1 - 2\delta > 0$,*

$$\mathbb{E}_{(\mathbf{x},y) \sim \mathcal{S}'} \big\| \nabla \ell(\mathbf{h}_{\mathbf{w}_{\mathcal{S}}}(\mathbf{x}), \widehat{\mathbf{h}}(\mathbf{x})) - \nabla R_{\mathcal{S}}(\mathbf{h}_{\mathbf{w}_{\mathcal{S}}}) \big\|_2^2 \leq 8\beta_1 \min_{\mathbf{w} \in \mathcal{W}} R_{\mathbb{P}}(\mathbf{h}_{\mathbf{w}})$$
$$+ C\sqrt{\frac{Mr_1(\beta_1 r_1 + b_1)d}{n}} + C\sqrt{\frac{Mr_1(\beta_1 r_1 + b_1)d}{n'}}$$
$$+ 3M\sqrt{\frac{2\log(6/\delta)}{n}} + M\sqrt{\frac{2\log(6/\delta)}{n'}}.$$

*where $C$ is a uniform constant.*

*Proof of Lemma 6.* By Propositions 2, 3 and Lemmas 4 and 5, with the probability at least $1 - 2\delta > 0$,

$$
\mathbb{E}_{(\mathbf{x},y)\sim\mathcal{S}'}\big\|\nabla\ell(\mathbf{h}_{\mathbf{w}_\mathcal{S}}(\mathbf{x}),\widehat{\mathbf{h}}(\mathbf{x})) - \nabla R_\mathcal{S}(\mathbf{h}_{\mathbf{w}_\mathcal{S}})\big\|_2^2
$$

$$
\leq 2\mathbb{E}_{(\mathbf{x},y)\sim\mathcal{S}'}\big\|\nabla\ell(\mathbf{h}_{\mathbf{w}_\mathcal{S}}(\mathbf{x}),\widehat{\mathbf{h}}(\mathbf{x}))\big\|_2^2 + 2\big\|\nabla R_\mathcal{S}(\mathbf{h}_{\mathbf{w}_\mathcal{S}})\big\|_2^2
$$

$$
\leq 4\beta_1\big(R_{\mathcal{S}'}(\mathbf{h}_{\mathbf{w}_\mathcal{S}},\widehat{\mathbf{h}}) + R_\mathcal{S}(\mathbf{h}_{\mathbf{w}_\mathcal{S}})\big) \leq 4\beta_1\big(R_{\mathcal{S}'}(\mathbf{h}_{\mathbf{w}_\mathcal{S}}) + R_\mathcal{S}(\mathbf{h}_{\mathbf{w}_\mathcal{S}})\big)
$$

$$
\leq 4\beta_1\Big[2\min_{\mathbf{w}\in\mathcal{W}} R_\mathbb{P}(\mathbf{h}_{\mathbf{w}}) + C\sqrt{\frac{Mr_1(\beta_1 r_1 + b_1)d}{n}} + C\sqrt{\frac{Mr_1(\beta_1 r_1 + b_1)d}{n'}}
$$

$$
+ 3M\sqrt{\frac{2\log(6/\delta)}{n}} + M\sqrt{\frac{2\log(6/\delta)}{n'}}\Big].
$$

$\square$

**Lemma 7.** *Let $\mathcal{S}_{wild}^{in} \subset \mathcal{S}_{wild}$ be samples drawn from $\mathbb{P}_{in}$. If Assumptions 1 and 2 hold, then for any data $\mathcal{S}_{wild} \sim \mathbb{P}_{wild}^m$ and $\mathcal{S} \sim \mathbb{P}_{in}^n$, with the probability at least $1 - \frac{7}{3}\delta > 0$,*

$$
\mathbb{E}_{\mathbf{x}\sim\mathcal{S}_{wild}^{in}}\big\|\nabla\ell(\mathbf{h}_{\mathbf{w}_\mathcal{S}}(\mathbf{x}),\widehat{\mathbf{h}}(\mathbf{x})) - \nabla R_\mathcal{S}(\mathbf{h}_{\mathbf{w}_\mathcal{S}})\big\|_2^2 \leq 8\beta_1\min_{\mathbf{w}\in\mathcal{W}} R_\mathbb{P}(\mathbf{h}_{\mathbf{w}})
$$

$$
+ 4\beta_1\Big[C\sqrt{\frac{Mr_1(\beta_1 r_1 + b_1)d}{n}} + C\sqrt{\frac{Mr_1(\beta_1 r_1 + b_1)d}{(1-\pi)m - \sqrt{m\log(6/\delta)/2}}}
$$

$$
+ 3M\sqrt{\frac{2\log(6/\delta)}{n}} + M\sqrt{\frac{2\log(6/\delta)}{(1-\pi)m - \sqrt{m\log(6/\delta)/2}}}\Big],
$$

*where $C$ is a uniform constant.*

*Proof of Lemma 7.* Lemma 3 and Lemma 6 imply this result. $\square$

**Lemma 8.** *If Assumptions 1 and 2 hold, then for any data $\mathcal{S} \sim \mathbb{P}_{in}^n$, with the probability at least $1 - \delta > 0$,*

$$
\big\|\nabla R_\mathcal{S}(\mathbf{h}_{\mathbf{w}_\mathcal{S}}(\mathbf{x}),\widehat{\mathbf{h}}(\mathbf{x})) - \nabla R_\mathcal{S}(\mathbf{h}_{\mathbf{w}_\mathcal{S}})\big\|_2^2 \leq 8\beta_1\min_{\mathbf{w}\in\mathcal{W}} R_\mathbb{P}(\mathbf{h}_{\mathbf{w}}) + 4M\sqrt{\frac{\log(1/\delta)}{2n}}.
$$

*Proof of Lemma 8.* With the probability at least $1 - \delta > 0$,

$$
\big\|\nabla R_\mathcal{S}(\mathbf{h}_{\mathbf{w}_\mathcal{S}}(\mathbf{x}),\widehat{\mathbf{h}}(\mathbf{x})) - \nabla R_\mathcal{S}(\mathbf{h}_{\mathbf{w}_\mathcal{S}})\big\|_2
$$

$$
\leq \big\|\nabla R_\mathcal{S}(\mathbf{h}_{\mathbf{w}_\mathcal{S}}(\mathbf{x}),\widehat{\mathbf{h}}(\mathbf{x}))\big\|_2 + \big\|\nabla R_\mathcal{S}(\mathbf{h}_{\mathbf{w}_\mathcal{S}})\big\|_2
$$

$$
\leq \sqrt{2\beta_1 R_\mathcal{S}(\mathbf{h}_{\mathbf{w}_\mathcal{S}}(\mathbf{x}),\widehat{\mathbf{h}}(\mathbf{x}))} + \sqrt{2\beta_1 R_\mathcal{S}(\mathbf{h}_{\mathbf{w}_\mathcal{S}})}
$$

$$
\leq 2\sqrt{2\beta_1\big(\min_{\mathbf{w}\in\mathcal{W}} R_\mathbb{P}(\mathbf{h}_{\mathbf{w}}) + M\sqrt{\frac{\log(1/\delta)}{2n}}\big)}.
$$

$\square$

**Theorem 5.** *If Assumptions 1 and 2 hold and there exists $\eta \in (0,1)$ such that $\Delta = (1-\eta)^2 \zeta^2 - 8\beta_1 \min_{\mathbf{w} \in \mathcal{W}} R_{\mathbb{P}_{in}}(\mathbf{h}_{\mathbf{w}}) > 0$, when*

$$n = \Omega\Big(\frac{\tilde{M} + M(r_1+1)d}{\Delta \eta^2} + \frac{M^2 d}{(\gamma - R_{in}^*)^2}\Big), \quad m = \Omega\Big(\frac{\tilde{M} + M(r_1+1)d}{\eta^2 \zeta^2}\Big),$$

*with the probability at least $97/100$,*

$$\mathbb{E}_{\tilde{\mathbf{x}}_i \sim \mathcal{S}_{wild}} \tau_i > \frac{98\eta^2 \zeta^2}{100}.$$

*Proof of Theorem 5.* **Claim 1.** With the probability at least $1 - 2\delta > 0$, for any $\mathbf{w} \in \mathcal{W}$,

$$d_{\mathbf{w}}^{\ell}(\mathbb{P}_{\text{in}}, \mathbb{P}_{\text{wild}}) - d_{\mathbf{w}}^{\ell}(\mathcal{S}_X^{\text{in}}, \mathcal{S}_{\text{wild}}) \le B\sqrt{\frac{2\log(2/\delta)}{n}} + B\sqrt{\frac{2\log(2/\delta)}{m}}$$
$$+ C\sqrt{\frac{M(r_1+1)d}{n}} + C\sqrt{\frac{M(r_1+1)d}{m}},$$

where $B = \sqrt{2\beta_1 M}$, $\mathcal{S}_X^{\text{in}}$ is the feature part of $\mathcal{S}^{\text{in}}$ and $C$ is a uniform constant.

We prove this Claim: by Lemma 2, it is notable that with the probability at least $1 - 2\delta > 0$,

$$d_{\mathbf{w}}^{\ell}(\mathbb{P}_{\text{in}}, \mathbb{P}_{\text{wild}}) - d_{\mathbf{w}}^{\ell}(\mathcal{S}_X^{\text{in}}, \mathcal{S}_{\text{wild}})$$
$$\le \big\|\nabla R_{\mathbb{P}_{\text{in}}}(\mathbf{h}_{\mathbf{w}}, \widehat{\mathbf{h}}) - \nabla R_{\mathbb{P}_{\text{wild}}}(\mathbf{h}_{\mathbf{w}}, \widehat{\mathbf{h}})\big\|_2 - \big\|\nabla R_{\mathcal{S}_X^{\text{in}}}(\mathbf{h}_{\mathbf{w}}, \widehat{\mathbf{h}}) - \nabla R_{\mathcal{S}_{\text{wild}}}(\mathbf{h}_{\mathbf{w}}, \widehat{\mathbf{h}})\big\|_2$$
$$\le \big\|\nabla R_{\mathbb{P}_{\text{in}}}(\mathbf{h}_{\mathbf{w}}, \widehat{\mathbf{h}}) - \nabla R_{\mathbb{P}_{\text{wild}}}(\mathbf{h}_{\mathbf{w}}, \widehat{\mathbf{h}}) - \nabla R_{\mathcal{S}_X^{\text{in}}}(\mathbf{h}_{\mathbf{w}}, \widehat{\mathbf{h}}) + \nabla R_{\mathcal{S}_{\text{wild}}}(\mathbf{h}_{\mathbf{w}}, \widehat{\mathbf{h}})\big\|_2$$
$$\le \big\|\nabla R_{\mathbb{P}_{\text{in}}}(\mathbf{h}_{\mathbf{w}}, \widehat{\mathbf{h}}) - \nabla R_{\mathcal{S}_X^{\text{in}}}(\mathbf{h}_{\mathbf{w}}, \widehat{\mathbf{h}})\big\|_2 + \big\|\nabla R_{\mathbb{P}_{\text{wild}}}(\mathbf{h}_{\mathbf{w}}, \widehat{\mathbf{h}}) - \nabla R_{\mathcal{S}_{\text{wild}}}(\mathbf{h}_{\mathbf{w}}, \widehat{\mathbf{h}})\big\|_2$$
$$\le B\sqrt{\frac{2\log(2/\delta)}{n}} + B\sqrt{\frac{2\log(2/\delta)}{m}} + C\sqrt{\frac{B(r_1+1)\beta_1 d}{n}} + C\sqrt{\frac{B(r_1+1)\beta_1 d}{m}}$$
$$\le B\sqrt{\frac{2\log(2/\delta)}{n}} + B\sqrt{\frac{2\log(2/\delta)}{m}} + C\sqrt{\frac{M(r_1+1)d}{n}} + C\sqrt{\frac{M(r_1+1)d}{m}}.$$

**Claim 2.** When

$$\sqrt{n} = \Omega\Big(\frac{M\sqrt{d} + M\sqrt{\log(6/\delta)}}{\gamma - \min_{\mathbf{w} \in \mathcal{W}} R_{\mathbb{P}_{\text{in}}}(\mathbf{h}_{\mathbf{w}})}\Big), \tag{27}$$

with the probability at least $1 - 4\delta > 0$,

$$\mathbb{E}_{\tilde{\mathbf{x}}_i \sim \mathcal{S}_{\text{wild}}} \tau_i \ge \Big(\zeta - B\sqrt{\frac{2\log(2/\delta)}{n}} - B\sqrt{\frac{2\log(2/\delta)}{m}}$$
$$- C\sqrt{\frac{M(r_1+1)d}{n}} - C\sqrt{\frac{M(r_1+1)d}{m}} - 2\sqrt{2\beta_1\big(\min_{\mathbf{w} \in \mathcal{W}} R_{\mathbb{P}}(\mathbf{h}_{\mathbf{w}}) + M\sqrt{\frac{\log(1/\delta)}{2n}}\big)}\Big)^2.$$

We prove this Claim: let $\mathbf{v}^*$ be the top-1 right singular vector computed in our algorithm, and

$$\tilde{\mathbf{v}} \in \underset{\|\mathbf{v}\| \le 1}{\arg\max} \Big\langle \mathbb{E}_{(\mathbf{x}_i, y_i) \sim \mathcal{S}^{\text{in}}} \nabla\ell(\mathbf{h}_{\mathbf{w}_{\mathcal{S}^{\text{in}}}}(\mathbf{x}_i), y_i) - \mathbb{E}_{\tilde{\mathbf{x}}_i \in \mathcal{S}_{\text{wild}}} \nabla\ell(\mathbf{h}_{\mathbf{w}_{\mathcal{S}^{\text{in}}}}(\tilde{\mathbf{x}}_i), \widehat{\mathbf{h}}(\tilde{\mathbf{x}}_i)), \mathbf{v} \Big\rangle.$$

Then with the probability at least $1 - 4\delta > 0$,

$$\mathbb{E}_{\tilde{\mathbf{x}}_i \sim \mathcal{S}_{\text{wild}}} \tau_i$$

$$= \mathbb{E}_{\tilde{\mathbf{x}}_i \sim \mathcal{S}_{\text{wild}}} \left( \left\langle \nabla\ell(\mathbf{h}_{\mathbf{w}_{\mathcal{S}^{\text{in}}}}(\tilde{\mathbf{x}}_i), \widehat{\mathbf{h}}(\tilde{\mathbf{x}}_i)) - \mathbb{E}_{(\mathbf{x}_j,y_j)\sim\mathcal{S}^{\text{in}}} \nabla\ell(\mathbf{h}_{\mathbf{w}_{\mathcal{S}^{\text{in}}}}(\mathbf{x}_j), y_j), \mathbf{v}^* \right\rangle \right)^2$$

$$\geq \mathbb{E}_{\tilde{\mathbf{x}}_i \sim \mathcal{S}_{\text{wild}}} \left( \left\langle \nabla\ell(\mathbf{h}_{\mathbf{w}_{\mathcal{S}^{\text{in}}}}(\tilde{\mathbf{x}}_i), \widehat{\mathbf{h}}(\tilde{\mathbf{x}}_i)) - \mathbb{E}_{(\mathbf{x}_j,y_j)\sim\mathcal{S}^{\text{in}}} \nabla\ell(\mathbf{h}_{\mathbf{w}_{\mathcal{S}^{\text{in}}}}(\mathbf{x}_j), y_j), \tilde{\mathbf{v}} \right\rangle \right)^2$$

$$\geq \left( \left\langle \mathbb{E}_{(\mathbf{x}_j,y_j)\sim\mathcal{S}^{\text{in}}} \nabla\ell(\mathbf{h}_{\mathbf{w}_{\mathcal{S}^{\text{in}}}}(\mathbf{x}_j), y_j) - \mathbb{E}_{\tilde{\mathbf{x}}_i \sim \mathcal{S}_{\text{wild}}} \nabla\ell(\mathbf{h}_{\mathbf{w}_{\mathcal{S}^{\text{in}}}}(\tilde{\mathbf{x}}_i), \widehat{\mathbf{h}}(\tilde{\mathbf{x}}_i)), \tilde{\mathbf{v}} \right\rangle \right)^2$$

$$= \left\| \mathbb{E}_{(\mathbf{x}_j,y_j)\sim\mathcal{S}^{\text{in}}} \nabla\ell(\mathbf{h}_{\mathbf{w}_{\mathcal{S}^{\text{in}}}}(\mathbf{x}_j), y_j) - \mathbb{E}_{\tilde{\mathbf{x}}_i \sim \mathcal{S}_{\text{wild}}} \nabla\ell(\mathbf{h}_{\mathbf{w}_{\mathcal{S}^{\text{in}}}}(\tilde{\mathbf{x}}_i), \widehat{\mathbf{h}}(\tilde{\mathbf{x}}_i)) \right\|_2^2$$

$$\geq \left( d^\ell_{\mathbf{w}_{\mathcal{S}^{\text{in}}}}(\mathcal{S}_X^{\text{in}}, \mathcal{S}_{\text{wild}}) \right.$$

$$- \left\| \mathbb{E}_{(\mathbf{x}_j,y_j)\sim\mathcal{S}^{\text{in}}} \nabla\ell(\mathbf{h}_{\mathbf{w}_{\mathcal{S}^{\text{in}}}}(\mathbf{x}_j), y_j) - \mathbb{E}_{(\mathbf{x}_j,y_j)\sim\mathcal{S}^{\text{in}}} \nabla\ell(\mathbf{h}_{\mathbf{w}_{\mathcal{S}^{\text{in}}}}(\mathbf{x}_j), \widehat{\mathbf{h}}(\mathbf{x}_j)) \right\|_2 \Big)^2$$

$$\geq \left( \zeta - B\sqrt{\frac{2\log(2/\delta)}{n}} - B\sqrt{\frac{2\log(2/\delta)}{m}} \right.$$

$$\left. -C\sqrt{\frac{M(r_1+1)d}{n}} - C\sqrt{\frac{M(r_1+1)d}{m}} - 2\sqrt{2\beta_1 (\min_{\mathbf{w}\in\mathcal{W}} R_{\mathbb{P}}(\mathbf{h}_{\mathbf{w}}) + M\sqrt{\frac{\log(1/\delta)}{2n}})} \right)^2.$$

In above inequality, we have used the results in Claim 1, Assumption 2, Lemma 5 and Lemma 8.

**Claim 3.** Given $\delta = 1/100$, then when

$$n = \Omega\big(\frac{\tilde{M} + M(r_1+1)d}{\Delta\eta^2}\big), \quad m = \Omega\big(\frac{\tilde{M} + M(r_1+1)d}{\eta^2\zeta^2}\big),$$

the following inequality holds:

$$\left( \zeta - B\sqrt{\frac{2\log(2/\delta)}{n}} - B\sqrt{\frac{2\log(2/\delta)}{m}} - C\sqrt{\frac{M(r_1+1)d}{n}} - C\sqrt{\frac{M(r_1+1)d}{m}} \right.$$

$$\left. -2\sqrt{2\beta_1 (\min_{\mathbf{w}\in\mathcal{W}} R_{\mathbb{P}}(\mathbf{h}_{\mathbf{w}}) + M\sqrt{\frac{\log(1/\delta)}{2n}})} \right)^2 > \frac{98\eta^2\theta^2}{100}.$$

We prove this Claim: when

$$n \geq \frac{64\sqrt{\log(10)}\beta_1 M}{\Delta},$$

it is easy to check that

$$(1-\eta)\zeta \geq 2\sqrt{2\beta_1 (\min_{\mathbf{w}\in\mathcal{W}} R_{\mathbb{P}}(\mathbf{h}_{\mathbf{w}}) + M\sqrt{\frac{\log(1/\delta)}{2n}})}.$$

Additionally, when

$$n \geq \frac{200^2 \log 200 B^2}{\eta^2\zeta^2} + \frac{200^2 C^2 M(r_1+1)d}{2\eta^2\zeta^2},$$

it is easy to check that

$$\frac{\eta\zeta}{100} \geq B\sqrt{\frac{2\log(200)}{n}} + C\sqrt{\frac{M(r_1+1)d}{n}}.$$

Because

$$\max\{\frac{200^2 \log 200 B^2}{\eta^2\zeta^2} + \frac{200^2 C^2 M(r_1+1)d}{2\eta^2\zeta^2}, \frac{64\sqrt{\log(10)}\beta_1 M}{\Delta}\} \leq O\big(\frac{\tilde{M} + M(r_1+1)d}{\Delta\eta^2}\big),$$

we conclude that when

$$n = \Omega\big(\frac{\tilde{M} + M(r_1+1)d}{\Delta\eta^2}\big),$$

$$\eta - 2\sqrt{2\beta_1(\min_{\mathbf{w}\in\mathcal{W}} R_{\mathbb{P}}(\mathbf{h_w}) + M\sqrt{\frac{\log(1/\delta)}{2n}})} - B\sqrt{\frac{2\log(200)}{n}} + C\sqrt{\frac{M(r_1+1)d}{n}} \geq \frac{99}{100}\eta\zeta. \quad (28)$$

When

$$m \geq \frac{200^2 \log 200 B^2}{\eta^2\zeta^2} + \frac{200^2 C^2 M(r_1+1)d}{2\eta^2\zeta^2},$$

we have

$$\frac{\eta\zeta}{100} \geq B\sqrt{\frac{2\log(200)}{m}} + C\sqrt{\frac{M(r_1+1)d}{m}}.$$

Therefore, if

$$m = \Omega\big(\frac{\tilde{M} + M(r_1+1)d}{\eta^2\zeta^2}\big),$$

we have

$$\frac{\eta\zeta}{100} \geq B\sqrt{\frac{2\log(200)}{m}} + C\sqrt{\frac{M(r_1+1)d}{m}}. \quad (29)$$

Combining inequalities 27, 28 and 29, we complete this proof.

$$\square$$

### E.4 NECESSARY LEMMAS FOR THEOREM 3

Let

$$R_{\mathcal{S}_T}^-(\mathbf{g_\theta}) = \mathbb{E}_{\mathbf{x}\sim\mathcal{S}_T}\ell_{\mathsf{b}}(\mathbf{g_\theta}(\mathbf{x}), y_-), \ R_{\mathcal{S}_{\text{wild}}^{\text{out}}}^-(\mathbf{g_\theta}) = \mathbb{E}_{\mathbf{x}\sim\mathcal{S}_{\text{wild}}^{\text{out}}}\ell_{\mathsf{b}}(\mathbf{g_\theta}(\mathbf{x}), y_-),$$

$$R_{\mathcal{S}^{\text{in}}}^+(\mathbf{g_\theta}) = \mathbb{E}_{\mathbf{x}\sim\mathcal{S}^{\text{in}}}\ell_{\mathsf{b}}(\mathbf{g_\theta}(\mathbf{x}), y_+), \ R_{\mathbb{P}_{\text{in}}}^+(\mathbf{g_\theta}) = \mathbb{E}_{\mathbf{x}\sim\mathcal{S}^{\text{in}}}\ell_{\mathsf{b}}(\mathbf{g_\theta}(\mathbf{x}), y_+),$$

and

$$R_{\mathbb{P}_{\text{out}}}^-(\mathbf{g_\theta}) = \mathbb{E}_{\mathbf{x}\sim\mathcal{S}^{\text{in}}}\ell_{\mathsf{b}}(\mathbf{g_\theta}(\mathbf{x}), y_-).$$

Let

$$\mathcal{S}_+^{\text{out}} = \{\tilde{\mathbf{x}}_i \in \mathcal{S}_{\text{wild}}^{\text{out}} : \tilde{\mathbf{x}}_i \leq T\}, \ \ \mathcal{S}_-^{\text{in}} = \{\tilde{\mathbf{x}}_i \in \mathcal{S}_{\text{wild}}^{\text{in}} : \tilde{\mathbf{x}}_i > T\},$$

$$\mathcal{S}_-^{\text{out}} = \{\tilde{\mathbf{x}}_i \in \mathcal{S}_{\text{wild}}^{\text{out}} : \tilde{\mathbf{x}}_i > T\}, \ \ \mathcal{S}_+^{\text{in}} = \{\tilde{\mathbf{x}}_i \in \mathcal{S}_{\text{wild}}^{\text{in}} : \tilde{\mathbf{x}}_i \leq T\}.$$

Then

$$S_T = \mathcal{S}_-^{\text{out}} \cup \mathcal{S}_-^{\text{in}}, \ \ \ S_{\text{wild}}^{\text{out}} = \mathcal{S}_-^{\text{out}} \cup \mathcal{S}_+^{\text{out}}.$$

Let

$$\Delta(n,m) = \frac{1 - \min\{1, \Delta_\zeta^\eta/\pi\}}{1 - T/M'} + O\Big(\frac{\beta_1 M\sqrt{d}}{1 - T/M'}\sqrt{\frac{1}{\pi^2 n}}\Big)$$

$$+ O\Big(\frac{\beta_1 M\sqrt{d} + \sqrt{1-\pi}\Delta_\zeta^\eta/\pi}{1 - T/M'}\sqrt{\frac{1}{\pi^2(1-\pi)m}}\Big).$$

$$\delta(n,m) = \frac{8\beta_1 R_{\text{in}}^*}{T} + O\Big(\frac{\beta_1 M\sqrt{d}}{T}\sqrt{\frac{1}{n}}\Big) + O\Big(\frac{\beta_1 M\sqrt{d}}{T}\sqrt{\frac{1}{m}}\Big).$$

**Lemma 9.** *Under the conditions of Theorem 1, with the probability at least* $9/10$,

$$|\mathcal{S}_T| \leq |\mathcal{S}_-^{in}| + |\mathcal{S}_{wild}^{out}| \leq \delta(n,m)|\mathcal{S}_{wild}^{in}| + |\mathcal{S}_{wild}^{out}|,$$

$$|\mathcal{S}_T| \geq |\mathcal{S}_-^{out}| \geq \big[1 - \Delta(n,m)\big]|\mathcal{S}_{wild}^{out}|.$$

$$|\mathcal{S}_+^{out}| \leq \Delta(n,m)|\mathcal{S}_{wild}^{out}|.$$

*Proof of Lemma 9.* It is a conclusion of Theorem 1. □

**Lemma 10.** *Under the conditions of Theorem 1, with the probability at least* $9/10$,

$$\frac{-\delta(n,m)|\mathcal{S}_{wild}^{in}|}{[\delta(n,m)|\mathcal{S}_{wild}^{in}| + |\mathcal{S}_{wild}^{out}|]|\mathcal{S}_{wild}^{out}|} \leq \frac{1}{|\mathcal{S}_T|} - \frac{1}{|\mathcal{S}_{wild}^{out}|} \leq \frac{\Delta(n,m)}{[1-\Delta(n,m)]|\mathcal{S}_{wild}^{out}|}.$$

*Proof of Lemma 10.* This can be conclude by Lemma 9 directly. □

**Lemma 11.** *Under the conditions of Theorem 1, with the probability at least* $9/10$,

$$R_{\mathcal{S}_T}^-(\mathbf{g}_{\boldsymbol{\theta}}) - R_{\mathcal{S}_{wild}^{out}}^-(\mathbf{g}_{\boldsymbol{\theta}}) \leq \frac{L\Delta(n,m)}{[1-\Delta(n,m)]} + \frac{L\delta(n,m)}{[1-\Delta(n,m)]} \cdot \left(\frac{1-\pi}{\pi} + O\left(\sqrt{\frac{1}{\pi^4 m}}\right)\right),$$

$$R_{\mathcal{S}_{wild}^{out}}^-(\mathbf{g}_{\boldsymbol{\theta}}) - R_{\mathcal{S}_T}^-(\mathbf{g}_{\boldsymbol{\theta}}) \leq \frac{L\Delta(n,m)}{[1-\Delta(n,m)]} + L\Delta(n,m).$$

*Proof of Lemma 11.* It is clear that

$$
\begin{aligned}
R_{\mathcal{S}_T}^-(\mathbf{g}_{\boldsymbol{\theta}}) - R_{\mathcal{S}_{wild}^{out}}^-(\mathbf{g}_{\boldsymbol{\theta}}) =& \frac{|\mathcal{S}_-^{out}|}{|\mathcal{S}_T|} R_{\mathcal{S}_-^{out}}^-(\mathbf{g}_{\boldsymbol{\theta}}) + \frac{|\mathcal{S}_-^{in}|}{|\mathcal{S}_T|} R_{\mathcal{S}_-^{in}}^-(\mathbf{g}_{\boldsymbol{\theta}}) \\
& - \frac{|\mathcal{S}_-^{out}|}{|\mathcal{S}_{wild}^{out}|} R_{\mathcal{S}_-^{out}}^-(\mathbf{g}_{\boldsymbol{\theta}}) - \frac{|\mathcal{S}_+^{out}|}{|\mathcal{S}_{wild}^{out}|} R_{\mathcal{S}_+^{out}}^-(\mathbf{g}_{\boldsymbol{\theta}}) \\
\leq& \frac{L\Delta(n,m)}{[1-\Delta(n,m)]} + \frac{L\delta(n,m)|\mathcal{S}_{wild}^{in}|}{[1-\Delta(n,m)]|\mathcal{S}_{wild}^{out}|} \\
\leq& \frac{L\Delta(n,m)}{[1-\Delta(n,m)]} + \frac{L\delta(n,m)}{[1-\Delta(n,m)]} \cdot \left(\frac{1-\pi}{\pi} + O\left(\sqrt{\frac{1}{\pi^4 m}}\right)\right). \\
R_{\mathcal{S}_{wild}^{out}}^-(\mathbf{g}_{\boldsymbol{\theta}}) - R_{\mathcal{S}_T}^-(\mathbf{g}_{\boldsymbol{\theta}}) =& -\frac{|\mathcal{S}_-^{out}|}{|\mathcal{S}_T|} R_{\mathcal{S}_-^{out}}^-(\mathbf{g}_{\boldsymbol{\theta}}) - \frac{|\mathcal{S}_-^{in}|}{|\mathcal{S}_T|} R_{\mathcal{S}_-^{in}}^-(\mathbf{g}_{\boldsymbol{\theta}}) \\
& + \frac{|\mathcal{S}_-^{out}|}{|\mathcal{S}_{wild}^{out}|} R_{\mathcal{S}_-^{out}}^-(\mathbf{g}_{\boldsymbol{\theta}}) + \frac{|\mathcal{S}_+^{out}|}{|\mathcal{S}_{wild}^{out}|} R_{\mathcal{S}_+^{out}}^-(\mathbf{g}_{\boldsymbol{\theta}}) \\
\leq& \frac{L\Delta(n,m)}{[1-\Delta(n,m)]} + L\Delta(n,m).
\end{aligned}
$$

□

**Lemma 12.** *Let* $\Delta(T) = 1 - \delta(T)$. *Under the conditions of Theorem 1, for any* $\eta' > 0$, *when*

$$n = \Omega\left(\frac{\tilde{M}^2 d}{\eta'^2 \pi^2 (1-T/M')^2 \Delta(T)^2}\right), \quad m = \Omega\left(\frac{\tilde{M}^2 d\pi^2 + \Delta_\zeta^\eta(1-\pi)}{\eta'^2 \pi^4 (1-\pi)(1-T/M')^2 \Delta(T)^2}\right),$$

*with the probability at least* $9/10$,

$$R_{\mathcal{S}_T}^-(\mathbf{g}_{\boldsymbol{\theta}}) - R_{\mathcal{S}_{wild}^{out}}^-(\mathbf{g}_{\boldsymbol{\theta}}) \leq \frac{L\Delta(n,m)}{(1-\eta')\Delta(T)} + \frac{L\delta(n,m)}{(1-\eta')\Delta(T)} \cdot \left(\frac{1-\pi}{\pi} + O\left(\sqrt{\frac{1}{\pi^4 m}}\right)\right),$$

$$R_{\mathcal{S}_{wild}^{out}}^-(\mathbf{g}_{\boldsymbol{\theta}}) - R_{\mathcal{S}_T}^-(\mathbf{g}_{\boldsymbol{\theta}}) \leq \frac{1.2L}{1-\delta(T)}\delta(T) + L\delta(T)$$

$$+ O\left(\frac{L\beta_1 M\sqrt{d}(1+T)}{\min\{\pi, \Delta_\zeta^\eta\}T}\sqrt{\frac{1}{n}}\right) + O\left(\frac{L(\beta_1 M\sqrt{d} + \Delta_\zeta^\eta)(1+T)}{\min\{\pi, \Delta_\zeta^\eta\}T}\sqrt{\frac{1}{\pi^2(1-\pi)m}}\right).$$

*Proof of Lemma 12.* This can be concluded by Lemma 11 and by the fact that $(1 - \eta')\Delta(T) \geq 1 - \Delta(n, m)$ directly. $\qquad\square$

**Lemma 13.** *Let* $\Delta(T) = 1 - \delta(T)$. *Under the conditions of Theorem 1, when*

$$n = \Omega\Big(\frac{\tilde{M}^2 d}{\min\{\pi, \Delta_\zeta^\eta\}^2}\Big), \quad m = \Omega\Big(\frac{\tilde{M}^2 d + \Delta_\zeta^\eta}{\pi^2(1 - \pi)\min\{\pi, \Delta_\zeta^\eta\}^2}\Big),$$

*with the probability at least* $9/10$, *for any* $0 < T < 0.9M' \min\{1, \Delta_\zeta^\eta/\pi\}$,

$$R_{\mathcal{S}_T}^-(\mathbf{g}_\theta) - R_{\mathcal{S}_{wild}^{out}}^-(\mathbf{g}_\theta) \leq \frac{1.2L}{1 - \delta(T)}\delta(T) + \frac{9L\beta_1(1 - \pi)}{T\pi(1 - \delta(T))}R_{in}^*$$

$$+ O\Big(\frac{L\beta_1 M \sqrt{d}(1 + T)}{\min\{\pi, \Delta_\zeta^\eta\}T}\sqrt{\frac{1}{n}}\Big) + O\Big(\frac{L(\beta_1 M \sqrt{d} + \Delta_\zeta^\eta)(1 + T)}{\min\{\pi, \Delta_\zeta^\eta\}T}\sqrt{\frac{1}{\pi^2(1 - \pi)m}}\Big),$$

$$R_{\mathcal{S}_{wild}^{out}}^-(\mathbf{g}_\theta) - R_{\mathcal{S}_T}^-(\mathbf{g}_\theta) \leq \frac{1.2L}{1 - \delta(T)}\delta(T) + L\delta(T)$$

$$+ O\Big(\frac{L\beta_1 M \sqrt{d}(1 + T)}{\min\{\pi, \Delta_\zeta^\eta\}T}\sqrt{\frac{1}{n}}\Big) + O\Big(\frac{L(\beta_1 M \sqrt{d} + \Delta_\zeta^\eta)(1 + T)}{\min\{\pi, \Delta_\zeta^\eta\}T}\sqrt{\frac{1}{\pi^2(1 - \pi)m}}\Big).$$

*Proof of Lemma 13.* Using Lemma 13 with $\eta = 8/9$, we obtain that

$$R_{\mathcal{S}_T}^-(\mathbf{g}_\theta) - R_{\mathcal{S}_{wild}^{out}}^-(\mathbf{g}_\theta)$$

$$\leq \frac{1.2L\delta(T)}{1 - \delta(T)} + \frac{9L\beta_1(1 - \pi)}{T\pi(1 - \delta(T))}R_{in}^* + \frac{L\epsilon(n)}{\Delta(T)} + \frac{L\bar{\epsilon}(n)}{\pi\Delta(T)} + \frac{L\epsilon(m)}{\Delta(T)}$$

$$+ \frac{8L\beta_1 R_{in}^*}{\pi^2\Delta(T)T}O\Big(\sqrt{\frac{1}{m}}\Big) + \frac{L\bar{\epsilon}(m)}{\pi^2\Delta(T)}O\Big(\sqrt{\frac{1}{m}}\Big) + \frac{L\bar{\epsilon}(m)}{\pi\Delta(T)} + \frac{L\bar{\epsilon}(n)}{\pi^2\Delta(T)}O\Big(\sqrt{\frac{1}{m}}\Big),$$

where

$$\epsilon(n) = O\Big(\frac{\beta_1 M \sqrt{d}}{1 - T/M'}\sqrt{\frac{1}{\pi^2 n}}\Big), \quad \epsilon(m) = O\Big(\frac{\beta_1 M \sqrt{d} + \sqrt{1 - \pi}\Delta_\zeta^\eta/\pi}{1 - T/M'}\sqrt{\frac{1}{\pi^2(1 - \pi)m}}\Big).$$

$$\bar{\epsilon}(n) = O\Big(\frac{\beta_1 M \sqrt{d}}{T}\sqrt{\frac{1}{n}}\Big), \quad \bar{\epsilon}(m) = O\Big(\frac{\beta_1 M \sqrt{d}}{T}\sqrt{\frac{1}{(1 - \pi)m}}\Big).$$

Using the condition that $0 < T < 0.9M' \min\{1, \Delta_\zeta^\eta/\pi\}$, we have

$$\frac{1}{\Delta(T)}\Big[\frac{1}{T} + \frac{1}{1 - T/M'}\Big] \leq O\Big(\frac{T + 1}{\min\{1, \Delta_\zeta^\eta/\pi\}T}\Big).$$

Then, we obtain that

$$R_{\mathcal{S}_T}^-(\mathbf{g}_\theta) - R_{\mathcal{S}_{wild}^{out}}^-(\mathbf{g}_\theta) \leq \frac{1.2L}{1 - \delta(T)}\delta(T) + \frac{9L\beta_1(1 - \pi)}{T\pi(1 - \delta(T))}R_{in}^*$$

$$+ O\Big(\frac{L\beta_1 M \sqrt{d}(1 + T)}{\min\{\pi, \Delta_\zeta^\eta\}T}\sqrt{\frac{1}{n}}\Big) + O\Big(\frac{L(\beta_1 M \sqrt{d} + \Delta_\zeta^\eta)(1 + T)}{\min\{\pi, \Delta_\zeta^\eta\}T}\sqrt{\frac{1}{\pi^2(1 - \pi)m}}\Big).$$

Using the similar strategy, we can obtain that

$$R_{\mathcal{S}_{wild}^{out}}^-(\mathbf{g}_\theta) - R_{\mathcal{S}_T}^-(\mathbf{g}_\theta) \leq \frac{1.2L}{1 - \delta(T)}\delta(T) + L\delta(T)$$

$$+ O\Big(\frac{L\beta_1 M \sqrt{d}(1 + T)}{\min\{\pi, \Delta_\zeta^\eta\}T}\sqrt{\frac{1}{n}}\Big) + O\Big(\frac{L(\beta_1 M \sqrt{d} + \Delta_\zeta^\eta)(1 + T)}{\min\{\pi, \Delta_\zeta^\eta\}T}\sqrt{\frac{1}{\pi^2(1 - \pi)m}}\Big).$$

$\qquad\square$

**Lemma 14.** *Under the conditions of Theorem 1, when*

$$n = \Omega\Big(\frac{\tilde{M}^2 d}{\min\{\pi, \Delta_\zeta^\eta\}^2}\Big), \quad m = \Omega\Big(\frac{\tilde{M}^2 d + \Delta_\zeta^\eta}{\pi^2(1-\pi)\min\{\pi, \Delta_\zeta^\eta\}^2}\Big),$$

*with the probability at least* $0.895$*, for any* $0 < T < 0.9M' \min\{1, \Delta_\zeta^\eta/\pi\}$,

$$R_{\mathcal{S}_T}^-(\mathbf{g}_\theta) - R_{\mathbb{P}_{out}}^-(\mathbf{g}_\theta) \leq \frac{1.2L}{1-\delta(T)}\delta(T) + \frac{9L\beta_1(1-\pi)}{T\pi(1-\delta(T))}R_{in}^*$$

$$+ O\Big(\frac{L\beta_1 M\sqrt{d}(1+T)}{\min\{\pi, \Delta_\zeta^\eta\}T}\sqrt{\frac{1}{n}}\Big) + O\Big(\frac{L\max\{\beta_1 M\sqrt{d}, \sqrt{d'}, \Delta_\zeta^\eta\}(1+T)}{\min\{\pi, \Delta_\zeta^\eta\}T}\sqrt{\frac{1}{\pi^2(1-\pi)m}}\Big),$$

$$R_{\mathbb{P}_{out}}^-(\mathbf{g}_\theta) - R_{\mathcal{S}_T}^-(\mathbf{g}_\theta) \leq \frac{1.2L}{1-\delta(T)}\delta(T) + L\delta(T)$$

$$+ O\Big(\frac{L\beta_1 M\sqrt{d}(1+T)}{\min\{\pi, \Delta_\zeta^\eta\}T}\sqrt{\frac{1}{n}}\Big) + O\Big(\frac{L\max\{\beta_1 M\sqrt{d}, \sqrt{d'}, \Delta_\zeta^\eta\}(1+T)}{\min\{\pi, \Delta_\zeta^\eta\}T}\sqrt{\frac{1}{\pi^2(1-\pi)m}}\Big).$$

*Proof.* By Lemmas 1 and 3, under the condition of this lemma, we can obtain that with the high probability,

$$\big|R_{\mathbb{P}_{out}}^-(\mathbf{g}_\theta) - R_{\mathcal{S}_{wild}^{out}}^-(\mathbf{g}_\theta)\big| \leq O\Big(L\sqrt{\frac{d'}{\pi m}}\Big).$$

Then by Lemma 13, we can prove this lemma. □

## F   EMPIRICAL VERIFICATION ON THE MAIN THEOREMS

**Verification on the regulatory conditions.** In Table 5, we provide empirical verification on whether the distribution discrepancy $\zeta$ satisfies the necessary regulatory condition in Theorem 2, i.e., $\zeta \geq 2.011\sqrt{8\beta_1 R_{in}^*} + 1.011\sqrt{\pi}$. We use CIFAR-100 as ID and TEXTURES as the wild OOD data.

Since $R_{in}^*$ is the optimal ID risk, i.e., $R_{in}^* = \min_{\mathbf{w}\in\mathcal{W}} \mathbb{E}_{(\mathbf{x},y)\sim\mathbb{P}_{\mathcal{X}\mathcal{Y}}}\ell(\mathbf{h}_\mathbf{w}(\mathbf{x}), y)$, it can be a small value close to 0 in over-parametrized neural networks (Frei et al., 2022; Bartlett et al., 2020). Therefore, we can omit the value of $2.011\sqrt{8\beta_1 R_{in}^*}$. The empirical result shows that $\zeta$ can easily satisfy the regulatory condition in Theorem 2, which means our bound is useful in practice.

Table 5: Discrepancy value $\zeta$ with different ratios $\pi$.

| $\pi$ | 0.05 | 0.1 | 0.2 | 0.5 | 0.7 | 0.9 | 1.0 |
|---|---|---|---|---|---|---|---|
| $\zeta$ | 0.91 | 1.09 | 1.43 | 2.49 | 3.16 | 3.86 | 4.18 |
| $1.011\sqrt{\pi}$ | 0.23 | 0.32 | 0.45 | 0.71 | 0.84 | 0.96 | 1.01 |

**Verification on the filtering errors and OOD detection results with varying $\pi$.** In Table 6, we empirically verify the value of $\text{ERR}_{out}$ and $\text{ERR}_{in}$ in Theorem 1 and the corresponding OOD detection results with various mixing ratios $\pi$. We use CIFAR-100 as ID and TEXTURES as the wild OOD data. The result aligns well with our observation of the bounds presented in Section 4.1 of the main paper.

## G   ADDITIONAL EXPERIMENTAL DETAILS

**Dataset details.** For Table 1, following WOODS (Katz-Samuels et al., 2022), we split the data as follows: We use 70% of the OOD datasets (including TEXTURES, PLACES365, LSUN-RESIZE

Table 6: The values of $\mathrm{ERR_{in}}$, $\mathrm{ERR_{out}}$ and the OOD detection results with various mixing ratios $\pi$.

| $\pi$ | 0.05 | 0.1 | 0.2 | 0.5 | 0.7 | 0.9 | 1.0 |
|---|---|---|---|---|---|---|---|
| $\mathrm{ERR_{out}}$ | 0.37 | 0.30 | 0.22 | 0.20 | 0.23 | 0.26 | 0.29 |
| $\mathrm{ERR_{in}}$ | 0.031 | 0.037 | 0.045 | 0.047 | 0.047 | 0.048 | 0.048 |
| FPR95 | 5.77 | 5.73 | 5.71 | 5.64 | 5.79 | 5.88 | 5.92 |

and LSUN-C) for the OOD data in the wild. We use the remaining samples for testing-time OOD detection. For SVHN, we use the training set for the OOD data in the wild and use the test set for evaluation.

**Training details.** Following WOODS (Katz-Samuels et al., 2022), we use Wide ResNet (Zagoruyko & Komodakis, 2016) with 40 layers and widen factor of 2 for the classification model $\mathbf{h_w}$. We train the ID classifier $\mathbf{h_w}$ using stochastic gradient descent with a momentum of 0.9, weight decay of 0.0005, and an initial learning rate of 0.1. We train for 100 epochs using cosine learning rate decay, a batch size of 128, and a dropout rate of 0.3. For the OOD classifier $\mathbf{g_\theta}$, we load the pre-trained ID classifier of $\mathbf{h_w}$ and add an additional linear layer which takes in the penultimate-layer features for binary classification. We set the initial learning rate to 0.001 and fine-tune for 100 epochs by Eq. 8. We add the binary classification loss to the ID classification loss and set the loss weight for binary classification to 10. The other details are kept the same as training $\mathbf{h_w}$.

# H ADDITIONAL RESULTS ON CIFAR-10

In Table 7, we compare our SAL with baselines with the ID data to be CIFAR-10, where the strong performance of SAL still holds.

Table 7: OOD detection performance on CIFAR-10 as ID. All methods are trained on Wide ResNet-40-2 for 100 epochs with $\pi = 0.1$. For each dataset, we create corresponding wild mixture distribution $\mathbb{P}_{wild} := (1 - \pi)\mathbb{P}_{in} + \pi\mathbb{P}_{out}$ for training and test on the corresponding OOD dataset. Values are percentages **averaged over 10 runs**. Bold numbers highlight the best results. Table format credit to (Katz-Samuels et al., 2022).

| Methods | SVHN | | PLACES365 | | LSUN-C | | LSUN-RESIZE | | TEXTURES | | Average | | ID ACC |
|---|---|---|---|---|---|---|---|---|---|---|---|---|---|
| | FPR95 | AUROC | FPR95 | AUROC | FPR95 | AUROC | FPR95 | AUROC | FPR95 | AUROC | FPR95 | AUROC | |
| | | | | | | With $\mathbb{P}_{in}$ only | | | | | | | |
| MSP | 48.49 | 91.89 | 59.48 | 88.20 | 30.80 | 95.65 | 52.15 | 91.37 | 59.28 | 88.50 | 50.04 | 91.12 | 94.84 |
| ODIN | 33.35 | 91.96 | 57.40 | 84.49 | 15.52 | 97.04 | 26.62 | 94.57 | 49.12 | 84.97 | 36.40 | 90.61 | 94.84 |
| Mahalanobis | 12.89 | 97.62 | 68.57 | 84.61 | 39.22 | 94.15 | 42.62 | 93.23 | 15.00 | 97.33 | 35.66 | 93.34 | 94.84 |
| Energy | 35.59 | 90.96 | 40.14 | 89.89 | 8.26 | 98.35 | 27.58 | 94.24 | 52.79 | 85.22 | 32.87 | 91.73 | 94.84 |
| KNN | 24.53 | 95.96 | 25.29 | 95.69 | 25.55 | 95.26 | 27.57 | 94.71 | 50.90 | 89.14 | 30.77 | 94.15 | 94.84 |
| ReAct | 40.76 | 89.57 | 41.44 | 90.44 | 14.38 | 97.21 | 33.63 | 93.58 | 53.63 | 86.59 | 36.77 | 91.48 | 94.84 |
| DICE | 35.44 | 89.65 | 46.83 | 86.69 | 6.32 | 98.68 | 28.93 | 93.56 | 53.62 | 82.20 | 34.23 | 90.16 | 94.84 |
| ASH | 6.51 | 98.65 | 48.45 | 88.34 | 0.90 | 99.73 | 4.96 | 98.92 | 24.34 | 95.09 | 17.03 | 96.15 | 94.84 |
| CSI | 17.30 | 97.40 | 34.95 | 93.64 | 1.95 | 99.55 | 12.15 | 98.01 | 20.45 | 95.93 | 17.36 | 96.91 | 94.17 |
| KNN+ | 2.99 | 99.41 | 24.69 | 94.84 | 2.95 | 99.39 | 11.22 | 97.98 | 9.65 | 98.37 | 10.30 | 97.99 | 93.19 |
| | | | | | | With $\mathbb{P}_{in}$ and $\mathbb{P}_{wild}$ | | | | | | | |
| OE | 0.85 | 99.82 | 23.47 | 94.62 | 1.84 | 99.65 | 0.33 | 99.93 | 10.42 | 98.01 | 7.38 | 98.41 | 94.07 |
| Energy (w/ OE) | 4.95 | 98.92 | 17.26 | 95.84 | 1.93 | 99.49 | 5.04 | 98.83 | 13.43 | 96.69 | 8.52 | 97.95 | 94.81 |
| WOODS | 0.15 | 99.97 | 12.49 | 97.00 | 0.22 | 99.94 | 0.03 | **99.99** | 5.95 | 98.79 | 3.77 | 99.14 | 94.84 |
| SAL | **0.02** | **99.98** | **2.57** | **99.24** | **0.07** | **99.99** | **0.01** | **99.99** | **0.90** | **99.74** | **0.71** | **99.78** | 93.65 |
| (Ours) | ±0.00 | ±0.00 | ±0.03 | ±0.00 | ±0.01 | ±0.00 | ±0.00 | ±0.00 | ±0.02 | ±0.01 | ±0.01 | ±0.00 | ±0.57 |

# I ADDITIONAL RESULTS ON UNSEEN OOD DATASETS

In Table 8, we evaluate SAL on unseen OOD datasets, which are different from the OOD data we use in the wild. Here we consistently use 300K RANDOM IMAGES as the unlabeled wild dataset and CIFAR-10 as labeled in-distribution data. We use the 5 different OOD datasets (TEXTURES, PLACES365, LSUN-RESIZE, SVHN and LSUN-C) for evaluation. When evaluating on 300K RANDOM IMAGES, we use 99% of the 300K RANDOM IMAGES dataset (Hendrycks et al., 2019) as the wild OOD data and the remaining 1% of the dataset for evaluation. $\pi$ is set to 0.1. We observe that SAL can perform competitively on unseen datasets as well, compared to the most relevant baseline WOODS.

Table 8: Evaluation on unseen OOD datasets. We use CIFAR-10 as ID and 300K RANDOM IMAGES as the wild data. All methods are trained on Wide ResNet-40-2 for 50 epochs. Bold numbers highlight the best results.

| Methods | OOD Datasets | | | | | | | | | | | | ID ACC |
| | SVHN | | PLACES365 | | LSUN-C | | LSUN-RESIZE | | TEXTURES | | 300K RAND. IMG. | | |
| | FPR95 | AUROC | FPR95 | AUROC | FPR95 | AUROC | FPR95 | AUROC | FPR95 | AUROC | FPR95 | AUROC | |
|---|---|---|---|---|---|---|---|---|---|---|---|---|---|
| OE | 13.18 | 97.34 | 30.54 | 93.31 | 5.87 | 98.86 | 14.32 | 97.44 | 25.69 | 94.35 | 30.69 | 92.80 | 94.21 |
| Energy (w/ OE) | 8.52 | 98.13 | 23.74 | 94.26 | 2.78 | 99.38 | 9.05 | 98.13 | 22.32 | 94.72 | 24.59 | 93.99 | 94.54 |
| WOODS | 5.70 | **98.54** | 19.14 | 95.74 | **1.31** | **99.66** | 4.13 | **99.01** | 17.92 | 96.43 | 19.82 | 95.52 | 94.74 |
| SAL | **4.94** | 97.53 | **14.76** | **96.25** | 2.73 | 98.23 | **3.46** | 98.15 | **11.60** | **97.21** | **10.20** | **97.23** | 93.48 |

Following (He et al., 2023), we use the CIFAR-100 as ID, Tiny ImageNet-crop (TINc)/Tiny ImageNet-resize (TINr) dataset as the OOD in the wild dataset and TINr/TINc as the test OOD. The comparison with baselines is shown below, where the strong performance of SAL still holds.

Table 9: Additional results on unseen OOD datasets with CIFAR-100 as ID. Bold numbers are superior results.

| Methods | OOD Datasets | | | |
| | TINR | | TINC | |
| | FPR95 | AUROC | FPR95 | AUROC |
|---|---|---|---|---|
| STEP | 72.31 | 74.59 | 48.68 | 91.14 |
| TSL | 57.52 | 82.29 | 29.48 | 94.62 |
| SAL (Ours) | **43.11** | **89.17** | **19.30** | **96.29** |

## J ADDITIONAL RESULTS ON NEAR OOD DETECTION

In this section, we investigate the performance of SAL on near OOD detection, which is a more challenging OOD detection scenario where the OOD data has a closer distribution to the in-distribution. Specifically, we use the CIFAR-10 as the in-distribution data and CIFAR-100 training set as the OOD data in the wild. During test time, we use the test set of CIFAR-100 as the OOD for evaluation. With a mixing ratio $\pi$ of 0.1, our SAL achieves an FPR95 of 24.51% and AUROC of 95.55% compared to 38.92% (FPR95) and 93.27% (AUROC) of WOODS.

In addition, we study near OOD detection in a different data setting, i.e., the first 50 classes of CIFAR-100 as ID and the last 50 classes as OOD. The comparison with the most competitive baseline WOODS is reported as follows.

Table 10: Near OOD detection with the first 50 classes of CIFAR-100 as ID and the last 50 classes as OOD. Bold numbers are superior results.

| Methods | OOD dataset | | |
| | CIFAR-50 | | |
| | FPR95 | AUROC | ID ACC |
|---|---|---|---|
| WOODS | 41.28 | 89.74 | 74.17 |
| SAL | **29.71** | **93.13** | 73.86 |

## K ADDITIONAL RESULTS ON USING MULTIPLE SINGULAR VECTORS

In this section, we ablate on the effect of using $c$ singular vectors to calculate the filtering score (Eq. 6). Specifically, we calculate the scores by projecting the gradient $\nabla \ell(\mathbf{h}_{\mathbf{w}_{\mathcal{S}^{\text{in}}}}(\tilde{\mathbf{x}}_i), \widehat{y}_{\tilde{\mathbf{x}}_i}) - \bar{\nabla}$ for the wild data $\tilde{\mathbf{x}}_i$ to each of the singular vectors. The final filtering score is the average over the $c$ scores. The result is summarized in Table 11. We observe that using the top 1 singular vector for projection achieves the best performance. As revealed in Eq. 7, the top 1 singular vector $\mathbf{v}$ maximizes the total distance from the projected gradients (onto the direction of $\mathbf{v}$) to the origin (sum over all points in $\mathcal{S}_{\text{wild}}$), where outliers lie approximately close to and thus leads to a better separability between the ID and OOD in the wild.

Table 11: The effect of the number of singular vectors used for the filtering score. Models are trained on Wide ResNet-40-2 for 100 epochs with $\pi = 0.1$. We use TEXTURES as the wild OOD data and CIFAR-100 as the ID.

| Number of singular vectors $c$ | FPR95 | AUROC |
|---|---|---|
| 1 | **5.73** | **98.65** |
| 2 | 6.28 | 98.42 |
| 3 | 6.93 | 98.43 |
| 4 | 7.07 | 98.37 |
| 5 | 7.43 | 98.27 |
| 6 | 7.78 | 98.22 |

## L  ADDITIONAL RESULTS ON CLASS-AGNOSTIC SVD

In this section, we evaluate our SAL by using class-agnostic SVD as opposed to class-conditional SVD as described in Section 3.1 of the main paper. Specifically, we maintain a class-conditional reference gradient $\bar{\nabla}_k$, one for each class $k \in [1, K]$, estimated on ID samples belonging to class $k$. Different from calculating the singular vectors based on gradient matrix with $\mathbf{G}_k$ (containing gradient vectors of wild samples being predicted as class $k$), we formulate a single gradient matrix $\mathbf{G}$ where each row is the vector $\nabla \ell(\mathbf{h}_{\mathbf{w}_{\mathcal{S}^{in}}}(\tilde{\mathbf{x}}_i), \widehat{y}_{\tilde{\mathbf{x}}_i}) - \bar{\nabla}_{\widehat{y}_{\tilde{\mathbf{x}}_i}}$, for $\tilde{\mathbf{x}}_i \in \mathcal{S}_{\text{wild}}$. The result is shown in Table 12, which shows a similar performance compared with using class-conditional SVD.

Table 12: The effect of using class-agnostic SVD. Models are trained on Wide ResNet-40-2 for 100 epochs with $\pi = 0.1$. CIFAR-100 is the in-distribution data. Bold numbers are superior results.

| Methods | OOD Datasets | | | | | | | | | | | | ID ACC |
|---|---|---|---|---|---|---|---|---|---|---|---|---|---|
| | SVHN | | PLACES365 | | LSUN-C | | LSUN-RESIZE | | TEXTURES | | Average | | |
| | FPR95 | AUROC | FPR95 | AUROC | FPR95 | AUROC | FPR95 | AUROC | FPR95 | AUROC | FPR95 | AUROC | |
| SAL (Class-agnostic SVD) | 0.12 | 99.43 | **3.27** | **99.21** | **0.04** | 99.92 | 0.03 | 99.27 | **5.18** | **98.77** | **1.73** | 99.32 | 73.31 |
| SAL | **0.07** | **99.95** | 3.53 | 99.06 | 0.06 | **99.94** | **0.02** | **99.95** | 5.73 | 98.65 | 1.88 | **99.51** | 73.71 |

## M  ADDITIONAL RESULTS ON POST-HOC FILTERING SCORE

We investigate the importance of training the binary classifier with the filtered candidate outliers for OOD detection in Tables 13 and 14. Specifically, we calculate our filtering score directly for the test ID and OOD data on the model trained on the labeled ID set $\mathcal{S}^{\text{in}}$ only. The results are shown in the row "SAL (Post-hoc)" in Tables 13 and 14. Without explicit knowledge of the OOD data, the OOD detection performance degrades significantly compared to training an additional binary classifier (a 15.73% drop on FPR95 for SVHN with CIFAR-10 as ID). However, the post-hoc filtering score can still outperform most of the baselines that use $\mathbb{P}_{\text{in}}$ only (c.f. Table 1), showcasing its effectiveness.

Table 13: OOD detection results of using post-hoc filtering score on CIFAR-10 as ID. SAL is trained on Wide ResNet-40-2 for 100 epochs with $\pi = 0.1$. Bold numbers are superior results.

| Methods | OOD Datasets | | | | | | | | | | | | ID ACC |
|---|---|---|---|---|---|---|---|---|---|---|---|---|---|
| | SVHN | | PLACES365 | | LSUN-C | | LSUN-RESIZE | | TEXTURES | | Average | | |
| | FPR95 | AUROC | FPR95 | AUROC | FPR95 | AUROC | FPR95 | AUROC | FPR95 | AUROC | FPR95 | AUROC | |
| SAL (Post-hoc) | 15.75 | 93.09 | 23.18 | 86.35 | 6.28 | 96.72 | 15.59 | 89.83 | 23.63 | 87.72 | 16.89 | 90.74 | 94.84 |
| SAL | **0.02** | **99.98** | **2.57** | **99.24** | **0.07** | **99.99** | **0.01** | **99.99** | **0.90** | **99.74** | **0.71** | **99.78** | 93.65 |

Table 14: OOD detection results of using post-hoc filtering score on CIFAR-100 as ID. SAL is trained on Wide ResNet-40-2 for 100 epochs with $\pi = 0.1$. Bold numbers are superior results.

| Methods | OOD Datasets | | | | | | | | | | | | ID ACC |
|---|---|---|---|---|---|---|---|---|---|---|---|---|---|
| | SVHN | | PLACES365 | | LSUN-C | | LSUN-RESIZE | | TEXTURES | | Average | | |
| | FPR95 | AUROC | FPR95 | AUROC | FPR95 | AUROC | FPR95 | AUROC | FPR95 | AUROC | FPR95 | AUROC | |
| SAL (post-hoc) | 39.75 | 81.47 | 35.94 | 84.53 | 23.22 | 90.90 | 32.59 | 87.12 | 36.38 | 83.25 | 33.58 | 85.45 | 75.96 |
| SAL | **0.07** | **99.95** | **3.53** | **99.06** | **0.06** | **99.94** | **0.02** | **99.95** | **5.73** | **98.65** | **1.88** | **99.51** | 73.71 |

## N    ADDITIONAL RESULTS ON LEVERAGING THE CANDIDATE ID DATA

In this section, we investigate the effect of incorporating the filtered wild data which has a score smaller than the threshold $T$ (candidate ID data) for training the binary classifier $\mathbf{g}_{\boldsymbol{\theta}}$. Specifically, the candidate ID data and the labeled ID data are used jointly to train the binary classifier. The comparison with SAL on CIFAR-100 is shown as follows:

Table 15: OOD detection results of selecting candidate ID data for training on CIFAR-100 as ID. SAL is trained on Wide ResNet-40-2 for 100 epochs with $\pi = 0.1$. Bold numbers are superior results.

| Methods | OOD Datasets | | | | | | | | | | | | ID ACC |
| | SVHN | | PLACES365 | | LSUN-C | | LSUN-RESIZE | | TEXTURES | | Average | | |
| | FPR95 | AUROC | FPR95 | AUROC | FPR95 | AUROC | FPR95 | AUROC | FPR95 | AUROC | FPR95 | AUROC | |
|---|---|---|---|---|---|---|---|---|---|---|---|---|---|
| Candidate ID data | 1.23 | 99.87 | **2.62** | **99.18** | **0.04** | **99.95** | **0.02** | 99.91 | 4.71 | **98.97** | **1.72** | **99.58** | 73.83 |
| SAL (Ours) | **0.07** | **99.95** | 3.53 | 99.06 | 0.06 | 99.94 | **0.02** | **99.95** | 5.73 | 98.65 | 1.88 | 99.51 | 73.71 |

The result of selecting candidate ID data (and combine with labeled ID data) shows slightly better performance, which echoes our theory that the generalization bound of the OOD detector will be better if we have more ID training data (Theorem 3).

## O    ANALYSIS ON USING RANDOM LABELS

We present the OOD detection result of replacing the predicted labels with the random labels for the wild data as follows. The other experimental details are kept the same as SAL.

Table 16: OOD detection results of using random labels for the wild data on CIFAR-100 as ID. SAL is trained on Wide ResNet-40-2 for 100 epochs with $\pi = 0.1$. Bold numbers are superior results.

| Methods | OOD Datasets | | | | | | | | | | | | ID ACC |
| | SVHN | | PLACES365 | | LSUN-C | | LSUN-RESIZE | | TEXTURES | | Average | | |
| | FPR95 | AUROC | FPR95 | AUROC | FPR95 | AUROC | FPR95 | AUROC | FPR95 | AUROC | FPR95 | AUROC | |
|---|---|---|---|---|---|---|---|---|---|---|---|---|---|
| w/ Random labels | 39.36 | 89.31 | 77.98 | 78.31 | 47.46 | 88.90 | 67.28 | 80.23 | 54.86 | 86.92 | 57.39 | 84.73 | 73.68 |
| SAL (Ours) | **0.07** | **99.95** | **3.53** | **99.06** | **0.06** | **99.94** | **0.02** | **99.95** | **5.73** | **98.65** | **1.88** | **99.51** | 73.71 |

As we can observe, using the random labels leads to worse OOD detection performance because the gradient of the wild data can be wrong. In our theoretical analysis (Theorem 5), we have proved that using the predicted label can lead to a good separation of the wild ID and OOD data. However, the analysis using random labels might hold since it violates the assumption (Definitions 2 and 3) that the expected gradient of ID data should be different from that of wild data.

## P    DETAILS OF THE ILLUSTRATIVE EXPERIMENTS ON THE IMPACT OF PREDICTED LABELS

For calculating the filtering accuracy, SAL is trained on Wide ResNet-40-2 for 100 epochs with $\pi = 0.1$ on two separate ID datasets. The other training details are kept the same as Section 5.1 and Appendix G.

## Q    DETAILS OF FIGURE 2

For Figure 2 in the main paper, we generate the in-distribution data from three multivariate Gaussian distributions, forming three classes. The mean vectors are set to $[-2, 0], [2, 0]$ and $[0, 2\sqrt{3}]$, respectively. The covariance matrix for all three classes is set to $\begin{bmatrix} 0.25 & 0 \\ 0 & 0.25 \end{bmatrix}$. For each class, we generate $1,000$ samples.

For wild scenario 1, we generate the outlier data in the wild by sampling $100,000$ data points from a multivariate Gaussian $\mathcal{N}([0, \frac{2}{\sqrt{3}}], 7 \cdot \mathbf{I})$ where $\mathbf{I}$ is $2 \times 2$ identity matrix, and only keep the $1,000$ data points that have the largest distance to the mean vector $[0, \frac{2}{\sqrt{3}}]$. For wild scenario 2, we

generate the outlier data in the wild by sampling $1,000$ data points from a multivariate Gaussian $\mathcal{N}([10, \frac{2}{\sqrt{3}}], 0.25 \cdot \mathbf{I})$. For the in-distribution data in the wild, we sample $3,000$ data points per class from the same three multivariate Gaussian distributions as mentioned before.

## R   SOFTWARE AND HARDWARE

We run all experiments with Python 3.8.5 and PyTorch 1.13.1, using NVIDIA GeForce RTX 2080Ti GPUs.

## S   RESULTS WITH VARYING MIXING RATIOS

We provide additional results of SAL with varying $\pi$, i.e., 0.05, 0.2, 0.5, 0.9, and contrast with the baselines, which are shown below (CIFAR-100 as the in-distribution dataset). We found that the advantage of SAL still holds.

Table 17: OOD detection results with multiple mixing ratios $\pi$ with CIFAR-100 as ID. SAL is trained on Wide ResNet-40-2 for 100 epochs. Bold numbers are superior results.

| Methods | SVHN | | PLACES365 | | LSUN-C | | LSUN-RESIZE | | TEXTURES | | ID ACC |
|---|---|---|---|---|---|---|---|---|---|---|---|
| | FPR95 | AUROC | FPR95 | AUROC | FPR95 | AUROC | FPR95 | AUROC | FPR95 | AUROC | |
| | | | | | $\pi = 0.05$ | | | | | | |
| OE | 2.78 | 98.84 | 63.63 | 80.22 | 6.73 | 98.37 | 2.06 | 99.19 | 32.86 | 90.88 | 71.98 |
| Energy w/ OE | 2.02 | 99.17 | 56.18 | 83.33 | 4.32 | 98.42 | 3.96 | 99.29 | 40.41 | 89.80 | 73.45 |
| WOODS | 0.26 | 99.89 | 32.71 | 90.01 | **0.64** | 99.77 | **0.79** | 99.10 | 12.26 | 94.48 | 74.15 |
| SAL (Ours) | **0.17** | **99.90** | **6.21** | **96.87** | 0.94 | **99.79** | 0.84 | **99.37** | **5.77** | **97.12** | 73.99 |
| | | | | | $\pi = 0.2$ | | | | | | |
| OE | 2.59 | 98.90 | 55.68 | 84.36 | 4.91 | 99.02 | 1.97 | 99.37 | 25.62 | 93.65 | 73.72 |
| Energy w/ OE | 1.79 | 99.25 | 47.28 | 86.78 | 4.18 | 99.00 | 3.15 | 99.35 | 36.80 | 91.48 | 73.91 |
| WOODS | 0.22 | 99.82 | 29.78 | 91.28 | 0.52 | 99.79 | 0.89 | 99.56 | 10.06 | 95.23 | 73.49 |
| SAL (Ours) | **0.08** | **99.92** | **2.80** | **99.31** | **0.05** | **99.94** | **0.02** | **99.97** | **5.71** | **98.71** | 73.86 |
| | | | | | $\pi = 0.5$ | | | | | | |
| OE | 2.86 | 99.05 | 40.21 | 88.75 | 4.13 | 99.05 | 1.25 | 99.38 | 22.86 | 94.63 | 73.38 |
| Energy w/ OE | 2.71 | 99.34 | 34.82 | 90.05 | 3.27 | 99.18 | 2.54 | 99.23 | 30.16 | 94.76 | 72.76 |
| WOODS | 0.17 | 99.80 | 21.87 | 93.73 | 0.48 | 99.61 | 1.24 | 99.54 | 9.95 | 95.97 | 73.91 |
| SAL (Ours) | **0.02** | **99.98** | **1.27** | **99.62** | **0.04** | **99.96** | **0.01** | **99.99** | **5.64** | **99.16** | 73.77 |
| | | | | | $\pi = 0.9$ | | | | | | |
| OE | 0.84 | 99.36 | 19.78 | 96.29 | 1.64 | 99.57 | 0.51 | 99.75 | 12.74 | 94.95 | 72.02 |
| Energy w/ OE | 0.97 | 99.64 | 17.52 | 96.53 | 1.36 | 99.73 | 0.94 | 99.59 | 14.01 | 95.73 | 73.62 |
| WOODS | 0.05 | 99.98 | 11.34 | 95.83 | 0.07 | **99.99** | 0.03 | **99.99** | 6.72 | 98.73 | 73.86 |
| SAL (Ours) | **0.03** | **99.99** | **2.79** | **99.89** | **0.05** | **99.99** | **0.01** | **99.99** | **5.88** | **99.53** | 74.01 |

## T   COMPARISON WITH WEAKLY SUPERVISED OOD DETECTION BASELINES

We have additionally compared with the two related works (TSL (He et al., 2023) and STEP (Zhou et al., 2021)). To ensure a fair comparison, we strictly follow the experimental setting in TSL, and rerun SAL under the identical setup. The comparison on CIFAR-100 is shown as follows.

Table 18: Comparison with relevant baselines on CIFAR-100. Bold numbers are superior results.

| Methods | LSUN-C | | LSUN-RESIZE | |
|---|---|---|---|---|
| | FPR95 | AUROC | FPR95 | AUROC |
| STEP | **0.00** | 99.99 | 9.81 | 97.87 |
| TSL | **0.00** | **100.00** | 1.76 | 99.57 |
| SAL (Ours) | **0.00** | 99.99 | **0.58** | **99.95** |

## U    ADDITIONAL RESULTS ON DIFFERENT BACKBONES

We have additionally tried ResNet-18 and ResNet-34 as the network architectures—which are among the most used in OOD detection literature. The comparison with the baselines on CIFAR-100 is shown in the following tables, where SAL outperforms all the baselines across different architectures. These additional results support the effectiveness of our approach.

Table 19: OOD detection performance on CIFAR-100 as ID. All methods are trained on ResNet-18 for 100 epochs. For each dataset, we create corresponding wild mixture distribution $\mathbb{P}_{\text{wild}} = (1-\pi)\mathbb{P}_{\text{in}} + \pi\mathbb{P}_{\text{out}}$ for training and test on the corresponding OOD dataset. Bold numbers highlight the best results.

| Methods | OOD Datasets | | | | | | | | | | | | ID ACC |
| | SVHN | | PLACES365 | | LSUN-C | | LSUN-RESIZE | | TEXTURES | | Average | | |
| | FPR95 | AUROC | FPR95 | AUROC | FPR95 | AUROC | FPR95 | AUROC | FPR95 | AUROC | FPR95 | AUROC | |
|---|---|---|---|---|---|---|---|---|---|---|---|---|---|
| | With $\mathbb{P}_{\text{in}}$ only | | | | | | | | | | | | |
| MSP | 81.32 | 77.74 | 83.06 | 74.47 | 70.11 | 83.51 | 82.46 | 75.73 | 85.11 | 73.36 | 80.41 | 76.96 | 78.67 |
| ODIN | 40.94 | 93.29 | 87.71 | 71.46 | 28.72 | 94.51 | 79.61 | 82.13 | 83.63 | 72.37 | 64.12 | 82.75 | 78.67 |
| Mahalanobis | 22.44 | 95.67 | 92.66 | 61.39 | 68.90 | 86.30 | 23.07 | 94.20 | 62.39 | 79.39 | 53.89 | 83.39 | 78.67 |
| Energy | 81.74 | 84.56 | 82.23 | 76.68 | 34.78 | 93.93 | 73.57 | 82.99 | 85.87 | 74.94 | 71.64 | 82.62 | 78.67 |
| KNN | 83.62 | 72.76 | 82.09 | 80.03 | 65.96 | 84.82 | 71.05 | 81.24 | 76.88 | 77.90 | 75.92 | 79.35 | 78.67 |
| ReAct | 70.81 | 88.24 | 81.33 | 76.49 | 39.99 | 92.51 | 54.47 | 89.56 | 59.15 | 87.96 | 61.15 | 86.95 | 78.67 |
| DICE | 54.65 | 88.84 | 79.58 | 77.26 | 0.93 | 99.74 | 49.40 | 91.04 | 65.04 | 76.42 | 49.92 | 86.66 | 78.67 |
| CSI | 49.98 | 89.57 | 82.87 | 75.64 | 76.39 | 80.38 | 74.21 | 83.34 | 58.23 | 81.04 | 68.33 | 81.99 | 74.23 |
| KNN+ | 43.21 | 90.21 | 84.62 | 74.21 | 50.12 | 82.48 | 71.05 | 80.81 | 63.21 | 84.91 | 63.61 | 82.52 | 77.03 |
| | With $\mathbb{P}_{\text{in}}$ and $\mathbb{P}_{\text{wild}}$ | | | | | | | | | | | | |
| OE | 3.29 | 97.93 | 62.90 | 80.23 | 7.07 | 95.93 | 4.06 | **97.98** | 33.27 | 90.03 | 22.12 | 92.42 | 74.89 |
| Energy (w/ OE) | 3.12 | 94.27 | 59.38 | 82.19 | 9.12 | 91.23 | 7.28 | 95.39 | 43.92 | 90.11 | 24.56 | 90.64 | 77.92 |
| WOODS | 3.92 | 96.92 | 33.92 | 86.29 | 5.19 | 94.23 | **2.95** | 96.23 | 11.95 | 94.65 | 11.59 | 93.66 | 77.54 |
| SAL | **2.29** | **97.96** | **6.29** | **96.66** | **3.92** | **97.81** | 4.87 | 97.10 | **8.28** | **95.95** | **5.13** | **97.10** | 77.71 |

Table 20: OOD detection performance on CIFAR-100 as ID. All methods are trained on ResNet-34 for 100 epochs. For each dataset, we create corresponding wild mixture distribution $\mathbb{P}_{\text{wild}} = (1-\pi)\mathbb{P}_{\text{in}} + \pi\mathbb{P}_{\text{out}}$ for training and test on the corresponding OOD dataset. Bold numbers highlight the best results.

| Methods | OOD Datasets | | | | | | | | | | | | ID ACC |
| | SVHN | | PLACES365 | | LSUN-C | | LSUN-RESIZE | | TEXTURES | | Average | | |
| | FPR95 | AUROC | FPR95 | AUROC | FPR95 | AUROC | FPR95 | AUROC | FPR95 | AUROC | FPR95 | AUROC | |
|---|---|---|---|---|---|---|---|---|---|---|---|---|---|
| | With $\mathbb{P}_{\text{in}}$ only | | | | | | | | | | | | |
| MSP | 78.89 | 79.80 | 84.38 | 74.21 | 83.47 | 75.28 | 84.61 | 74.51 | 86.51 | 72.53 | 83.12 | 75.27 | 79.04 |
| ODIN | 70.16 | 84.88 | 82.16 | 75.19 | 76.36 | 80.10 | 79.54 | 79.16 | 85.28 | 75.23 | 78.70 | 79.11 | 79.04 |
| Mahalanobis | 87.09 | 80.62 | 84.63 | 73.89 | 84.15 | 79.43 | 83.18 | 78.83 | 61.72 | 84.87 | 80.15 | 79.53 | 79.04 |
| Energy | 66.91 | 85.25 | 81.41 | 76.37 | 59.77 | 86.69 | 66.52 | 84.49 | 79.01 | 79.96 | 70.72 | 82.55 | 79.04 |
| KNN | 81.12 | 73.65 | 79.62 | 78.21 | 63.29 | 85.56 | 73.92 | 79.77 | 73.29 | 80.35 | 74.25 | 79.51 | 79.04 |
| ReAct | 82.85 | 70.12 | 81.75 | 76.25 | 80.70 | 83.03 | 67.40 | 83.28 | 74.60 | 81.61 | 77.46 | 78.86 | 79.04 |
| DICE | 83.55 | 72.49 | 85.05 | 75.92 | 94.05 | 73.59 | 75.20 | 80.90 | 79.80 | 77.83 | 83.53 | 76.15 | 79.04 |
| CSI | 44.53 | 92.65 | 79.08 | 76.27 | 75.58 | 83.78 | 76.62 | 84.98 | 61.61 | 86.47 | 67.48 | 84.83 | 77.89 |
| KNN+ | 39.23 | 92.78 | 80.74 | 77.58 | 48.99 | 89.30 | 74.99 | 82.69 | 57.15 | 88.35 | 60.22 | 86.14 | 78.32 |
| | With $\mathbb{P}_{\text{in}}$ and $\mathbb{P}_{\text{wild}}$ | | | | | | | | | | | | |
| OE | 2.11 | 98.23 | 60.12 | 83.22 | 6.08 | 96.34 | 3.94 | 98.13 | 30.00 | 92.27 | 20.45 | 93.64 | 75.72 |
| Energy (w/ OE) | 1.94 | 95.03 | 68.84 | 85.94 | 7.66 | 92.04 | 6.86 | 97.63 | 40.82 | 93.07 | 25.22 | 92.74 | 78.75 |
| WOODS | 2.08 | 97.33 | 25.37 | 88.93 | 4.26 | 97.74 | 1.05 | 97.30 | 8.85 | 96.86 | 8.32 | 95.63 | 78.97 |
| SAL | **0.98** | **99.94** | **2.98** | **99.08** | **0.07** | **99.94** | **0.03** | **99.96** | **4.01** | **98.83** | **1.61** | **99.55** | 78.01 |

## V    BROADER IMPACT

Our project aims to improve the reliability and safety of modern machine learning models. From the theoretical perspective, our analysis can facilitate and deepen the understanding of the effect of unlabeled wild data for OOD detection. In Appendix F, we properly verify the necessary conditions and the value of our error bound using real-world datasets. Hence, we believe our theoretical framework has a broad utility and significance.

From the practical side, our study can lead to direct benefits and societal impacts, particularly when the wild data is abundant in the models' operating environment, such as in safety-critical applications i.e., autonomous driving and healthcare data analysis. Our study does not involve any human subjects or violation of legal compliance. We do not anticipate any potentially harmful consequences to our work. Through our study and releasing our code, we hope to raise stronger research and societal awareness towards the problem of exploring unlabeled wild data for out-of-distribution detection in real-world settings.

