# OpenReview forum: "How Does Unlabeled Data Provably Help Out-of-Distribution Detection?"
_ICLR.cc/2024/Conference — ICLR 2024 poster_

### Official Review · Reviewer_ZN1X · 2023-10-28

**Soundness:** 3 good
**Presentation:** 3 good
**Contribution:** 3 good
**Rating:** 6
**Confidence:** 3

**Summary:**

This study addresses wild Out-Of Distribution (OOD) detection, benefiting from the availability of more unlabeled data to enhance OOD data identification. The paper proposes the utilization of the top singular value as a criterion to differentiate between In-Distribution (ID) and OOD samples. The authors, grounded in novel theoretical insights, posit that ID samples should exhibit larger top singular values compared to OOD samples. Both experimental results and theoretical analyses corroborate the effectiveness of the proposed method, Separate And Learn (SAL). Overall, this work stands out for its solid foundation.

**Strengths:**

- Using gradients to make distinctions is a common and intuitive approach that makes sense to me. How ever, the method of distinguishing based on the top singular value is both intriguing and non-obvious. This innovative discovery holds significant importance for OOD detection.

- The availability of extra unlabeled data undoubtedly enables wild OOD detection to outperform traditional OOD detection, which does not utilize unlabeled data. However, what I find astonishing is that the performance of wild OOD detection surpasses even that of outlier exposure, showcasing its remarkable effectiveness.

- The method is supported by a theoretical examination of its generalization bounds, ensuring a solid foundation for reliable ML.

**Weaknesses:**

- One of my major concerns with the paper lies in the discrepancy assumption stated in Theorem 1. To validate this, additional experiments with diverse datasets are imperative to ascertain whether the assumption is consistently met in practical scenarios. While I acknowledge that assumptions are essential for theoretical foundations, and the experiments in Appendix F provide a good start, they are insufficient for a comprehensive evaluation. I recommend extending the empirical analysis to additional datasets (e.g., CIFAR-10 as ID and another data as OOD), with the goal of thoroughly investigating the generality of the assumption and its ability to underpin SAL robustly.

- In the majority of the experiments conducted, the value of pi is set to 0.1. To ensure a comprehensive evaluation, it is crucial to conduct additional experiments with varied values of pi.

**Questions:**

Please address the issues outlined in the weaknesses. This work is solid and good. So addressing all of my concerns comprehensively will lead me to reconsider and potentially raise my score.

---

> ### Author Response · Authors · 2023-11-14
> **Response to Reviewer ZN1X--part I**
>
> We are deeply encouraged that you recognize our method to be novel, significant, and solid in both the algorithm and theory and with remarkable empirical results.
>
> Your summary and comments are insightful and spot-on :)
>
> **A1. Verification of the assumptions on additional datasets**
>
> Thank you for the suggestion! As suggested, we verified the assumption of distribution discrepancy using CIFAR-10 as ID and five other OOD datasets, i.e.,  SVHN, PLACES365, LSUN-C, TEXTURES, and LSUN-R. The result is shown as follows and in **Appendix Section W**, and we can conclude that $\zeta$ can indeed satisfy the regulatory condition in Theorem 2, i.e., $\zeta > 1.011\sqrt{\pi}$.
>
> #### SVHN
> |  $\pi$      | 0.05|    0.1     | 0.2     |   0.5 |        0.7 |      0.9   | 1.0 |
> | ------ | ----- | ----- |----- | ----- |----- | ----- |----- |
> | $\zeta$ | 0.26 | 0.37 |0.49 |  0.71 |0.97 |  1.24| 1.36|
> | 1.011 $\sqrt{\pi}$ |  0.23 |0.32| 0.45 |0.71| 0.84| 0.96| 1.0 |
>
> #### PLACES365
> |  $\pi$      | 0.05|    0.1     | 0.2     |   0.5 |        0.7 |      0.9   | 1.0 |
> | ------ | ----- | ----- |----- | ----- |----- | ----- |----- |
> | $\zeta$ | 0.28 | 0.33 |0.53 |   0.77| 0.85|0.98  | 1.04|
> | 1.011 $\sqrt{\pi}$ |  0.23 |0.32| 0.45 |0.71| 0.84| 0.96| 1.0 |
>
> #### LSUN-C
> |  $\pi$      | 0.05|    0.1     | 0.2     |   0.5 |        0.7 |      0.9   | 1.0 |
> | ------ | ----- | ----- |----- | ----- |----- | ----- |----- |
> | $\zeta$ | 0.29 | 0.34 |0.47 |  0.72 | 0.87| 1.09 | 1.20  |
> | 1.011 $\sqrt{\pi}$ |  0.23 |0.32| 0.45 |0.71| 0.84| 0.96| 1.0 |
>
> #### TEXTURES
> |  $\pi$      | 0.05|    0.1     | 0.2     |   0.5 |        0.7 |      0.9   | 1.0 |
> | ------ | ----- | ----- |----- | ----- |----- | ----- |----- |
> | $\zeta$ |  0.28| 0.33 |0.46 | 0.74  |0.85 |  0.96| 1.05|
> | 1.011 $\sqrt{\pi}$ |  0.23 |0.32| 0.45 |0.71| 0.84| 0.96| 1.0 |
>
> #### LSUN-R
> |  $\pi$      | 0.05|    0.1     | 0.2     |   0.5 |        0.7 |      0.9   | 1.0 |
> | ------ | ----- | ----- |----- | ----- |----- | ----- |----- |
> | $\zeta$ | 0.28 | 0.35 |0.47 | 0.73  | 0.87|  1.10 | 1.22 |
> | 1.011 $\sqrt{\pi}$ |  0.23 |0.32| 0.45 |0.71| 0.84| 0.96| 1.0 |

---

> > ### Author Response · Authors · 2023-11-14
> > **Response to Reviewer ZN1X--part II**
> >
> > **A2. Additional experiment results with varying $\pi$**
> >
> > We absolutely agree with this concern. We would like to point the reviewer to the **Appendix Table 6** of the original submission, where we report the OOD detection result and the filtering error on SVHN with different mixing ratios $\pi$. The result aligns well with our observation of the bounds presented in Section 4.1 of the main paper.
> >
> > During rebuttal, we provide additional results on more OOD datasets with varying $\pi$, i.e., 0.05, 0.2, 0.5, 0.9, and contrast with the baselines, which are added to **Appendix Section T**, and also shown below (CIFAR-100 as the in-distribution dataset). We found that the advantage of SAL still holds.
> >
> > |        | SVHN|         | Places365  |       |LSUN-C |        |LSUN-R |     | Textures|        | ID ACC |
> > | ------ | ----- | ----- |----- | ----- |----- | ----- |----- | ----- |----- | ----- |----- |
> > | $\pi=0.05$|
> > | Method | FPR95 | AUROC |FPR95 |  AUROC |FPR95 | AUROC |FPR95 | AUROC |FPR95 | AUROC | |  |  |
> > |   OE | 2.78 | 98.84|63.63 |80.22 | 6.73| 98.37| 2.06| 99.19 | 32.86| 90.88| 71.98|
> > | Energy w/ OE|2.02 | 99.17| 56.18|  83.33| 4.32| 98.42| 3.96| 99.29|  40.41 | 89.80 | 73.45|
> >  | WOODS |0.26 |99.89 |32.71   | 90.01| **0.64**| 99.77| **0.79**| 99.10 | 12.26| 94.48| 74.15|
> >  |SAL (Ours)| **0.17** | **99.90** | **6.21** | **96.87** |0.94 | **99.79**|0.84 |**99.37** |**5.77** | **97.12**|73.99 | |
> > | $\pi=0.2$|
> > | Method | FPR95 | AUROC |FPR95 |  AUROC |FPR95 | AUROC |FPR95 | AUROC |FPR95 | AUROC | |  |  |
> > |   OE | 2.59|  98.90| 55.68| 84.36| 4.91|99.02 | 1.97| 99.37|25.62 |93.65 | 73.72|
> > | Energy w/ OE|1.79 | 99.25| 47.28|86.78 |  4.18| 99.00| 3.15| 99.35|36.80 | 91.48| 73.91|
> > | WOODS |  0.22| 99.82 | 29.78 | 91.28 | 0.52|99.79 | 0.89|99.56 |10.06 |95.23 |73.49|
> > |SAL (Ours)| **0.08** | **99.92**| **2.80**| **99.31**|**0.05**| **99.94** | **0.02**| **99.97**| **5.71**| **98.71**| 73.86|
> > | $\pi=0.5$|
> > | Method | FPR95 | AUROC |FPR95 |  AUROC |FPR95 | AUROC |FPR95 | AUROC |FPR95 | AUROC | |  |  |
> > |   OE |  2.86| 99.05| 40.21| 88.75| 4.13|99.05  | 1.25| 99.38|22.86 | 94.63| 73.38|
> > | Energy w/ OE| 2.71| 99.34| 34.82|90.05 | 3.27|99.18 |  2.54|99.23| 30.16| 94.76|72.76|
> > | WOODS | 0.17| 99.80| 21.87| 93.73| 0.48| 99.61| 1.24| 99.54  | 9.95|95.97 | 73.91|
> > |SAL (Ours)| **0.02** | **99.98** |  **1.27**| **99.62**| **0.04**| **99.96** | **0.01**| **99.99**| **5.64**|**99.16** |73.77|
> > | $\pi=0.9$|
> > | Method | FPR95 | AUROC |FPR95 |  AUROC |FPR95 | AUROC |FPR95 | AUROC |FPR95 | AUROC | |  |  |
> > |   OE | 0.84 | 99.36| 19.78| 96.29|  1.64| 99.57  | 0.51| 99.75  |12.74  |94.95 | 72.02 |
> > | Energy w/ OE| 0.97| 99.64|17.52 |96.53 | 1.36|99.73 |0.94 | 99.59  | 14.01 | 95.73| 73.62|
> > | WOODS | 0.05 | 99.98| 11.34| 95.83| 0.07 | **99.99** | 0.03| **99.99**|6.72 |98.73 | 73.86|
> > |SAL (Ours)|  **0.03** | **99.99** |  **2.79**|**99.89** |**0.05**| **99.99**| **0.01**| **99.99**| **5.88**| **99.53**| 74.01|

---

### Official Review · Reviewer_dSEG · 2023-10-29

**Soundness:** 2 fair
**Presentation:** 3 good
**Contribution:** 2 fair
**Rating:** 6
**Confidence:** 5

**Summary:**

Leveraging unlabeled data has shown potential in enhancing the safety and reliability of machine learning models for out-of-distribution (OOD) detection, despite the challenges posed by the heterogeneity of both in-distribution (ID) and OOD data. This paper introduces SAL (Separate And Learn), a novel learning framework that addresses the existing gap in understanding how unlabeled data aids OOD detection by providing strong theoretical guarantees and empirical effectiveness. SAL operates by isolating potential outliers from the unlabeled data, training an OOD classifier with these outliers and labeled ID data, and achieving state-of-the-art performance on standard benchmarks, thereby validating the theoretical framework and its components.

**Strengths:**

1. The manuscript presents some theoretical analyses as well as a number of intriguing illustrations.
2. The experimental results look promising, with comparisons made against numerous baseline methods.

**Weaknesses:**

1. The manuscript lacks crucial baselines, such as [1] and [2], which are essential for a comprehensive evaluation, and fails to provide an analysis or comparison with them.
2. The essential topic of the article is weakly supervised out-of-distribution detection, although it is described from different perspectives.

[1] Zhou, Zhi, et al. "Step: Out-of-distribution detection in the presence of limited in-distribution labeled data." Advances in Neural Information Processing Systems 34 (2021): 29168-29180.

[2] He, Rundong, et al. "Topological structure learning for weakly-supervised out-of-distribution detection." arXiv preprint arXiv:2209.07837 (2022).

**Questions:**

1. Near OOD Scenario: It is unclear how effective the method presented in this article would be in a near OOD scenario, such as treating the first 50 classes of CIFAR-100 as ID and the last 50 classes as OOD. This specific situation could pose challenges since near OOD samples may share similarities with the ID data, potentially affecting the method's performance.
2. Generalization Performance Across Different Backbones: The article does not provide information on how well the method generalizes when different backbone architectures are used. The performance stability of the method when transitioning between various model architectures is a critical aspect to consider for its widespread applicability.
3. Generalization Performance on unseen OOD data: There is no discussion on the method's effectiveness when the test OOD data and the OOD data in the unlabeled set are not identically distributed. Understanding how the method handles such disparities is crucial for evaluating its robustness in real-world scenarios.

---

> ### Author Response · Authors · 2023-11-14
> **Response to Reviewer dSEG--part I**
>
> We thank you for recognizing our method to be novel with promising results. We thank the reviewer for the thorough comments and suggestions, which we address below:
>
> **A1. Baselines**
>
> As suggested, we have additionally compared with the two related works (TSL [1] and STEP [2]). To ensure a fair comparison, we strictly follow the experimental setting in TSL [1], and rerun SAL under the identical setup. The comparison on CIFAR-100 is shown as follows. Accordingly, we have also added discussion and proper citations of the mentioned papers in the revised paper (See **related work section and Appendix Section U**). We thank you for pointing them out!
>
> |           |LSUN-C |        |LSUN-R |     |
> | ------ | ----- | ----- |----- | ----- |
> | Method | FPR95 | AUROC |FPR95 |  AUROC |
> | STEP | **0.00**  |99.99 |9.81 |97.87|
> |   TSL |**0.00** | **100.00**| 1.76|  99.57 |
> |SAL (Ours)|**0.00**| 99.99 |**0.58**| **99.95**|
>
> **A2. Discussion on weakly supervised OOD detection**
>
> We agree that weakly supervised OOD detection is indeed similar to the problem setting of SAL. We have already updated the **related work section** of our paper and included more discussions/citations on weakly supervised OOD detection. Thank you for pointing this out!
>
> **A3. Near-OOD Scenario**
>
> We are glad you bring that up. We have already evaluated the near-OOD detection in **Appendix J**. Specifically, we use the CIFAR-10 as the in-distribution data and the CIFAR-100 as the OOD data in the wild. During test time, we use the test set of CIFAR-100 as the OOD for evaluation. With a mixing ratio $\pi$ of 0.1, SAL achieves an FPR95 of 24.51% and AUROC of 95.55% compared to 38.92% (FPR95) and 93.27% (AUROC) of WOODS.
>
>
> In addition, we follow the suggested data setting by the reviewer, i.e., the first 50 classes of CIFAR-100 as ID and the last 50 classes as OOD. The comparison with the most competitive baseline is reported as follows. We have also added the new results to **Appendix Section J** in the revised manuscript.
>
> |        |  CIFAR-50|              | |
> | ------ | ----- | ----- |----- |
> | Method | FPR95 | AUROC |ID ACC|
> | WOODS | 41.28 | 89.74| 74.17|
> |SAL (Ours)| **29.71**|**93.13** |73.86|

---

> > ### Author Response · Authors · 2023-11-14
> > **Response to Reviewer dSEG--part II**
> >
> > **A4. Generalization Performance Across Different Backbones**
> >
> > As suggested, we have additionally tried ResNet-18 and ResNet-34 as the network architectures---which are among the most used in OOD detection literature. The comparison with the baselines on CIFAR-100 is shown in the following tables and **Appendix Section V**, where SAL outperforms all the baselines across different architectures. These additional results support the effectiveness of our approach.
> > #### ResNet-18
> > |        | SVHN|         | Places365  |       |LSUN-C |        |LSUN-R |     | Textures|   | Average |    | ID ACC |
> > | ------ | ----- | ----- |----- | ----- |----- | ----- |----- | ----- |----- | ----- |----- |----- |----- |
> > | Method | FPR95 | AUROC |FPR95 |  AUROC |FPR95 | AUROC |FPR95 | AUROC |FPR95 | AUROC |FPR95 | AUROC |  |
> > | MSP    |81.32 | 77.74|  83.06 | 74.47|70.11|83.51 | 82.46| 75.73|  85.11|73.36 |80.41 | 76.96| 78.67|
> > |ODIN| 40.94 | 93.29| 87.71|71.46|28.72|94.51|79.61|82.13|83.63|72.37| 64.12|82.75 |78.67|
> > |Mahalanobis| 22.44 |95.67| 92.66 |61.39| 68.90| 86.30| 23.07| 94.20 |62.39| 79.39|53.89|83.39|78.67|
> > |Energy | 81.74|84.56|  82.23|76.68|34.78|93.93|73.57|82.99| 85.87|74.94|71.64| 82.62|78.67 |
> > |KNN| 83.62|72.76| 82.09|80.03|65.96|	84.82 | 71.05	|81.24|76.88|	77.90| 75.92| 79.35| 78.67|
> > |ReAct| 70.81|88.24|81.33|76.49|39.99|92.51|54.47|89.56|59.15|87.96|61.15|86.95|78.67|
> > |DICE| 54.65| 88.84| 79.58|  77.26| 0.93| 99.74| 49.40| 91.04|65.04| 76.42| 49.92|86.66 |78.67 |
> > |CSI|49.98 |89.57| 82.87| 75.64 |76.39| 80.38 |74.21 |83.34| 58.23| 81.04 |68.33 |81.99| 74.23|
> > | KNN+| 43.21 |90.21| 84.62| 74.21| 50.12| 82.48| 76.92| 80.81| 63.21| 84.91| 63.61| 82.52| 77.03|
> > |OE |3.29  |97.93 | 62.90 | 80.23 | 7.07 | 95.93 | 4.06 | **97.98** | 33.27 | 90.03 | 22.12 |92.42 | 74.89 |
> > |Energy (w/ OE) |3.12| 94.27| 59.38| 82.19| 9.12| 91.23 |7.28 |95.39| 43.92| 90.11|24.56|90.64| 77.92|
> > |WOODS| 3.92| 96.92 |33.92| 86.29| 5.19| 94.23 |**2.95**| 96.23 |11.95 |94.65 |11.59|93.66 | 77.54|
> > |SAL (Ours)|**2.29**| **97.96** |**6.29** |**96.66** |**3.92**| **97.81**| 4.87 |97.10| **8.28** |**95.95** |**5.13**|**97.10**|77.71 |
> >
> > #### ResNet-34
> > |        | SVHN|         | Places365  |       |LSUN-C |        |LSUN-R |     | Textures|   | Average |    | ID ACC |
> > | ------ | ----- | ----- |----- | ----- |----- | ----- |----- | ----- |----- | ----- |----- |----- |----- |
> > | Method | FPR95 | AUROC |FPR95 |  AUROC |FPR95 | AUROC |FPR95 | AUROC |FPR95 | AUROC |FPR95 | AUROC |  |
> > | MSP    |78.89 | 79.80|  84.38|  74.21|  83.47|  75.28|  84.61 | 74.51|  86.51 | 72.53 | 83.12|  75.27|79.04|
> > |ODIN| 70.16| 84.88 |82.16| 75.19 |76.36 |80.10| 79.54| 79.16 |85.28 |75.23| 78.70| 79.11|79.04|
> > |Mahalanobis| 87.09| 80.62 |84.63 |73.89 |84.15| 79.43 |83.18 |78.83 |61.72 |84.87| 80.15 |79.53|79.04|
> > |Energy| 66.91 |85.25 |81.41 |76.37 |59.77 |86.69 |66.52 |84.49 |79.01| 79.96| 70.72| 82.55| 79.04|
> > | KNN|81.12 |73.65  | 79.62| 78.21 | 63.29| 85.56 | 73.92 | 79.77| 73.29 | 80.35 | 74.25 | 79.51 | 79.04 |
> >  |ReAct|  82.85 | 70.12 |81.75 | 76.25| 80.70 | 83.03 |67.40 | 83.28 |74.60| 81.61 | 77.46 | 78.86 |79.04|
> > |DICE|83.55 | 72.49  | 85.05 | 75.92 | 94.05 | 73.59 |75.20 | 80.90 |79.80 | 77.83 |83.53 | 76.15 | 79.04 |
> > |CSI|44.53 |92.65| 79.08| 76.27 |75.58| 83.78 |76.62 |84.98| 61.61| 86.47 |67.48 |84.83|77.89|
> > |KNN+| 39.23 |92.78| 80.74| 77.58| 48.99| 89.30| 74.99| 82.69| 57.15| 88.35| 60.22| 86.14|78.32|
> > |OE |2.11  |98.23 | 60.12 | 83.22 | 6.08 | 96.34 | 3.94 | 98.13 | 30.00 | 92.27 | 20.45 | 93.64|  75.72|
> > |Energy (w/ OE) |1.94| 95.03| 68.84| 85.94| 7.66| 92.04 |6.86 |97.63| 40.82| 93.07| 25.22|92.74 | 78.75|
> > |WOODS|2.08| 97.33 |25.37| 88.93| 4.26| 97.74 |1.05| 97.30 |8.85 |96.86 | 8.32|  95.63| 78.97|
> > |SAL (Ours)| **0.98** |**99.94** |**2.98** |**99.08**| **0.07** |**99.94**| **0.03**| **99.96**| **4.01** |**98.83**|  **1.61**| **99.55**| 78.01 |

---

> > > ### Author Response · Authors · 2023-11-14
> > > **Response to Reviewer dSEG--part III**
> > >
> > > **A5. Generalization Performance on unseen OOD data**
> > >
> > > Another great point! In the original submission, we have included the result where the OOD data in the wild is different from the test OOD data (please see **Appendix Section I and Table 8**). Specifically, we use 300K RANDOM IMAGES as the wild OOD dataset and SVHN, PLACES365, LSUN-C, LSUN-RESIZE, and TEXTURES as the test OOD data. We observe that SAL can perform competitively on unseen OOD datasets as well, compared to the most relevant baseline WOODS.
> > >
> > > In addition, following [1], we use the CIFAR-100 as ID, TINc/TINr dataset as the OOD in the wild dataset and TINr/TINc as the test OOD. The comparison with baselines is shown below and in **Appendix Section I**, where the strong performance of SAL still holds.
> > >
> > > |        | TINr|         | TINc|         |
> > > | ------ | ----- | ----- |----- | ----- |
> > > | Method | FPR95 | AUROC |FPR95 | AUROC |
> > > |STEP| 72.31| 74.59|48.68  |91.14 |
> > > | TSL| 57.52 | 82.29| 29.48|94.62 |
> > > |SAL (Ours)|**43.11** |**89.17** | **19.30**| **96.29**|
> > >
> > >
> > > [1] He, Rundong, et al. "Topological structure learning for weakly-supervised out-of-distribution detection." arXiv preprint arXiv:2209.07837 (2022).
> > >
> > > [2] Zhou, Zhi, et al. "Step: Out-of-distribution detection in the presence of limited in-distribution labeled data." Advances in Neural Information Processing Systems 34 (2021): 29168-29180.

---

> > > > ### Comment · Reviewer_dSEG · 2023-11-15
> > > > **Reply to the author**
> > > >
> > > > Thank you for the author's response, which solves most of my questions. I will improve my score. Moreover, please add your response to the final version.

---

> ### Author Response · Authors · 2023-11-15
> **Response to Reviewer dSEG**
>
> Dear Reviewer dSEG,
>
> Thank you for taking the time to read our response and increasing your score!  We are glad to hear that the response solved your concern. We will make sure the discussion and results will be added in the final version!
>
> Thanks,
>
> Authors

---

### Official Review · Reviewer_XJRS · 2023-10-30

**Soundness:** 3 good
**Presentation:** 2 fair
**Contribution:** 3 good
**Rating:** 8
**Confidence:** 3

**Summary:**

This paper introduces a novel setting for OOD detection, termed as ”wild OOD detection,” building upon the foundation established by the preceding work ”Training OOD Detectors in their Natural Habitats.” A novel methodology, denoted as SAL, is presented, encapsulating at wo-stage process comprising filtering and classification components. Empirical evaluations have demonstrated that SAL achieves SOTA performance, boasting substantial improvements over existing methods. Additionally, theoretical underpinnings are provided to bolster the credibility and effectiveness of SAL.

**Strengths:**

1. A theory has been established to investigate aspects of separability and learnability. This contribution is both novel and significant.

2. Experimental evaluations conducted on standard benchmarks demonstrate that SAL achieves SOTA performance.

3. A novel method grounded in theory has been developed to advance safe machine learning practices. Theory serves as a crucial driver in this endeavour. I am very happy to see the novel work on provable OOD detection.

**Weaknesses:**

1. Could you provide explanations or conduct experiments to elucidate the factors contributing to the decreased ID accuracy depicted in Table 1?

2. Why is pi set to 0.1 in most experiments? Could you conduct additional experiments to investigate whether pi remains robust across a range of values? Furthermore, does the performance of pi align with theoretical predictions?

3. It appears that the top singular vector is crucial for SAL. Have you conducted any experiments to demonstrate the performance when considering the top 2, top 3, ..., top k singular vectors?

4. What would occur if you were to use the gradient norm in place of the top singular vector? Could you elucidate the rationale behind opting for the top singular vector instead of the norm?

**Questions:**

Please refer to the Weaknesses.

---

> ### Author Response · Authors · 2023-11-14
> **Response to Reviewer XJRS**
>
> We are glad to see that the reviewer finds our work significant and novel from various perspectives. We thank the reviewer for the thorough comments and suggestions. We are happy to clarify as follows:
>
> **A1. ID accuracy**
>
> Great observation! As explained in **paragraph 2 of Section 5.2** in the original submission, the slight discrepancy is due to that our method only observes 25,000 labeled ID samples, whereas baseline methods (without using wild data) utilize the entire CIFAR training data with 50,000 samples. We have used bold fonts to highlight it in the revision.
>
> **A2. Additional experiment results with varying $\pi$**
>
> Thank you for your suggestion! In our main experiment, we default $\pi$ to be 0.1, which strictly follows the original setting in WOODS [1]. This reflects the practical scenario that the majority of test data may remain ID. Compared to larger $\pi$, our setting with $\pi=0.1$ is also more challenging due to limited information of OOD data.
>
> We would like to point the reviewer to the **Appendix Table 6** of the original submission, where we report the OOD detection result and the filtering error on SVHN with different mixing ratios $\pi$. The result aligns well with our observation of the bounds presented in Section 4.1 of the main paper.
>
> During rebuttal, we provide additional results on more OOD datasets with varying $\pi$, i.e., 0.05, 0.2, 0.5, 0.9, and contrast with the baselines, which are added to **Appendix Section T**, and also shown below (CIFAR-100 as the in-distribution dataset). We found that the advantage of SAL still holds.
>
> |        | SVHN|         | Places365  |       |LSUN-C |        |LSUN-R |     | Textures|        | ID ACC |
> | ------ | ----- | ----- |----- | ----- |----- | ----- |----- | ----- |----- | ----- |----- |
> | $\pi=0.05$|
> | Method | FPR95 | AUROC |FPR95 |  AUROC |FPR95 | AUROC |FPR95 | AUROC |FPR95 | AUROC | |  |  |
> |   OE | 2.78 | 98.84|63.63 |80.22 | 6.73| 98.37| 2.06| 99.19 | 32.86| 90.88| 71.98|
> | Energy w/ OE|2.02 | 99.17| 56.18|  83.33| 4.32| 98.42| 3.96| 99.29|  40.41 | 89.80 | 73.45|
>  | WOODS |0.26 |99.89 |32.71   | 90.01| **0.64**| 99.77| **0.79**| 99.10 | 12.26| 94.48| 74.15|
>  |SAL (Ours)| **0.17** | **99.90** | **6.21** | **96.87** |0.94 | **99.79**|0.84 |**99.37** |**5.77** | **97.12**|73.99 | |
> | $\pi=0.2$|
> | Method | FPR95 | AUROC |FPR95 |  AUROC |FPR95 | AUROC |FPR95 | AUROC |FPR95 | AUROC | |  |  |
> |   OE | 2.59|  98.90| 55.68| 84.36| 4.91|99.02 | 1.97| 99.37|25.62 |93.65 | 73.72|
> | Energy w/ OE|1.79 | 99.25| 47.28|86.78 |  4.18| 99.00| 3.15| 99.35|36.80 | 91.48| 73.91|
> | WOODS |  0.22| 99.82 | 29.78 | 91.28 | 0.52|99.79 | 0.89|99.56 |10.06 |95.23 |73.49|
> |SAL (Ours)| **0.08** | **99.92**| **2.80**| **99.31**|**0.05**| **99.94** | **0.02**| **99.97**| **5.71**| **98.71**| 73.86|
> | $\pi=0.5$|
> | Method | FPR95 | AUROC |FPR95 |  AUROC |FPR95 | AUROC |FPR95 | AUROC |FPR95 | AUROC | |  |  |
> |   OE |  2.86| 99.05| 40.21| 88.75| 4.13|99.05  | 1.25| 99.38|22.86 | 94.63| 73.38|
> | Energy w/ OE| 2.71| 99.34| 34.82|90.05 | 3.27|99.18 |  2.54|99.23| 30.16| 94.76|72.76|
> | WOODS | 0.17| 99.80| 21.87| 93.73| 0.48| 99.61| 1.24| 99.54  | 9.95|95.97 | 73.91|
> |SAL (Ours)| **0.02** | **99.98** |  **1.27**| **99.62**| **0.04**| **99.96** | **0.01**| **99.99**| **5.64**|**99.16** |73.77|
> | $\pi=0.9$|
> | Method | FPR95 | AUROC |FPR95 |  AUROC |FPR95 | AUROC |FPR95 | AUROC |FPR95 | AUROC | |  |  |
> |   OE | 0.84 | 99.36| 19.78| 96.29|  1.64| 99.57  | 0.51| 99.75  |12.74  |94.95 | 72.02 |
> | Energy w/ OE| 0.97| 99.64|17.52 |96.53 | 1.36|99.73 |0.94 | 99.59  | 14.01 | 95.73| 73.62|
> | WOODS | 0.05 | 99.98| 11.34| 95.83| 0.07 | **99.99** | 0.03| **99.99**|6.72 |98.73 | 73.86|
> |SAL (Ours)|  **0.03** | **99.99** |  **2.79**|**99.89** |**0.05**| **99.99**| **0.01**| **99.99**| **5.88**| **99.53**| 74.01|
>
> [1] Julian Katz-Samuels et al. Training ood detectors in their natural habitats. In International Conference on Machine Learning, 2022.
>
> **A3. Additional experiments on multiple principal components**
>
> Another great point! In our original submission, we reported results using multiple principal components in **Appendix Section K**. We observed that using the top 1 singular vector for projection achieves the best performance.
>
> **A4. Discussion on using gradient norm**
>
> We have already evaluated the GradNorm score as suggested in **Table 2**, where we replace the filtering score in SAL with the GradNorm score and then train the OOD classifier. The result underperforms SAL, showcasing the effectiveness of our filtering score.
>
> We have also extensively discussed the design rationale of the filtering scores of SAL in **Section 3.1**, saying that the scores in SAL for ID and OOD data are shown to be provably well-separated (**Remark 1**) and thus ensure a low filtering error, while the norm of the gradient is not. Both the theoretical result and empirical verification can demonstrate the advantage of SAL compared with GradNorm.

---

> > ### Comment · Reviewer_XJRS · 2023-11-22
> > **Official Comment by Reviewer XJRS**
> >
> > Thank you for your reply. The clarifications and additional experimental results solved all of my concerns. Therefore, I have raised my rating of this work.

---

> > > ### Author Response · Authors · 2023-11-22
> > > **Thank you!**
> > >
> > > Dear Reviewer XJRS,
> > >
> > > Thank you for taking the time to read our response and increasing your score! We are glad to hear that the response solved your concern. We will make sure the discussion and results are added in the final version!
> > >
> > > Thanks,
> > >
> > > Authors

---

### Official Review · Reviewer_BtG8 · 2023-11-01

**Soundness:** 4 excellent
**Presentation:** 3 good
**Contribution:** 3 good
**Rating:** 6
**Confidence:** 2

**Summary:**

This paper presents a novel framework for Out-Of-Distribution (OOD) detection, named SAL, which aims to improve machine learning models through regularization using unlabeled data. SAL comprises two main components: (1) Filtering–distinguishing potential outliers from the general dataset, and (2) Classification utilizing the identified candidate outliers to train an OOD classifier. The paper includes pertinent theoretical proofs to substantiate the proposed method, and experimental results are provided to demonstrate its effectiveness.

**Strengths:**

1 SAL’s methodology is structured around two distinct phases—screening and classification—which can be independently optimized, offering enhanced flexibility.
2 Utilizing a Large Volume of Unlabeled Data: SAL effectively leverages substantial amounts of unlabeled data to extract valuable information, thereby bolstering its detection capabilities.
3 Theoretical Support: Beyond its impressive empirical performance, SAL is underpinned by robust theoretical foundations.

**Weaknesses:**

1 In scenarios where the actual OOD data markedly diverges from the outliers present in the unlabeled dataset, there arises a question regarding the preservation of SAL’s performance.
 2 The efficacy of SAL is significantly influenced by the quality of the unlabeled data employed, indicating a substantial dependence on data integrity.

**Questions:**

1 Could you design experiments to further confirm and answer issues in weaknesses?
2 In table 1, how can this discrepancy be accounted for in datasets or scenarios that perform well in other methods but have degraded performance (ID ACC) in SAL?

---

> ### Author Response · Authors · 2023-11-14
> **Response to Reviewer BtG8**
>
> We thank the reviewer for the thorough comments and suggestions. We are encouraged that you recognize our method to be novel and effective, and with robust theoretical analysis. We address your questions below:
>
>
> **A1. Different outlier dataset from the actual test OOD**
>
> We are glad you bring that up! In the original submission, we have included the result where the OOD data in the wild is different from the test OOD data (please see **Appendix Section I and Table 8**). Specifically, we use 300K RANDOM IMAGES from outlier exposure [1] to create the wild OOD training dataset. We evaluate on SVHN, PLACES365, LSUN-C, LSUN-RESIZE, and TEXTURES as the test unseen OOD data. We observe that SAL can perform competitively on unseen OOD datasets as well, compared to the most relevant baseline WOODS.
>
> [1] Hendrycks et al. Deep anomaly detection with outlier exposure. In Proceedings of the International Conference on Learning Representations, 2019.
>
> **A2. Quality of the unlabeled data**
>
> To address your concern, we have designed the following experiment where the quality of the unlabeled data deteriorates. The results of SAL and competitive baselines are shown in the table below and have also been added to the **Appendix Section S**.
>
> Specifically, we corrupt the outlier data in the wild with additive Gaussian noise [2]. As such, the filtered candidate outliers will have a much lower quality compared to the outliers in SAL. We use the CIFAR-10 as the in-distribution dataset and keep other configurations the same.
>
> |        | SVHN|         | Places365  |       |LSUN-C |        |LSUN-R |     | Textures|   | Average |    | ID ACC |
> | ------ | ----- | ----- |----- | ----- |----- | ----- |----- | ----- |----- | ----- |----- |----- |----- |
> | Method | FPR95 | AUROC |FPR95 |  AUROC |FPR95 | AUROC |FPR95 | AUROC |FPR95 | AUROC |FPR95 | AUROC |  |
> |   OE | 23.11 |  86.61| 32.01| 86.27| 22.98| 82.75 | 19.53| 87.43  | 25.68 |84.46 |  24.66 |85.50 |93.81 |
>  | Energy w/ OE| 26.76 | 85.91|26.09 | 87.48|22.32 | 82.26|  22.69 | 85.77|27.49 |82.18 | 25.07 |84.72 |92.38 |
> | WOODS | 18.33 | 89.83| 23.45| 90.04| 19.70|  84.27| 17.79| 90.82| 22.37| 84.83| 20.33|   87.95| 94.00|
> |SAL (Ours)|**15.23** | **91.22**| **18.23**| **93.51**| **14.62**| **89.04**| **13.93**| **91.82**| **18.58**| **92.42**|**16.12** |**91.60** | 93.91|
>
>
> [2] Hendrycks et.al., Benchmarking neural network robustness to common corruptions and surface variations. In Proceedings of the International Conference on Learning Representations, 2019.
>
> **A3. ID accuracy**
>
> Great observation! As explained in **paragraph 2 of Section 5.2** in the original submission, the slight discrepancy is due to that our method only observes 25,000 labeled ID samples, whereas baseline methods (without using wild data) utilize the entire CIFAR training data with 50,000 samples. We have used bold fonts to highlight it in the revision.

---

### Author Response · Authors · 2023-11-14
**General Response**

We thank all the reviewers for their time and valuable comments. We are encouraged to see that ALL reviewers find our paper **novel** and **significant** (BtG8, XJRS, dSEG, ZN1X). Reviewers also recognize our work providing a **solid and crucial foundation** for reliable ML (XJRS, ZN1X), and the results are **promising**, **impressive**, achieving **state-of-the-art** (BtG8, XJRS, dSEG, ZN1X).

As recognized by multiple reviewers, the significance of our work can be summarized as follows:

- Our work offers a new algorithmic framework to effectively exploit the unlabeled wild data for OOD detection. This algorithm has broad utility since unlabeled data is ubiquitous in many ML applications, but the principled way of utilizing them for OOD detection is currently lacking in the field.
- Moreover, we provide new theories from the lens of separability and learnability, to formally justify the two components in our algorithm.
- Empirically, we show that SAL can be broadly applicable to modern neural networks, and establish state-of-the-art performance on common OOD detection tasks, reinforcing our theoretical insights.

We respond to each reviewer's comments in detail below. We also revised the manuscript according to the reviewers' suggestions (blue text), and we believe this makes our paper stronger.

---

### Meta-Review · Area_Chair_uZBi · 2023-12-09

**Metareview:**

The paper introduces a novel framework for Out-Of-Distribution (OOD) detection, named SAL, with the goal of enhancing machine learning models through regularization using unlabeled data. SAL consists of two main components: Filtering, which distinguishes potential outliers from the general dataset, and Classification, utilizing identified outliers to train an OOD classifier. This novel approach is positioned in the context of "wild OOD detection," extending the prior work on "Training OOD Detectors in their Natural Habitats." Empirical evaluations show that SAL achieves state-of-the-art performance, demonstrating substantial improvements over existing methods. The paper supports the proposed methodology with theoretical proofs, reinforcing the credibility and effectiveness of SAL. The reviewers are unanimously positive about the paper's quality and contribution.

**Justification For Why Not Higher Score:**

The paper's impact is slightly limited by the required assumptions for the established theorems and an incomplete positioning and comparisons with some existing and relevant literature (but formulated in different ways).

**Justification For Why Not Lower Score:**

With the above side, the presented results are solid and justify the proposed method's advantage.

---

### Decision · Program_Chairs · 2024-01-16

Accept (poster)